# Bottom RedOx Model (BROM, v.1.1): a coupled benthic-pelagic model for simulation of water and sediment biogeochemistry

E.V.Yakushev[1,2], E.A.Protsenko[2,1], J.Bruggeman[3], P.Wallhead[4], S.V.Pakhomova[5,2], S.Yakubov[2], R.G.J.Bellerby[6,4], R.-M. Couture[1,7]

[1]Norwegian Institute for Water Research (NIVA), Gaustadalléen 21, 0349 Oslo, Norway
[2]P.P.Shirshov Institute of Oceanology RAS, Nakhimovskiy prosp. 36, 117991, Moscow, Russia
[3]Plymouth Marine Laboratory, Prospect Place, The Hoe, Plymouth, United Kingdom
[4]Norwegian Institute for Water Research (NIVA Vest), Thormøhlensgate 53 D, 5006 Bergen, Norway
[5]Norwegian Institute for Air Research (NILU), P.O. Box 100, NO-2027 Kjeller, Norway
[6]State Key Laboratory for Estuarine and Coastal Research, East China Normal University, Shanghai, China
[7]University of Waterloo, Earth and Environmental Sciences, Ecohydrology Group, 200 University Avenue West, N2L3G2, Canada

*Correspondence to*: E.V. Yakushev (eya@niva.no)

**Abstract.** Interactions between seawater and benthic systems play an important role in global biogeochemical cycling. Benthic fluxes of some chemical elements (e.g. C, N, P, O, Si, Fe, Mn, S) alter the redox state and marine carbonate system (i.e. pH and carbonate saturation state), which in turn modulate the functioning of benthic and pelagic ecosystems. The redox state of the near bottom layer in many regions can change with time, responding to the supply of organic matter, physical regime and coastal discharge. Due to the high spatial and temporal variability of the drivers of pelagic-benthic exchange and its sensitivity to environmental and climate change it is difficult to represent these processes though observations alone. We developed a model (BROM) to represents key biogeochemical processes in the water and sediments and to simulate changes occurring in the bottom boundary layer. BROM consists of a transport module (BROM-transport) and several biogeochemical modules that are fully compatible with the Framework for the Aquatic Biogeochemical Models, allowing independent coupling to hydrophysical models in 1D, 2D or 3D. We demonstrate that BROM is capable of simulating the seasonality in production and mineralization of organic matter as well as the mixing that leads to variations in redox conditions. BROM can be used for analyzing and interpreting data on sediment-water exchange, and for simulating the consequences of forcings such as climate change, external nutrient loading, ocean acidification, carbon storage leakage, and point-source metal pollution.

**Key Words** – modeling; Bottom Boundary Layer; benthic fluxes; nutrient cycles; anoxic conditions; carbonate system.

## 1 Background

Oxygen depletion and anoxia are increasingly common features observed in the World Ocean, inland seas and coastal areas. Observations show a decline in dissolved oxygen concentrations at continental margins in many regions and this has been linked to both an increase in anthropogenic nutrient loadings and a decrease in vertical

mixing e.g. (Diaz and Rosenberg, 2008; Rabalais et al., 2002; Richardson and Jørgensen, 1996). Although bottom waters may be permanently oxic or anoxic, they oscillate seasonally between these extremes in many water bodies (Morse and Eldridge, 2007). Such oscillations typically result from variation in the supply of organic matter (OM) to the sediment-water interface (SWI), from the hydrophysical regime (mixing/ventilation), and from nutrient supply (river run-off). Frequently, oxic conditions during periods of intense mixing are followed by near-bottom suboxia or anoxia after the seasonal pycnocline forms, restricting aeration of the deeper layers. This occurs for instance on the Louisiana shelf (Morse and Eldridge, 2007; Yu et al., 2015) and in Corpus Christi Bay (McCarthy et al., 2008), the Sea of Azov (Debolskaya et al., 2008), and Elefsis Bay (Pavlidou et al., 2013).

The redox state and oxygenation of near-bottom water varies due to the transport of oxidized and reduced species across the SWI and biogeochemical processes occurring in the sediments (Cooper and Morse, 1996; Jorgensen et al., 1990; Roden and Tuttle, 1992; Sell and Morse, 2006). The sediments generally consume oxygen due to the deposition of labile OM and the presence of reduced forms of chemical elements. Their capacity to exchange oxygen with the pelagic layer is limited, as near bottom water is usually characterized by low water velocity and reduced mixing in the vicinity of the SWI (Glud, 2008). In some cases, a high benthic oxygen demand (BOD) associated with local OM mineralization and low mixing rates can cause anoxia in the bottom water. This may lead to death, migration, or changed behavior of the benthic macro and meio faunal organisms responsible for bioturbation and bioirrigation (Blackwelder et al., 1996; Sen Gupta et al., 1996; Morse and Eldridge, 2007), which in turn can greatly slow down the transport of solid and dissolved species inside the sediments and therefore the rates of oxidative reactions. Under such conditions, sedimentary sulfides can build up, and dissolution of carbonate minerals may come to a halt (Morse and Eldridge, 2007). When oxic conditions return, there can be an "oxygen debt" of reduced species in the water column (Yakushev et al., 2011) which may buffer and delay reoxygenation of the sediments (Morse and Eldridge, 2007).

The processes taking place in the water column and in the sediments are therefore tightly coupled in areas experiencing seasonal hypoxia/anoxia, and an accurate understanding of physical, chemical, and biological processes driving changes in redox conditions is needed to predict the distribution of hypoxia/anoxia in a given environment. Also, the distinct environments of the water column and sediments may be strongly coupled by the exchange of matter on a range of time scales. This "benthic–pelagic coupling" is broadly defined by fluxes of OM to the sediments and return fluxes of inorganic nutrients to the water column. Variations in supply, dynamics and reactivity of OM affect the benthic communities (Pearson and Rosenberg, 1978), the sediment and porewater geochemistry (Berner, 1980), and the nutrient and oxygen fluxes at the SWI (Boudreau, 1997). The impact of OM on the benthos is generally more noticeable in shallow environments such as shelf seas, bays and lakes.

A number of recent studies demonstrate the capability of sophisticated reactive transport codes for integrated modelling of biogeochemical cycles in sediments (Boudreau, 1996; Van Cappellen and Wang, 1996; Couture et al., 2010; Jourabchi et al., 2005; Paraska et al., 2014; Soetaert et al., 1996). The water column redox interface was also specifically targeted in the models of (Konovalov et al., 2006; Yakushev et al., 2006, 2007). However,

the process of integrating of such models with pelagic biogeochemical models to produce benthic-pelagic coupled models has only begun in recent years.

As of the year 2000, benthic-pelagic coupling was largely neglected or crudely approximated in many pelagic biogeochemical and early diagenetic models, which latter can in fact be regarded as benthic biogeochemical models (Soetaert et al., 2000). One of the first fully coupled physical–pelagic–benthic biogeochemical modes was developed for the Goban Spur shelf-break area to examine the impact of in-situ atmospheric conditions on ecosystem dynamics, to understand biogeochemical distributions in the water column and the sediments, and to derive a nitrogen budget for the area. This model was most suited to testing the impact of short-term physical forcing on the ecosystem (Soetaert et al., 2001).

Later, several coupled benthic-pelagic models were produced with an emphasis on studying eutrophication (Cerco et al., 2006; Fennel et al., 2011; Soetaert and Middelburg, 2009) or hypoxia in various locations including Tokyo bay (Sohma et al., 2008), the Baltic Sea (Reed et al., 2011), the North Sea Oyster Grounds (Meire et al., 2013) and Southern Bight (Lancelot et al., 2005). Another model was created to investigate early diagenesis of silica in Scheldt estuary, with benthic-pelagic coupling only of silica (Arndt and Regnier, 2007).

By coupling two quite sophisticated models ECOHAM1 and C.CANDI, a 3D model for the North Sea was created where pelagic model output was used as a forcing for a benthic biogeochemical module (Luff and Moll, 2004). Another physical-biological model for the North Sea, PROWQM, is more complex than ECOHAM1 and has been coupled to a benthic module to simulate seasonal changes of chlorophyll, nutrients and oxygen at the PROVESS north site, south–east of the Shetland Islands (Lee et al., 2002). (Brigolin et al., 2011) developed a spatially explicit model for the northwestern Adriatic coastal zone by coupling a 1D transient early diagenesis model with a 2D reaction-transport pelagic biogeochemical model. Currently, the most known and established coupled model is ERSEM – the European Regional Seas Ecosystem Model that was initially developed as a coastal ecosystem model for the North Sea and which has evolved into a generic tool for ecosystem simulations from shelf seas to the global ocean (Butenschön et al., 2015).

The BROM model described herein is a fully coupled benthic-pelagic model with a special focus on deoxygenation and redox biogeochemistry in the sediments and Benthic Boundary Layer (BBL). The BBL is "the part of the marine environment that is directly influenced by the presence of the interface between the bed and its overlying water" (Dade et al., 2001). Physical scientists tend to prefer the term "bottom boundary layer", but this largely synonymous with the BBL (Thorpe, 2005). Within BROM, the term BBL is used to refer to the lower parts of the fluid bottom boundary layer where bottom friction strongly inhibits current speed and vertical mixing, hence including the viscous and logarithmic sublayers up to at most a few meters above the sediment. This calm-water layer plays a critical role in mediating the interaction of the water column and sediment biogeochemistry and in determining e.g. near-bottom oxygen levels, yet it remains poorly resolved in most physical circulation models. For BROM we have developed an accompanying offline transport module "BROM-transport" that uses output from hydrodynamic water column models but solves the transport-reaction equations for a "full" grid including both water column and sediments. BROM-transport uses greatly increased spatial resolution near to the SWI, and thereby provides explicit spatial resolution of the BBL and sediments.

The goal of this work was to develop a model that captures key biogeochemical processes in the water and

sediment and to analyze the changes occurring in the BBL and SWI. As a result, BROM differs from existing

biogeochemical models in several key respects. BROM features explicit, detailed descriptions of many chemical

transformations under different redox conditions, and tracks the fate of several chemical elements (Mn, Fe, and

S) and compounds ($MnCO_3$, FeS, S0, $S_2O_3$) that rarely appear in other models. BROM also allows for spatially

explicit representations of the vertical structure in the sediments and BBL. This distinguishes it from e.g.

ERSEM (Butenschön et al., 2015), which has a more detailed representation of larger benthic organisms

(meiofauna and different types of macrofauna), but limits its chemistry to the dissolved phase to $CO_2$, $O_2$ and

macronutrients, its benthic bacteria to two functional groups, and its sedimentary vertical structure to an implicit

three-layer representation that relies on equilibrium profiles of solutes and idealized profiles of particulates.

Third, BROM offers a near-comprehensive representation of all processes affecting oxygen levels in the BBL

and sediments, and should therefore provide a useful tool for studies focused on deoxygenation in deep water

and sediments. Finally, BROM is conceived and programmed as a flexible model that can be applied in a broad

range of marine and lake environments and modelling problems. As a component of the Framework for Aquatic

Biogeochemical Modelling (FABM, Bruggeman and Bolding, 2014), BROM can be very easily coupled online

to any hydrodynamic model within the FABM, and can also be driven offline by hydrodynamic model output

saved in NetCDF or text format using the purpose-built offline transport solver BROM-transport.

**2   BROM description**

Here we present the one-dimensional vertical transport and reaction model Bottom RedOx Model, BROM. It

consists of two modules, BROM-biogeochemistry and BROM-transport. BROM-biogeochemistry is based on

ROLM (RedOx Layer Model), a model constructed to simulate basic biogeochemical structure of the water

column oxic/anoxic interface in the Black Sea, Baltic Sea, and Norwegian fjords (He et al., 2012; Stanev et al.,

2014; Yakushev et al., 2009, 2006, 2007, 2011). In BROM–biogeochemistry we extended the list of modelled

compounds and processes (Figure 1). BROM considers interconnected transformations of species of (N, P, Si, C,

O, S, Mn, Fe) and resolves OM in nitrogen currency. OM dynamics include parameterizations of OM production

(via photosynthesis and chemosynthesis) and OM decay via oxic mineralization, denitrification, metal reduction,

sulfate reduction and methanogenesis. In order to provide a detailed representation of changing redox conditions,

OM in BROM is mineralized by several different electron acceptors and dissolved oxygen is consumed during

both mineralization of OM and oxidation of various reduced compounds. Process inhibition in accordance with

redox potential is parameterized by various redox-dependent switches. BROM also includes a module describing

the carbonate equilibria; this allows BROM to be used to investigate acidification and impacts of changing pH

and saturation states on water and sediment biogeochemistry.

The physical domain of BROM-transport spans the water column, BBL and upper layers of the sediments in a

continuous fashion. This allows for an explicit, high-resolution representation of the BBL and upper sediments,

while also allowing the boundary conditions to be moved as far as possible from these foci of interest i.e. to the

air-sea interface and to deep in the sediment.

BROM is built upon an existing modular platform (FABM) and is therefore coded as a set of reusable "lego-brick" components, including the offline transport driver BROM-transport and separate modules for ecology, redox chemistry, and carbonate chemistry. This means that BROM-transport can be used with all biogeochemical modules available in FABM, including e.g. the modules comprising ERSEM, and that BROM biogeochemical modules can be used in all other 1D and 3D hydrodynamic models supported by FABM (e.g., GOTM, GETM, MOM5, NEMO, FVCOM). Individual BROM modules can also be coupled to existing ecological models to expand their scope, e.g. by providing descriptions of redox and carbonate chemistry. Using the FABM framework thus facilitates the transparent and consistent setup of complex biogeochemical reaction networks for the prediction of hypoxia/anoxia while harnessing the capabilities of various hydrophysical drivers.

## 2.1    Biogeochemical module

### 2.1.1    General description

BROM-biogeochemistry consists of 3 biogeochemical submodules: BROM_bio (ecological model), BROM_redox (redox processes) and BROM_carb (carbonate system).

In BROM, reactions are either defined as kinetic processes (e.g. OM degradation) or equilibrium processes (e.g. carbonate system equilibration) (Boudreau, 1996; Jourabchi et al., 2008; Luff et al., 2001). In general, the redox reactions are fast in comparison with the other reactions and with the time step at which the model is typically integrated. Therefore, equilibrium concentrations of the chemical element species involved in such reactions can be calculated using mass action laws and equilibrium constants for seawater (Millero, 1995). This eliminates the need to include a separate state variable e.g. for pH. Instead, the total scale pH is calculated as a diagnostic variable at every time step as a function of DIC and Alk (which are state variables). In turn, the total scale pH is used in calculations of the chemical equilibrium constants required to describe related processes (i.e. carbonate precipitation/dissolution, carbonate system parameters etc.).

The model has 33 state variables, described in Table 1. This includes frequently measured components such as hydrogen sulfide ($H_2S$) and phosphate ($PO_4$), as well as rarely measured variables such as elemental sulfur ($S^0$), thiosulfate ($S_2O_3$), trivalent manganese species Mn(III), and bacteria. Variables of the latter category were included because their contribution to biogeochemical transformations is believed to be substantial. For instance, bacteria play an important role in many modelled processes and can consume or release nutrients in organic and inorganic forms (Canfield et al., 2005; Kappler et al., 2005). We acknowledge that for many of these additional variables, site-specific estimates of associated model parameters and initial/boundary conditions may be difficult or impossible to obtain, and may in practice require some crude assumptions and approximations (e.g. universal default parameter values, no-flux boundary conditions, initial conditions from a steady annual cycle). Nevertheless, we believe that for many applications this will be a price worth paying for the additional process resolution/realism provided by BROM for important biogeochemical processes in the BBL and sediments.  The equations and parameters employed in BROM are given in Tables 2 and 3, and a flow chart is shown in Figure 1.

## 2.1.2 Ecosystem and redox models

The BROM modules for ecosystem and redox processes are equivalent to those featured in ROLM. The overall goal of the ecosystem representation is to parameterize the key features of OM production and decomposition, which is based on Redfield and Richards stoichiometry (Richards, 1965). We divide all the living OM (biota) into Phy (photosynthetic biota), Het (non-microbial heterotrophic biota), and 4 groups of "bacteria" which may be considered to include microbial fungi. These latter are: Baae (aerobic chemoautotrophic bacteria), Baan (anaerobic chemoautotrophic bacteria), Bhae (aerobic heterotrophic bacteria), and Bhan (anaerobic heterotrophic bacteria). OM is produced photosynthetically by Phy and chemosynthetically by bacteria, specifically by Baae in oxic conditions and by Baan in anoxic conditions. Growth of heterotrophic bacteria is tied to mineralization of OM, favouring Bhae in oxic conditions and Bhan in anoxic conditions. Secondary production is represented by Het which consumes phytoplankton as well as all types of bacteria and dead particulate organic matter (detritus, which is also explicitly modelled). The effect of suboxia and anoxia is parameterized by letting the mortality of aerobic organisms depend on the oxygen availability.

As was mentioned before, BROM is based on ROLM which was designed to simulate redox processes that affect inorganic species of nitrogen, sulfur, manganese, iron, and phosphorus. Their detailed description is given in (Yakushev et al., 2007, 2013a) but the process parametrization, chemical reactions, rate and stoichiometric constants values are summarized in Tables 2-4. Table 2 also describes the redox-dependent switches, nutrient limitation and heterotrophic transfer functions. The redox-dependent switches are mostly based on hyperbolic tangent functions which improve system stability compared with discrete switches. The nutrient limitation and heterotrophic transfer functions are based on squared Monod laws for Nutrient/Biomass ratio, which also stabilizes the system compared with Michaelis-Menten and Ivlev formulations. Here we describe the parameterization of carbon that was not considered in ROLM and was not described in (Yakushev, 2013).

## 2.1.3 Total alkalinity

Total alkalinity, $A_T$, is a model state variable. Following the formal definition of $A_T$ (Dickson, 1992; Wolf-Gladrow et al., 2007; Zeebe and Wolf-Gladrow, 2001) the following alkalinity components were considered:

$$A_T = A_{TCO2} + A_B + A_{TPO4} + A_{Si} + A_{NH3} + A_{H2S} + [OH^-] - A_{SO4} - A_{HF} - A_{HNO3} - [H^+]$$

where the carbonate alkalinity $A_{TCO2} = [HCO_3^-] + 2[CO_3^{2-}]$, phosphoric alkalinity $A_{TPO4} = [HPO_4^{2-}] + 2[PO_4^{3-}] - [H_3PO_4]$, silicic alkalinity $A_{Si} = [H_3SiO_4^-]$, ammonia alkalinity $A_{NH3} = [NH_3]$, and hydrogen sulfide alkalinity $A_{H2S} = [HS^-]$ were calculated from the corresponding model state variables (Table 1) according to (Luff et al., 2001; Volkov, 1984). The boric alkalinity $A_B = [B(OH)_4^-]$ was estimated from salinity. $[OH^-]$ and $[H^+]$ were calculated using the ion product of water (Millero, 1995). The hydrogen sulfate alkalinity $A_{SO4} = [HSO_4^-]$, hydrofluoric alkalinity $A_{HF} = [HF]$, and nitrous acid alkalinity $A_{HNO3} = [HNO_2]$ were ignored due to their insignificant impact on $A_T$ variations in most natural marine and freshwater systems.

Biogeochemical processes can lead to either increase or decrease of alkalinity, and alkalinity can be used as an

indicator of specific biogeochemical processes (Soetaert et al., 2007). Organic matter production can affect

alkalinity via the 'nutrient-$H^+$ compensating principle' formulated by Wolf-Gladrow et al. (2007): during uptake

or release of charged nutrient species, electroneutrality is maintained by consumption or production of a proton

(i.e. during uptake of nitrate for photosynthesis or denitrification, or production of nitrate by nitrification).

BROM also considers the effect on alkalinity of the following redox reactions occurring in suboxic and anoxic

conditions via production or consumption of [$OH^-$] and [$H^+$] and changes in other "standard" alkalinity

components $A_{TCO2}$ and $A_{H2S}$ (see bold font):

$4Mn^{2+} + O_2 + \mathbf{4H^+} \rightarrow 4Mn^{3+} + 2H_2O$

$2Mn^{3+} + 3H_2O + 0.5O_2 \rightarrow 2MnO_2 + \mathbf{6H^+}$

$2MnO_2 + \mathbf{7H^+} + \mathbf{HS^-} \rightarrow 2Mn^{3+} + 4H_2O + S^0$

$2Mn^{3+} + \mathbf{HS^-} \rightarrow 2Mn^{2+} + S^0 + \mathbf{H^+}$

$Mn^{2+} + \mathbf{HS^-} \leftrightarrow MnS + \mathbf{H^+}$

$Mn^{2+} + \mathbf{CO_3^{2-}} \leftrightarrow MnCO_3$

$2\,MnCO_3 + O_2 + 2H_2O \rightarrow 2MnO_2 + \mathbf{2HCO_3^-} + \mathbf{2H^+}$

$4Fe^{2+} + O_2 + 10H_2O \rightarrow Fe(OH)_3 + \mathbf{8H^+}$

$2Fe^{2+} + MnO_2 + 4H_2O \rightarrow Fe(OH)_3 + Mn^{2+} + \mathbf{2H^+}$

$2Fe(OH)_3 + \mathbf{HS^-} + \mathbf{5H^+} \rightarrow 2Fe^{2+} + S^0 + 6H_2O$

$Fe^{2+} + \mathbf{HS^-} \leftrightarrow FeS + \mathbf{H^+}$

$FeS + 2.25O_2 + 2.5H_2O \rightarrow Fe(OH)_3 + \mathbf{2H^+} + SO_4^{2-}$

$FeS_2 + 3.5O_2 + H_2O \rightarrow Fe^{2+} + 2SO_4^{2-} + \mathbf{2H^+}$

$Fe^{2+} + \mathbf{CO_3^{2-}} \leftrightarrow FeCO_3$

$NH_4^+ + 1.5O_2 \rightarrow NO_2^- + \mathbf{2H^+} + H_2O$

$0.75CH_2O + \mathbf{H^+} + NO_2^- \rightarrow 0.5N_2 + 1.25H_2O + 0.75CO_2$

$4S^0 + 3H_2O \rightarrow 2H_2S + S_2O_3^{2-} + \mathbf{2H^+}$

$2S^0 + O_2 + H_2O \rightarrow S_2O_3^{2-} + \mathbf{2H^+}$

$4S^0 + 3NO_3^- + 7H_2O \rightarrow 4SO_4^{2-} + 3NH_4^+ + \mathbf{2H^+}$

$S_2O_3^{2-} + 2O_2 + \mathbf{2OH^-} \rightarrow 2SO_4^{2-} + H_2O$

$5H_2S + 8NO_3^- + \mathbf{2OH^-} \rightarrow 5SO_4^{2-} + 4N_2 + 6H_2O$

$Ca^{2+} + \mathbf{CO_3^{2-}} \leftrightarrow CaCO_3$

Standard alkalinity components were also affected by other reactions considered in the model (see Table 2).

**2.1.4  Carbonate system**

Equilibration of the carbonate system was considered as a fast process occurring within a few seconds (Zeebe

and Wolf-Gladrow, 2001). Accordingly, the equilibrium solution was calculated at every time step using an

iterative procedure. The carbonate system was described using standard approaches (Lewis and Wallace, 1998;

Munhoven, 2013; Roy et al., 1993a; Wanninkhof, 2014; Wolf-Gladrow et al., 2007; Zeebe and Wolf-Gladrow, 2001). The set of constants of (Roy et al., 1993a) was used for carbonic acid. Constants for boric, hydrofluoric, and hydrogen sulfate alkalinity were calculated according to (Dickson, 1992), for silicic alkalinity according to (Millero, 1995), for ammonia alkalinity according to (Luff et al., 2001), and for hydrogen sulfide alkalinity according to (Luff et al., 2001) and (Volkov, 1984). The ion product of water was calculated according to (Millero, 1995). Total scale pH was calculated using the Newton-Raphson method with the modifications proposed in (Munhoven, 2013). Precipitation and dissolution of calcium carbonate were modelled following the approach of (Luff et al., 2001) (Table 2).

## 2.2 Physical environment

As mentioned above, BROM-biogeochemistry can be very simply coupled "online" to any hydrodynamic model with FABM support. Typically, however, such couplings only cover biogeochemistry within the interior of the water column; the hydrodynamic model code may require extensive adaptation to resolve the BBL and upper sediments. We therefore developed a simple 1D offline transport-reaction model, BROM-transport, whose model domain spans the water column, BBL, and upper layers of the sediments, with enhanced spatial resolution in the BBL and sediments. All options and parameter values for BROM-transport are specified in a run-time input file brom.yaml. A step-by-step guide to running BROM-transport is provided in Appendix A.

### 2.2.1 BROM-transport model formulation

The time space evolution of state variables in BROM-transport is described by a system of 1D transport-reaction equations in Cartesian coordinates. In the water column the dynamics are:

$$\frac{\partial \hat{C}_i}{\partial t} = \frac{\partial}{\partial z} D \frac{\partial \hat{C}_i}{\partial z} - \frac{\partial}{\partial z} v_i \hat{C}_i + \varepsilon_h (\hat{C}_{0i} - \hat{C}_i) + T_{birr(i)} + R_i \tag{1}$$

where $\hat{C}_i$ is the concentration in units [mmol/m$^3$ total volume] of the $i^{\text{th}}$ state variable, $D(z,t)$ is the vertical diffusivity, $v_i$ is the settling or sinking velocity, $\varepsilon_h(z,t)$ is a rate of horizontal mixing with an external concentration $\hat{C}_{0i}(z,t)$ (or alternatively, a restoring rate to a climatological concentration), $T_{birr(i)}$ is a tendency due to bioirrigation (only non-zero for dissolved substances in the bottom layer of the water column, see below), and $R_i$ is the combined sources-minus-sinks (in this study provided by BROM-biogeochemistry, but in principle any biogeochemical model in FABM could be used). Values for $D$, $\varepsilon_h$, $\hat{C}_{0i}$, and other forcings used by $R_i$ are configured at run time through input files (see section 2.2.7). Sinking velocities $v_i$ are non-zero only for particulate (non-dissolved) variables and are determined at each time step by the biogeochemical module (through FABM). BROM-biogeochemistry assumes constant sinking velocities for phytoplankton, zooplankton, bacteria, detritus, and inorganic particles (Table 3.5).

In the sediments, dissolved substances or solutes obey the dynamics:

$$\varphi \frac{\partial C_i}{\partial t} = \frac{\partial}{\partial z} \varphi D_C \frac{\partial C_i}{\partial z} - \frac{\partial}{\partial z} \varphi u C_i + T_{birrC(i)} + R_i \tag{2}$$

where $\varphi$ is the porosity, assumed constant in time, $D_C$ is the total solute diffusivity, $u$ is the solute burial velocity, and $C_i$ is the porewater concentration in units [mmol/m$^3$ porewater]. Particulate substances become part of the solid matrix in the sediments. These obey:

$(1-\varphi)\frac{\partial B_i}{\partial t} = \frac{\partial}{\partial z}(1-\varphi)D_B\frac{\partial B_i}{\partial z} - \frac{\partial}{\partial z}(1-\varphi)wB_i + R_i$          (3)

where $D_B$ is the particulate (bioturbation) diffusivity, $w$ is the particulate burial velocity, and $B_i$ is the particulate

concentration in units [mmol/m$^3$ total solids].

The porosity $\varphi(z)$ in (2) and (3) is prescribed as an exponential decay, following (Soetaert et al., 1996):

$\varphi = \varphi_\infty + (\varphi_0 - \varphi_\infty)e^{-\frac{(z-z_{SWI})}{\delta}}$               (4)

where $\varphi_\infty$ is the deep (compacted) porosity, $\varphi_0$ is the sediment surface porosity, $z_{SWI}$ is the depth of the SWI, and

$\delta$ is a decay scale defining the rate of compaction.

Diffusion within the sediments is assumed to be strictly "intraphase" (Boudreau, 1997), hence the Fickian

gradients in (2) and (3) are formed using the concentration per unit volume porewater for solutes and per unit

volume total solids for particulates. The total solute diffusivity $D_C = D_m + D_B$, where $D_m$ is the apparent

molecular/ionic diffusivity and $D_B$ is the bioturbation diffusivity due to animal movement and

ingestion/excretion. The apparent molecular diffusivity $D_m(z) = \theta^{-2}D_0\frac{\mu_0}{\mu_{sw}}$ is derived from the infinite-dilution

molecular diffusivity $D_0$ (an input parameter) assuming a constant relative dynamic viscosity $\frac{\mu_0}{\mu_{sw}}$ (default value

0.94, cf. (Boudreau, 1997), Table 4.10) and a tortuosity parameterized as: $\theta^2 = 1 - 2\ln\varphi$ from (Boudreau,

1997)Eqn. 4.120. The bioturbation diffusivity $D_B(z,t)$ is modelled as a Michaelis-Menten function of the

dissolved oxygen concentration in the bottom layer of the water column:

$D_B(z,t) = D_{Bmax}(z)\frac{O_{2s}}{O_{2s}+K_{O2s}}$             (5)

where $D_{Bmax}(z)$ is a constant over a fixed mixed layer depth in the surface sediments then decays to zero with

increasing depth, and $K_{O2s}$ is a half-saturation constant. The rationale for (5) is that the animals (worms etc.)

that cause bioturbation require a source of oxygen at the sediment surface for respiration.

Diffusion between the sediments and water column, i.e. across the SWI, raises a subtle issue in regard to

particulates. Here any diffusive flux cannot be strictly intraphase, because particulates are modelled as

[mmol/m$^3$ total solids] in the sediments but as [mmol/m$^3$ total volume] in the water column. In BROM-

transport, the bottom layer of the water column is considered a "fluff layer"; particles enter through the upper

interface at their sinking velocity and leave through the layer interface (SWI) at the particulate burial velocity. It

follows that a portion of the particulate matter in the fluff layer must be considered as settled fluff, but that

portion is not predicted by the model. BROM-transport therefore offers two options. In the first approach, the

bioturbation diffusivity is set to zero on the SWI, so that only solutes can diffuse across the SWI by molecular

diffusion. Since the present version of BROM-transport does not parameterize resuspension through the SWI

due to fluid turbulence, the SWI thus becomes a one-way street for particulate matter, whose components can

only reenter the water column after dissolution. In the second approach, the bioturbation diffusivity is given by

(5) on the SWI, but the bioturbation flux is interphase, mixing concentrations in units [mmol/m$^3$ total volume]

for both solutes and particulates. This option is appropriate if bioturbation can be assumed to exchange fluff and

sediment, or if it contributes significantly to particulate resuspension.

The burial velocities $u$ and $w$ in (2) and (3) can be inferred from the porosity profile under the assumptions of

steady state compaction ($\varphi$ constant in time) and no externally-impressed porewater flow (Berner, 1971, 1980;

Boudreau, 1997; Meysman et al., 2005). Here, BROM-transport again offers two approaches. In the first

approach, the reactions of particles in the sediments are assumed to have negligible impact on the volume

fraction of total solids, and the deep particulate burial velocity $w_\infty$ in compacted sediments (where $\varphi = \varphi_\infty$) is

assumed to be a known constant $w_{b\infty}$ (an input parameter). Since compaction ceases at this (possibly infinite)

depth, the solute burial velocity must here equal the particulate burial velocity ($u_\infty = w_{b\infty}$). Steady state then

implies the following burial velocities (Appendix B):

$\qquad w = \frac{(1-\varphi_\infty)}{(1-\varphi)} w_{b\infty} - \frac{1}{(1-\varphi)} D_B^{inter} \frac{\partial \varphi}{\partial z}$ $\qquad\qquad\qquad\qquad\qquad$ (6)

$\qquad u = \frac{\varphi_\infty}{\varphi} w_{b\infty} + \frac{1}{\varphi} D_B^{inter} \frac{\partial \varphi}{\partial z}$ $\qquad\qquad\qquad\qquad\qquad\qquad$ (7)

where $D_B^{inter}$ is the interphase bioturbation diffusivity, non-zero only at the SWI and only if bioturbation across

the SWI is enabled. In the second approach, the reactions of the modelled particulate substances in the

sediments modify the total solid volume fraction, and the modelled sinking fluxes from the water column modify

the flux of solid volume at the SWI. The velocities in (6, 7) then define background velocities ($w_b$, $u_b$) due to

non-modelled particulates. Again assuming steady state compaction leads to the following corrections to the

background burial velocities (see Appendix B):

$\qquad w' = \frac{1}{(1-\varphi)} \sum_i^{N_p} \frac{1}{\rho_i} \left[ v_{f(i)} \hat{C}_{sf(i)} + \int_{z_{SWI}}^z R_i(z')dz' \right]$ $\qquad\qquad\qquad$ (8)

$\qquad u' = \frac{1}{\varphi} (w'_\infty - (1-\varphi)w')$ $\qquad\qquad\qquad\qquad\qquad\qquad$ (9)

where $w' = w - w_b$, $u' = u - u_b$, $N_p$ is the number of particulate variables, $\rho_i$ is the density of the $i^{th}$ particle

type, $v_{f(i)}$ is the sinking velocity in the fluff layer, $\hat{C}_{sf(i)}$ is the suspended particulate concentration in the fluff

layer, $R_i$ is the particulate reaction term, and $w'_\infty$ is the correction to the deep particulate burial velocity, in

practice approximated by the deepest value of $w'$. Since the suspended portion $\hat{C}_{sf(i)}$ is not explicitly modelled,

it is approximated as the minimum of the particulate concentrations in the fluff layer and the layer immediately

above. In our applications we have found that (8) and (9) can improve the realism of sediment organic matter

distributions, mainly by increasing the burial rate following pelagic production and export events such as the

spring bloom.

Finally, the process of bioirrigation, whereby worms flush out their burrows with water from the sediment

surface, is modelled as a non-local solute exchange following (Meile et al., 2001; Rutgers Van Der Loeff and

Boudreau, 1997; Schlüter et al., 2000):

$\qquad T_{birrC(i)} = \alpha \varphi \frac{O_{2s}}{O_{2s}+K_{O2s}} \left( \hat{C}_{f(i)} - C_i \right)$ $\quad$ (for solutes) $\qquad\qquad\qquad$ (10)

where $\alpha(z)$ is the bioirrigation rate in oxic conditions, $\hat{C}_{f(i)}$ is the flushing concentration of solute in the fluff

layer, and the Michaelis-Menten function again accounts for the suppression of worm activity in anoxic

conditions. The oxic bioirrigation rate $\alpha(z)$ is parameterized as an exponential decay from the sediment surface

as in Schluter et al. (2000). The total mass transfer to/from the sediment column must be balanced by a flux into/out of the fluff layer (see equation (1)):

$$T_{birr(i)} = \frac{1}{h_f} \frac{O_{2s}}{O_{2s} + K_{O2s}} \int_{z_{SWI}}^{z_{max}} \alpha\varphi\left(C_i - \hat{C}_{f(i)}\right) dz' \qquad \text{(for solutes)} \qquad (11)$$

where $h_f$ is the thickness of the fluff layer and $z_{max}$ is the depth of the bottom of the modelled sediment column. $T_{birrC(i)}, T_{birr(i)} = 0$ for all particulate variables.

### 2.2.2   BROM-transport numerical integration

Equations (1-3) are integrated numerically over a single combined grid (water column plus sediments) and using the same model time step in both water column and sediments. All concentrations are stored internally and input/output in units [mmol/m³ total volume]. Time stepping follows an operator splitting approach (Butenschön et al., 2012): concentrations are successively updated by contributions over one time step of diffusion, bioirrigation, reaction, and sedimentation, in that order. If any state variable has any 'not-a-number' values at the end of the time step then the program is terminated. Diffusive updates are calculated either by a simple forward-time central-space (FTCS) algorithm or by a semi-implicit, central-space algorithm adapted from a routine in the General Ocean Turbulence Model, GOTM (Umlauf et al., 2005). Bioirrigation and reaction updates are calculated as forward Euler time steps, using the FABM to compute $R_i$, and sedimentation updates are calculated using a simple first-order upwind differencing scheme. After each update, Dirichlet boundary conditions (see below) are reimposed and all concentrations are low-bounded by a minimum value (default = $10^{-11}$ μM) to avoid negative values. Maximum diffusive and advective Courant numbers can optionally be output after every time step or when/if a 'not-a-number' value is detected. Before starting the integration, the program calculates Courant numbers due to eddy/molecular diffusion and returns a warning message if the maximum value on high on any given day and the FTCS option is selected.

BROM-transport also provides the ability to divide the diffusion and sedimentation updates into smaller time steps related to the sources-minus-sinks time step by fixed factors, since the physical transport processes are often numerically limiting (Butenschön et al., 2012). The default time step is 0.0025 days or 216 seconds, which is much longer than the characteristic equilibration timescale of the $CO_2$ kinetics (Zeebe and Wolf-Gladrow, 2001).

### 2.2.3   BROM-transport vertical grid

The vertical grid in BROM-transport is divided into the pelagic water column, the BBL, and the sediments. The pelagic water column grid is either set as uniform with height/spacing set by the brom.yaml file (see Appendix C), or it is read from the NetCDF forcing input file (see below), with an option to decrease resolution by subsampling. In principle, the NetCDF input from the hydrodynamic model may already include a fully-resolved BBL, but in practice we find this is rarely the case. BROM-transport therefore allows the user to "insert" a high-resolution BBL into the bottom of the input water column. This BBL has non-uniform grid spacing with layer

thickness decreasing geometrically towards the SWI, reaching 2(cm) thickness for the fluff layer, based on parameters from the brom.yaml file. For the upper sediments, the layer thickness is increased geometrically moving down from the SWI, from 0.5(mm) thickness in the surface layer to 1(cm) thickness deeper in the sediments, again based on brom.yaml parameters. The result is a full grid with non-uniform spacing and maximum resolution near the SWI. As in many ocean models (e.g. ROMS, GOTM) the vertical grid in BROM-transport is staggered: temperature, salinity, and biogeochemical concentrations are defined at layer midpoints, while diffusivities, sinking/burial velocities, and resulting transport fluxes are all defined on layer interfaces.

### 2.2.4    BROM-transport initial conditions

Initial conditions for all concentrations in equations (1-3) can be provided by either using the initialization values defined in the fabm.yaml file (see Appendix D) as uniform initial conditions for each variable, or by providing the initial conditions for all variables at every depth in a text file with a specific format. Typically these initial condition text files are generated by running the model to a steady state annual cycle and saving the final values as the desired start date. Alternatively they could be generated by interpolating /smoothing data, in which case the user should note that the input concentrations must be in units [mmol/m$^3$ total volume].

### 2.2.5    BROM-transport boundary conditions

BROM-transport presently allows the user to choose between four different types of boundary condition for each variable and for upper and lower boundaries: 1) no-gradient at the bottom boundary (no diffusive flux) or no-flux at the surface boundary, except where parameterized by the FABM biogeochemical model (i.e. for $O_2$ and DIC in the case of BROM-biogeochemistry); 2) a fixed constant value; 3) a fixed sinusoidal variation in time defined by amplitude, mean value, and phase parameters; or 4) an arbitrary fixed variation in time read from the input NetCDF file. All boundary condition options and parameters are set in the brom.yaml file (see Appendix C). Note that options 2-4 are Dirichlet boundary conditions which define implicit fluxes of matter into and out of the model domain, and that all boundary concentrations should be in units [mmol/m$^3$ total volume (water+solids)]. The default option 1 is generally the preferred choice, but the Dirichlet options can also be useful to allow a simple representation of e.g. fluxes of nutrients into and out of the surface layer due to lateral riverine input. A possible alternative is to use the forcings parameters for horizontal mixing (see equation (1)) to specify horizontal exchanges or restoring terms to observed climatology (see section 2.2.7).

Under option 1, and using BROM-biogeochemistry, a surface $O_2$ flux representing exchange with the atmosphere is parameterized as:

$$Q_{O_2} = K_{660} * \left(Sc/660\right)^2 * (Oxsat - O_2) \qquad (12)$$

where Oxsat is the oxygen saturation as a function of temperature and salinity, according to UNESCO (1986), Sc is the Schmidt number, and $k_{660}$ is the reference gas-exchange transfer velocity, parameterized as $k_{660} = 0.365u^2 + 0.46u$ where u is the wind speed 10 m above the sea surface [m s$^{-1}$]. Air-sea exchange of

CO$_2$ in BROM-biogeochemistry is parameterized using the differences of the particle pressures in water (pCO$_2$$^{water}$) and air pCO$_2$$^{air}$). The formulation and coefficient were those used in ERSEM (Butenschön et al., 2016)

$$Q_{O_2} = F_{wind} * \left( pCO_2{}^{air} - pCO_2{}^{water} \right) \tag{13}$$

where $F_{wind}$ =(0.222u$^2$ + 0.333u)(Sc/660)$^{-0.5}$ is a wind parameter, u is the wind speed, and Sc is a Schmidt number.

### 2.2.6    BROM-transport irradiance model

BROM-transport includes  two simple Beer-Lambert attenuation models to calculate in situ 24-hour average photosynthetically active radiation (PAR) as needed by BROM-biogeochemistry and many other biogeochemical models. The first is derived from the current ERSEM default model (Blackford et al., 2004; Butenschön et al., 2016) and models the total attenuation as:

$$k_t = k_0 + k_{Phy}Phy + k_{PON}PON + k_sS \tag{14}$$

where $k_0$ is the background attenuation of seawater, $k_{Phy}$ , $k_{PON}$ are the specific attenuations due to phytoplankton and detritus respectively, and $k_s$ is the specific attenuation due "other" optically active substances with concentration $S$ (currently a constant input parameter).  The second model includes attenuation due to other optically active concentrations that are modelled by BROM-biogeochemistry:

$$k_t = k_0 + k_{Phy}Phy + k_{PON}PON + k_{Het}Het + k_{DON}DON + k_{PB}B + k_{PIV}PIV + k_sS \tag{15}$$

where $B$ is the total bacterial concentration ( $= Baae + Baan + Bhae + Bhan$) and $PIV$ is the total volume fraction of modelled inorganic particles, calculated from the concentrations using input densities of each inorganic solid.  The final irradiance is scaled by a constant parameter representing either the photosynthetically active fraction of the in situ irradiance or the relationship between surface PAR in water and the forcing surface irradiance (Mobley and Boss, 2012).  The forcing surface irradiance $Eair(t)$ can be read from NetCDF input or otherwise calculated using a sinusoidal function (Yakushev et al., 2013b).  In addition, the surface attenuation due to ice cover can be accounted for as a simple linear function of a NetCDF input ice thickness variable $hice(t)$.

### 2.2.7    BROM-transport input forcings

BROM-transport requires forcing inputs at least for temperature, salinity, and vertical diffusivity at all depths in the pelagic water column and for each day of the simulation. These may be provided from an input subroutine that creates simple, hypothetical profiles, or from text/NetCDF files containing data from interpolations of measurements or hydrodynamic model output.  Forcing time series of surface irradiance and ice thickness may also be read as NetCDF input.  BROM-transport then uses these inputs in combination with parameters set in the

1 run-time input file brom.yaml (see Appendix C) to solve the transport-reaction equations on a "full" vertical grid

including pelagic water column, BBL, and sediment subgrids.

In order to run, BROM-transport must extend the input pelagic (temperature, salinity, diffusivity) forcings over

the full grid. Temperature and salinity in the BBL and sediments are set as uniform and equal to the values at the

bottom of the input pelagic water column for each day. The vertical diffusivity needs a more careful treatment

as it is the main defining characteristic of the pelagic vs. BBL vs. sediment environments. Within the water

column, the total vertical diffusivity $D = D_m + D_e$ for solutes and $D = D_e$ for particulates, where $D_m$ is a constant

molecular diffusivity at infinite dilution, and $D_e$ is the eddy diffusivity read from the input file for the pelagic

water column. For the BBL, $D_e$ can be defined as "dynamic", in which case it is linearly interpolated for each

10 day between the deepest input forcing value above the SWI and zero at a depth $h_{DBL}$ above the SWI, where $h_{DBL}$

is the diffusive boundary layer (DBL) thickness (default value 0.5 mm). This option is likely appropriate for

shallow water applications where $D_e$ may be strongly time-dependent within the user-defined BBL (default

thickness 0.5 m). Alternatively, a static, fixed profile $D_{eBBL}(z)$ may be more appropriate for deep water BBLs,

where time dependence may be weak and deepest values from hydrodynamic models may be relatively far above

the SWI. In this case, BROM-transport offers two options for $D_{eBBL}(z)$: 1) a constant value, dropping to zero in

the DBL, or 2) a linear variation between a fixed value at the top of the BBL and zero at the top of the DBL.

Option 1) defines a simplest-possible assumption, while option 2) corresponds to the assumption of a log layer

for the current speed e.g. (Boudreau and Jorgensen, 2001). Eddy diffusivity is strictly zero in the DBL, on the

SWI, and within the sediments. Diffusivity in the sediments is due to molecular diffusion and bioturbation and is

parameterized as described in section 2.2.1.

Optional forcings for BROM-transport include 24-hour average surface irradiance $Eair(t)$, which is often

supplied by hydrodynamic models (e.g. ROMS), a surface ice thickness forcing $hice(t)$, and depth-time arrays of

horizontal mixing rates $\varepsilon_h(z,t)$ and horizontal mixing concentrations $\hat{C}_{0i}(z,t)$ (see equation (1)). Horizontal

mixing rates within the inserted BBL and sediments are set to zero. Note that these horizontal mixing forcings

can also be used to define relaxation or restoring fluxes to climatological values within the pelagic water column,

which may in some cases provide a valid means of accounting for horizontal flux divergence effects that are

missing in the 1D model.

**3    BROM demonstration run**

**3.1    Model setup**

A North Sea hydrodynamic scenario was used to demonstrate the ability of BROM to reproduce the

biogeochemical mechanisms of oxic/anoxic transformations. Complete lists of the model options and parameter

values used are given in Appendix C (brom.yaml input file for BROM-transport) and Appendix D (fabm.yaml

input file for BROM-biogeochemistry).

The BROM-transport water column extended from 0 to 110 m, with a pelagic spatial resolution of 1 m inherited

from the GOTM hydrodynamic model used to provide forcings. A high-resolution BBL was inserted from 109.5

to 110 m, with layer thickness decreasing from approximately 25 cm to 3 cm in the fluff layer. Sediment grid

points were added to cover the upper 10 cm of sediments with layer thickness increasing from 0.5 mm in the

surface layer to 1 cm at depth. The model time step for BROM-transport was set to 0.0025 days (216 seconds).

Upper boundary conditions included sinusoidal, time-varying Dirichlet boundary conditions for nitrate,

phosphate and silicate, implying net influxes and outfluxes of surface nutrients, as well as the default

parameterized air-sea fluxes of $O_2$ and DIC (see Appendix C). Lower boundary conditions assumed (by default)

zero diffusive flux for all reduced components (i.e. hydrogen sulfide, solid phase concentrations of metal sulfides

and carbonates, silicon and OM). The simulation therefore focuses on the consequences of the supply of fresh

OM as a main reducer in both water column and sediments.

The pelagic water column was forced by output from a GOTM hydrodynamical simulation for temperature,

salinity, and vertical diffusivity (taken from the salinity diffusivity) and surface irradiance calculated using the

sinusoidal option. We aimed for a solution representative for "present day" and therefore treated the GOTM

forcing as representative for a "normal year". BROM-transport was spun up from vertically-homogeneous initial

conditions for 100 model years with repeated-year forcings and boundary conditions. After this time, a quasi-

stationary solution with seasonally forced oscillations of the biogeochemical variables had been reached.

The results of these calculations were written to an output file in NetCDF format, including the daily vertical

distributions of model state variables, diagnostic rates of biogeochemical transformations, and fluxes associated

with diffusion and sedimentation. This output can be visualized by any NetCDF-compatible software.

**3.2    Results**

The model simulated the periodic replacement of oxic with anoxic conditions in the BBL following seasonal

mixing and OM production. The simulation demonstrates the characteristic features of biogeochemical profiles

in the water column, BBL and upper sediments, as well as their variability under changing redox conditions

(Figs. 2-4).

During intensive mixing conditions in winter, the water column is well oxygenated and the oxic/anoxic interface

is located at several centimeters depth in the sediments (Figs. 2, 3). In summer, just after the spring bloom, an

enrichment of the sediment surface with fresh OM and a restricted oxygen supply leads to the consumption of $O_2$

by OM mineralization and close to suboxic conditions (Fig. 2). The second bloom in the autumn leads to a

further decrease of oxygen concentrations to complete depletion. There is a concomitant increase in reduced

forms of N, Mn, Fe and finally of hydrogen sulfide in the bottom water (Figs. 2, 4). The redox interface thus

moves from the sediment to the BBL.

Figure 5 shows the rate of OM mineralization with a variety of electron acceptors. Oxygen is consumed during

OM mineralization in summer and autumn and, after its complete depletion, denitrification dominates, with both

nitrate reduction and nitrite reduction playing significant roles. The rate of mineralization of OM with Mn and Fe

oxides is small, but as these processes prevent mineralization with sulfate, they cause a lag of a few days

between the depletion of oxygen and the appearance of hydrogen sulfide in the water column (Figs. 2, 5). The

amount of labile degradable OM is relatively small and mineralization with sulfate completely removes the

remaining OM, thus preventing methanogenesis (Fig. 5).

The seasonal variability of the sediment-water fluxes clearly demonstrates the appearance in the bottom water of

reduced forms of N, Mn, Fe and phosphate (Fig 6).

Generally, the concentrations, vertical distributions and benthic-pelagic fluxes of the parameters considered in

the model are reasonable and are within observed ranges for the North Sea (Queirós et al., 2014) and some other

regions with temporary bottom anoxia (Almroth et al., 2009; McCarthy et al., 2008; Morse and Eldridge, 2007;

Pakhomova et al., 2007; Queirós et al., 2014; Yu et al., 2015).

## 4    Conclusion and future work

This paper presents a description of BROM, a fully-coupled pelagic-benthic model that provides a integrated

framework to study the biogeochemistry of a water column and upper sediments. BROM simulates changes in

redox conditions and their impact on the distributions of a wide range of biogeochemical variables. In particular,

BROM provides a detailed description of the fate and availability of dissolved oxygen and hydrogen sulfide, the

former essential for macroscopic marine life, the latter highly toxic to it. BROM can therefore provide valuable

information to ecological studies, particularly in the context of multistressor impacts. The model suggests that

the timing of hydrogen sulfide release into the pelagic is linked to the dynamics of several electron acceptors that

are themselves of limited interest for biogeochemical and ecological purposes, and that are therefore rarely

included in models. The ability of BROM to simulate and forecast $H_2S$ toxicity is in fact the direct result of its

inclusion of several of these rarely modelled chemical compounds (e.g., Mn(IV), Fe(III)).

This paper was not devoted to a detailed validation of BROM with in situ data; we plan to explore this in future

work.  A qualitative analysis of the model results (Chapter 3) suggests that the model can produce realistic

distributions and fluxes of key biogeochemical variables during periodic changes in redox conditions.  More

detailed evaluations of the model will be presented in the separate papers devoted to the studies in the selected

regions, namely, for the Lagoon Berre, Fjord Sælenvatnet , and for the Gulf of Finland (in preparation).

In summary, we present a new benthic-pelagic biogeochemical model (BROM) that combines a relatively simple

pelagic ecosystem model with a detailed biogeochemical model of the coupled cycles of (N, P, Si, C, O, S, Mn,

Fe) in the water column, benthic boundary layer, and sediments, with a focus on oxygen and redox state.  BROM

should be an interesting tool for the study of benthic nutrient recycling, redox biogeochemistry, eutrophication,

industrial pollution from mineral effluent and organic loading, deoxygenation, acidification, and the potential

associated release of contaminants such as hydrogen sulfide in marine and freshwater environments.

## Code availability

32    The model as presented consists of two components. The first is a set of biogeochemical modules (brom/redox,

33    brom/bio, brom/carb, brom/eqconst), available as part of the official FABM distribution (http://fabm.net);

34    BROM-specific files are located in subdirectory src/models/niva/brom). The second is a hydrophysical driver

(BROM-transport) that provides the 1D vertical context and resolves transport; this is available separately from

https://github.com/e-yakushev/brom-git.git. When combined, the 1D BROM model as presented is obtained.

Both FABM and BROM-transport are coded in object-oriented Fortran 2003, have a build system based on

CMake (https://cmake.org), and use YAML files (http://yaml.org) for run-time configuration. The code is

platform independent and only requires a Fortran 2003-capable compiler, e.g., gfortran 4.7 or higher, or the Intel

Fortran compiler version 12.1 or higher. BROM-transport includes facilities for producing results as NetCDF

files, which can be read by a variety of software on different platforms.

Also you can run BROM without any Fortran compiler using a Win32 executable file which can be downloaded

from https://github.com/e-yakushev/brom-git/releases/tag/v1.1

As BROM's biogeochemical modules are built on FABM, they can be used from a wide range of 1D and 3D

hydrodynamic models, including GOTM, GETM, ROMS, MOM, NEMO and FVCOM (a ROMS-FABM

coupler has been developed by P.W.; NEMO-FABM and FVCOM-FABM couplers have been developed by the

Plymouth Marine Laboratory; contact J.B. for information).

Results shown in this paper were produced with BROM-transport tag v1.1. and the BROM-biogeochemistry

code in FABM tag v0.95.3, available from the above repositories. The simulation was run using the

netCDF/.yaml input files found in the data/ folder of the BROM-transport repository. However, we envisage

BROM to be further developed in a backward compatible manner, and encourage users to adopt the latest

version of the code. Step-by-step instructions for running BROM are found in Appendix A.  Both FABM and

BROM-transport are distributed under the GNU General Public License (http://www.gnu.org/licenses/).

**Author contributions**: Development of the model code was made by EY, EP, JB, PW, SY, analyses of the model results and

discussions were conducted by RB, RC and SP, and all authors contributed to the writing of the manuscript.

**Acknowledgements**

We wish to thank J. Middelburg, O.P. Savchuk, and G. Munhoven for detailed and constructive reviews
that ultimately led to a major improvement in the manuscript. We acknowledge funding from the EC 7th
Framework Program (FP7/2007-2013) under grant agreement no 265847 ('Sub-seabed CO2 Storage: Impact on
Marine Ecosystems', ECO2) and 240837 ('Research into Impacts and Safety in CO2 Storage', RISCS), with
additional development funds from FME SUCCESS, CO2Base, EEA CO2MARINE, Norwegian Research
Council project no. 236658 ('New knowledge on sea deposits' NYKOS), the Research Council of Norway
through its Centers of Excellence funding scheme, project number 223268/F50 (CERAD), contract no. 208279
(NIVA Strategic Institute Initiative "Climate effects from Mountains to Fjords"), NIVA OASIS, and VISTA – a
basic research program and collaborative partnership between the Norwegian Academy of Science and Letters
and Statoil, project no.6164. The work of J.B. was funded by NERC National Capability in Marine Modelling.

**Appendix A: Running BROM step-by-step**

1. Installation. A Fortran-2003-capable compiler, e.g., gfortran 4.7 or higher, or the Intel Fortran compiler
version 12.1 or higher should be installed. In our demonstration we used the Intel Fortran Compiler version
15.0.4.221. Additionally, a NetCDF library compatible with the chosen Fortran compiler is required. CMake
software should be installed. After ensuring these prerequisites are in place, create a directory to hold the BROM

model code and associated input and output files. Detailed instructions for installation are provided at BROM

repository https://github.com/e-yakushev/brom-git.git

2. Preparation of input files. The model reads two .yaml files with the model parameters (fabm.yaml and

brom.yaml), as well as a NetCDF or text file with the hydrophysical forcing data. Optionally the biogeochemical

initial conditions can be read from an text file **start.dat**; this may be a file written by a previous simulation (the

final model state is written to a file named **finish.dat** at the end of every simulation).

i. brom.yaml (see Appendix C). This file specifies the values of transport model parameters as well as various

option switches and input/output file and variable names.  Text comments provide guidance and references for

setting parameter values. If using NetCDF input the user should pay careful attention to the NetCDF input

parameters and names, ensuring that this information is consistent with the input NetCDF file.  The selected year

parameter year must refer to a year that is covered by the input forcing data.

ii. fabm.yaml (see Appendix D). This file specifies the values of biogeochemical model parameters , default

initial values for state variables, and the coupling of FABM modules.  Text comments provide annotation and

references.

iii. nns_annual.nc (in the example). This file contains input forcing data that may be derived from observations

or hydrodynamical model output (GOTM in our demonstration). It can be replaced by a text (.dat) file if this is

the format of the hydrodynamical model output.

iv. start.dat. Text file with initial values for model state variables at every depth. This file may be created by

renaming the output of a previous simulation (finish.dat: the state at the 1$^{st}$ of January of the last modeled year).

3. Output files. These are NetCDF and headed text files generated automatically by the model during the

simulation. Output files can be readily imported into various software packages for visualization and further

analysis. Certain output files (Vertical_grid.dat and Hydrophysics.dat) are generated early in the simulation and

should be checked by the user to ensure that the model grid and hydrophysical forcings are set up as intended.

i. Vertical_grid.dat. Text file with model layer indices, midpoint depths, increments between midpoint depths,

and thicknesses.

ii. Hydrophysics.dat. Text file with daily profiles of hydrophysical variables (temperature, salinity, diffusivity,

porosity, tortuosity, burial velocities).

iii. finish.dat. Text file with the state variables for the 1$^{st}$ of January of the last modeled year. Can be used for

visualization or as initial conditions for further calculations.

iv. output_NNday.dat.  Optional text file with the state variables and diagnostic variables for day NN to make

plots of vertical distributions (e.g. Fig. 3)

v. BROM_out.nc. NetCDF file with daily profiles of state variables, rates of biogeochemical transformations,

vertical fluxes.

4. Visualization. For NetCDF output file can be used any software with NetCDF input. In the example we used

PyNcView.  To visualize vertical distributions from text files we used the Python script available at

https://github.com/lisapro/_brom-pics.git

**Appendix B: Derivation of burial velocities**

**The conservation equations for liquid and total solid volume fractions in the sediments can be written as:**

$$\frac{\partial \varphi}{\partial t} = \frac{\partial}{\partial z} D_B^{inter} \frac{\partial \varphi}{\partial z} - \frac{\partial}{\partial z} u\varphi - \sum_{i=1}^{N_p} \rho_i^{-1} R_i \qquad (B1)$$

$$\frac{\partial (1-\varphi)}{\partial t} = \frac{\partial}{\partial z} D_B^{inter} \frac{\partial (1-\varphi)}{\partial z} - \frac{\partial}{\partial z} w(1-\varphi) + \sum_{i=1}^{N_p} \rho_i^{-1} R_i \qquad (B2)$$

where $D_B^{inter}$ is the interphase bioturbation diffusivity (possibly non-zero only at the SWI), $\rho_i$ is the density of the $i^{th}$ particulate substance, and $R_i$ is the corresponding reaction term. Equations (B1) and (B2) assume that the densities of liquid and total solid are both constant, and they retain the net contributions of reactive terms although these are often considered negligible e.g. (Boudreau, 1997; Meysman et al., 2005). Summing (B1) and (B2) and integrating over depth gives a useful and quite general relationship:

$$\varphi u + (1-\varphi)w = U \qquad (B3)$$

where $U(t)$ is only a function of time. If we now assume no externally impressed porewater flow, it follows that at some (possibly infinite) depth where compaction ceases ($\frac{\partial \varphi}{\partial z}=0$, $\varphi = \varphi_\infty$), the solute burial velocity $u$ must here equal the particulate burial velocity $w$, hence $u_\infty = w_\infty$. Equation (B3) becomes:

$$\varphi u + (1-\varphi)w = w_\infty \qquad (B4)$$

Now assuming steady state compaction ($\frac{\partial \varphi}{\partial t}=0$), equation (B2) can be integrated from the SWI to a depth $z$ within the sediments:

$$(1-\varphi)w + D_B^{inter} \frac{\partial \varphi}{\partial z} = (1-\varphi_{SWI})w_{SWI} + D_{BSWI}^{inter} \frac{\partial \varphi}{\partial z}_{SWI} + \sum_i^{N_p} \frac{1}{\rho_i} \int_{z_{SWI}}^{z} R_i(z')dz' \qquad (B5)$$

To determine the first term on the RHS of (B5), we assume that the total solid volume flux across the SWI is equal to the total solid volume flux from the sinking of suspended particulate matter in the fluff layer:

$$(1-\varphi_{SWI})w_{SWI} + D_{BSWI}^{inter} \frac{\partial \varphi}{\partial z}_{SWI} = F_b + \sum_i^{N_p} \frac{1}{\rho_i} v_{f(i)} \hat{C}_{sf(i)} \qquad (B6)$$

where $F_b$ defines a constant background solid volume flux due to non-modelled particles, $v_{f(i)}$ is the sinking velocity in the fluff layer, and $\hat{C}_{sf(i)}$ is the suspended particulate concentration in the fluff layer. Substituting into (B5) we have:

$$(1-\varphi)w + D_B^{inter} \frac{\partial \varphi}{\partial z} = F_b + \sum_i^{N_p} \frac{1}{\rho_i} \left[ v_{f(i)} \hat{C}_{sf(i)} + \int_{z_{SWI}}^{z} R_i(z')dz' \right] \qquad (B7)$$

Since $D_B^{inter} \frac{\partial \varphi}{\partial z}$ is zero at depth, the constant surface flux term is given by $F_b = (1-\varphi_\infty)w_{b\infty}$, where both $\varphi_\infty$ and $w_{b\infty}$ are input parameters. Hence we have:

$$(1-\varphi)w + D_B^{inter} \frac{\partial \varphi}{\partial z} = (1-\varphi_\infty)w_{b\infty} + \sum_i^{N_p} \frac{1}{\rho_i} \left[ v_{f(i)} \hat{C}_{sf(i)} + \int_{z_{SWI}}^{z} R_i(z')dz' \right] \qquad (B7)$$

Equation (6) directly follows from (B7) by neglecting the modelled settling flux and reaction terms, then equation (7) follows by application of (B4). Equations (8) and (9) follow by considering the additional particulate burial velocity due to modelled fluxes and reactions (from the last term in B7) and applying (B4) to obtain the additional solute burial velocity.

**Appendix C: run-time input file for BROM-transport (brom.yaml)**

```
# IMPORTANT !!!! _ <TAB> is NOT allowed here, used <Space> only !!!!
# Each entry must have 6 spaces before the parameter name
instances:
  brom:
    initialization:
##---Paramters for grid-------(see io_ascii.f90/make_vert_grid for a grid diagram)-----------------
      water_layer_thickness: 95.  # Thickness of the water column [m] (may overriden by netCDF input, see below)
      k_wat_bbl: 18            # Number of levels above the water/BBL boundary  (may be overriden by netCDF input, see
below)
      bbl_thickness: 0.5         # Thickness of the high-resolution layer overlying the sediments (model "benthic boundary
layer") [m] (default = 0.5 m)
                        #  This should be thinner than the full viscous+logarithmic layer, but thicker than the viscous layer
                        #  Typical thicknesses for full viscous+logarithmic layer are 1 m and 10 m for deep sea and shelf
respectively (Wimbush 2012)
      hz_bbl_min: 0.02         # Minimum allowed layer thickness in the BBL near the SWI [m] (default = 0.02 m)
      hz_sed_min: 0.0005       # Minimum layer thickness in the sediments near the SWI [m] (default = 0.0005 m)
      hz_sed_max: 0.01         # Maximum layer thickness deeper in the sediments [m] (default = 0.01 m)
      k_min: 1               # Minimum k number defining the layer that is in contact with the atmosphere (default = 1)
      k_points_below_water: 17   # Number of levels below the water/BBL boundary (default = 20)
      i_min: 1                 # Minimum i number (default = 1)
      i_water: 1               # Number of i for water column (default = 1)
      i_max: 1                 # Maximum i number  (default = 1)
#Note: (i_min,i_water,i_max) should be (1,1,1) for 1D applications
#
#
##---Boundary conditions-------------------------------------------------------------------------
#
#Here we set the type of boundary condition using bctype_top_<variable name> and bctype_bottom_<variable name>
#    0 to use surface fluxes from FABM where parameterized, otherwise no flux (default, does not need to be explicitly set)
#    1 for constant Dirichlet, specified by bc_top_<variable name> or bc_bottom_<variable name>
#    E.g. bctype_bottom_niva_brom_bio_O2: 1
#       bc_bottom_niva_brom_bio_O2: 0.
#    2 for sinusoidal Dirichlet, specified by bcpar_top_<variable name> or bcpar_bottom_<variable name>
#       The model is: phi(t) = a1 + a2*sin(omega*(day-a3)) where omega = 2*pi/365
#                      => max(phi(t)) = a1+a2, min(phi(t)) = a1-a2, mean(phi(t)) = a1, peak at 91.25+a3 days
#       Model parameters are specified by a1top_<variable name> etc.
#    E.g. bctype_top_niva_brom_bio_NO3: 2
#       a1top_niva_brom_bio_NO3: 3.0
#       a2top_niva_brom_bio_NO3: 3.0
#       a3top_niva_brom_bio_NO3: 60.
#    3 for arbitrary Dirichlet, read from netCDF file (see I/O options to specify netCDF variable names)
#
#     bctype_bottom_niva_brom_bio_O2: 1
#     bc_bottom_niva_brom_bio_O2: 0.
#
      bctype_top_niva_brom_redox_SO4: 1
      bc_top_niva_brom_redox_SO4: 25000.
      bctype_bottom_niva_brom_redox_SO4: 1
      bc_bottom_niva_brom_redox_SO4: 25000.
#
      bctype_top_niva_brom_redox_Mn4: 1
      bc_top_niva_brom_redox_Mn4: 20.E-4
#
      bctype_top_niva_brom_redox_Fe3: 1
      bc_top_niva_brom_redox_Fe3: 5.E-4
#
      bctype_top_niva_brom_carb_Alk: 1
      bc_top_niva_brom_carb_Alk: 2200.
#     bctype_bottom_niva_brom_carb_Alk: 1
#     bc_bottom_niva_brom_carb_Alk: 3200.
#
```

```
#       bctype_bottom_niva_brom_carb_DIC: 1
#       bc_bottom_niva_brom_carb_DIC: 2850.
#
#       bctype_bottom_niva_brom_bio_NH4: 1
#       bc_bottom_niva_brom_bio_NH4: 10.
#
bctype_top_niva_brom_bio_NO3: 2
a1top_niva_brom_bio_NO3: 1. # 3
a2top_niva_brom_bio_NO3: 1.
a3top_niva_brom_bio_NO3: 320.
bctype_bottom_niva_brom_bio_NO3: 1
bc_bottom_niva_brom_bio_NO3: 0.
#
bctype_top_niva_brom_bio_PO4: 2
a1top_niva_brom_bio_PO4: 0.7 #0.8
a2top_niva_brom_bio_PO4: 0.7
a3top_niva_brom_bio_PO4: 320. #60.
#     bctype_bottom_niva_brom_bio_PO4: 1
#     bc_bottom_niva_brom_bio_PO4: 10.
#
bctype_top_niva_brom_redox_Si: 2
a1top_niva_brom_redox_Si: 1.5
a2top_niva_brom_redox_Si: 1.5
a3top_niva_brom_redox_Si: 320.
#     bctype_bottom_niva_brom_redox_Si: 1
#     bc_bottom_niva_brom_redox_Si: 100.
#
#
##---Horizontal mixing parameters----------------------------------------------------------------
#
#Here we specify horizontal mixing model using hmix_<variable name>
#   0 to assume no horizontal mixing (default, does not need to be explicitly set)
#   1 for "box model" mixing model: hmix = hmix_rate*(X_0 - X) with X_0 specified by netCDF input file and hmix_rate
specified here
#
hmix_niva_brom_bio_NO3: 0
hmix_niva_brom_bio_NH4: 0
hmix_niva_brom_bio_PO4: 0
hmix_niva_brom_redox_Si: 0
hmix_niva_brom_bio_O2: 0
#
#
##---Ice model parameters-------------------------------------------------------------------
use_hice: 0      # 1 to use ice thickness forcing "hice" from netCDF input
#
#
##---Constant forcings-----------------------------------------------------------------------
density: 1000.
wind_speed: 8.   # Wind speed 10 m above sea surface [m/s] (default = 8 m/s)
pco2_atm: 380.   # Atmospheric partial pressure of CO2 [ppm] (default = 380 ppm)
#
#
##---Surface irradiance model parameters-----------------------------------------------------
use_Eair: 0      # 1 to use 24-hr average surface downwelling shortwave irradiance in air from netCDF input
lat_light: 50    # Latitude of modelled site [degrees north], e.g. Hardangerfjord station H6 is at 60.228N; Sleipner=50N;
Saelen=60.33N
Io: 80.          # Theoretical maximum 24-hr average surface downwelling shortwave irradiance in air [W/m2] (default =
80 W/m2)
#  This should include that effect of average cloud cover (local)
light_model: 0   # Specify light model: 0 for simple model based on ersem/light.f90
#                      1 for extended model accounting for other particulates in BROM
#
#
```

```
##---Light absorption model parameters ---------------------------------------------------------
Eair_to_PAR0: 0.5  # Factor to convert input or calculated surface downward irradiance Eair to surface PAR in water
(default = 0.5, units dependent on Eair)
#  Factor of ~0.48 to convert shortwave (0.3-4 um) to PAR-band (0.4-0.7 um) in [W/m2]
#  Further factor of 0.8-0.95 to convert downward-in-air to net-in-water (Mobley and Boss, 2012, Figs. 2c,
4b, 8a)
#  Latter factor becomes 0.45-0.55 if modelling PAR in terms of photon flux (Mobley and Boss, 2012, Figs.
5b, 8b)
k0r:  0.04       # Background PAR attenuation [m-1] (default = 0.04 m-1, from ERSEM shortwave attenuation default)
kESS: 4e-05       # Specific PAR attenuation by silt [m^2/mg] (default = 4e-05 m^2/mg, from ERSEM shortwave
attenuation default)
ESS:  0.        # Assumed (constant) concentration of silt [mg/m^3] (default = 0. mg/m^3, from ERSEM shortwave
attenuation default)
kPhy: 0.00023      # Specific PAR attenuation by phytoplankton [m^2/mg N] (default = 0.0023 m^2/mg N, from ERSEM
shortwave attenuation default)
#  From ERSEM Blackford (P1-P4), default = 0.0004 m^2/mg C * 5.68 mg C/mg N (Redfield ratio 106/16
17  mol/mol)
#  Note misprint "e-3" instead of "e-4" in Blackford et al. (2004) Table 1
kPON: 0.         # Specific PAR attenuation due to PON [m^2/mg N] (default = 0. m^2/mg N)
# The following are only used if light_model = 1
kHet: 0.         # Specific PAR attenuation due to zooplankton [m^2/mg N] (default = 0. m^2/mg N)
kDON: 0.          # Specific PAR attenuation due to DON [m^2/mg N] (default = 0. m^2/mg N)
kB:  0.         # Specific PAR attenuation due to bacteria [m^2/mg N] (default = 0. m^2/mg N)
kPIV: 0.          # Specific PAR attenuation due to total particulate inorganic volume fraction (default = 0. m^-1)
#
#
##---Assumed densities for particles in the model (may be used in light/sedimentation models)------
#
# Densities are specified by rho_<full variable name> and in same units as the model concentration
# Any missing values will use the default density rho_def
rho_def:  3.0E7              # Default density of solid particles [mmol/m3]
rho_niva_brom_bio_Phy: 1.5E7        # Density of (living) phytoplankton [mmolN/m3] (default = 1.4E6 mmolN/m3 from
PON default)
rho_niva_brom_bio_PON: 1.5E7        # Density of (dead) particulate organic matter [mmolN/m3] (default = 1.4E6
mmolN/m3, from: 1.23 g WW/cm3 (Alldredge, Gotschalk, 1988), mg DW/mg WW=0.18 and mg DW /mg C=2 (Link et
al.,2006))
rho_niva_brom_bio_Het: 1.5E7        # Density of (living) non-bacterial heterotrophs [mmolN/m3] (default = 1.4E6
mmolN/m3 from PON default)
rho_niva_brom_redox_Baae: 1.5E7      # Density of (living) aerobic autotrophic bacteria [mmolN/m3] (default = 1.4E6
mmolN/m3 from PON default)
rho_niva_brom_redox_Bhae: 1.5E7      # Density of (living) aerobic heterotrophic bacteria [mmolN/m3] (default = 1.4E6
mmolN/m3 from PON default)
rho_niva_brom_redox_Baan: 1.5E7      # Density of (living) anaerobic autotrophic bacteria [mmolN/m3] (default = 1.4E6
mmolN/m3 from PON default)
rho_niva_brom_redox_Bhan: 1.5E7      # Density of (living) anaerobic heterotrophic bacteria [mmolN/m3] (default =
1.4E6 mmolN/m3 from PON default)
rho_niva_brom_redox_CaCO3: 2.80E7    # Density of calcium carbonate [mmolCa/m3] (default = 2.80E7 mmolCa/m3)
rho_niva_brom_redox_Fe3: 3.27E7      # Density of Fe3 [mmolFe/m3] (default = 3.27E7 mmolFe/m3)
rho_niva_brom_redox_FeCO3: 2.93E7    # Density of FeCO3 [mmolFe/m3] (default = 2.93E7 mmolFe/m3)
rho_niva_brom_redox_FeS: 5.90E7      # Density of FeS [mmolFe/m3] (default = 5.90E7 mmolFe/m3)
rho_niva_brom_redox_FeS2: 4.17E7     # Density of FeS2 [mmolFe/m3] (default = 4.17E7 mmolFe/m3)
rho_niva_brom_redox_Mn4: 5.78E7      # Density of Mn4 [mmolMn/m3] (default = 5.78E7 mmolMn/m3)
rho_niva_brom_redox_MnCO3: 3.20E7    # Density of MnCO3 [mmolMn/m3] (default = 3.20E7 mmolMn/m3)
rho_niva_brom_redox_MnS: 4.60E7      # Density of MnS [mmolMn/m3] (default = 4.60E7 mmolMn/m3)
rho_niva_brom_redox_S0: 6.56E7      # Density of S0 [mmolS/m3] (default = 6.56E7 mmolS/m3)
rho_niva_brom_redox_Sipart: 4.40E7   # Density of particulate silicate [mmolSi/m3] (default = 4.40E7 mmolSi/m3)
#
#
##---Time stepping parameters---------------------------------------------------------------------
dt:       0.0025    # Time step in [days] (default = 0.0025 days)
freq_turb:  1         # Physical mixing time step = dt/freq_turb (default = 1)
freq_sed:  1         # Sinking / bhc frequency (default = 1)
```

```
1     year:      1998      # Selected year (for reading netCDF inputs) WARNING: This must be a year present in the netCDF
file, and nc_year0 must be correctly specified below
days_in_yr: 365       # Number of days in repeated period (typically 365 or 366, default = 365)
last_day:   3650      # Last day in simulation (~ days_in_yr * no. repeated years, default = 365)
cc0:       1.0E-11    # Resilient (minimum) concentration for all variables [mmol/m3] (default = 1.0E-11 mmol/m3)
surf_flux_with_diff: 0  # 1 to include surface fluxes in diffusion update, 0 to include in biogeochemical update (default =
0)
#
#
##---Vertical diffusivity parameters----------------------------------------------------------------
diff_method:   1      # Numerical method to treat vertical diffusion (default = 1):
# 0 for FTCS approach (Forward-Time Central-Space scheme)
# 1 for GOTM approach (semi-implicit in time) using diff_center from GOTM lake (converting
input/output units)
# 2 for GOTM approach (semi-implicit in time) using modified version of original GOTM diff_center
(no units conversion required, should give very similar results to diff_method = 1)
# Note: If diff_method>0 and bioturb_across_SWI = 1 below, only one modified GOTM subroutine can
be used (diff_center2)
cnpar:       0.6      # "Implicitness" parameter for GOTM vertical diffusion (default = 0.6):
# 0 => Forward Euler (fully explicit, first-order accurate)
# 1 => Backward Euler (fully implicit, first-order accurate)
# 0.5 => Crank-Nicolson (semi-implicit, second-order accurate)
dynamic_kz_bbl: 0     # 1 for dynamic (time-dependent) kz_bbl, 0 for static kz_bbl (default = 0)
# For deep water (e.g. >500 m) a static kz_bbl may be a reasonable approximation.
# For shallower water, probably better to set dynamic_kz_bbl = 1; kz in the BBL is then determined by
linearly interpolating between zero at the SWI and the value at the bottom of the hydrodynamic model input water column
kz_bbl_type:   1      # Type of variation of eddy diffusion kz(z) assumed over the benthic boundary layer:
# 0 => constant = kz_bbl_max, 1 => linear (~=> log-layer for velocity, Holtappels & Lorke, 2011)
# This is only used if assuming a static kz_bbl (dynamic_kz_bbl = 0)
kz_bbl_max:    5.E-6  # Maximum eddy diffusivity in the benthic boundary layer [m2/s] (default = 1.0E-5 m2/s)
# This is only used if assuming a static kz_bbl (dynamic_kz_bbl = 0)
dbl_thickness:  0.0005 # Thickness of the diffusive boundary layer [m] (default = 0.0005 m = 0.5 mm)
# Jorgensen and Revsbech (1985) quote a range 1-2 mm over the deep sea floor (Boudreau and
Guinasso, 1982, Wimbush 1976)
# and down to 0.1-0.2 mm over more exposed sediments (Santschi et al., 1983)
# All layers within the DBL (midpoint height above SWI < dbl_thickness) have kz = kz_mol0 (no eddy
diffusivity)
kz_mol0:     1.0E-9   # Molecular diffusivity at infinite dilution [m2/s] (default = 1.0E-9 m2/s)
# Cf. range (0.5-2.7)E-9 m2/s in Boudreau 1997, Table 4.8
# This sets a single constant value for all variables that is subsequently corrected for viscosity and
tortuosity
mu0_musw:       0.94   # Inverse relative viscosity of saline pore water (default = 0.94 from Boudreau 1997 Table 4.10)
# This relates the diffusivity in saline pore water to the infinite-dilution diffusivity
# assuming the approximation from Li and Gregory (1974), see Boudreau (1997) equation 4.107
kz_bioturb_max: 1.0E-11 # Maximum diffusivity due to bioturbation in the sediments [m2/s] (default = 1.0E-11 m2/s)
# Cf. range (1-100) cm2/yr = (0.3-30)E-11 m2/s cited in Soetaert and Middelburg (2009), citing
Middelburg et al. (1997)
# This sets value for upper z_const_bioturb metres, then bioturbation diffusivity decays with scale
z_decay_bioturb.
z_const_bioturb: 0.01  # "Mixed layer depth" in sediments over which bioturbation diffusivity = kz_bioturb_max [m]
(default = 0.02 m)
# Cf. values 0.05 m and 0.01 m used by Soetaert and Middelburg (2009) for well-mixed and anoxic
conditions respectively
# Meire et al. (2013) use 0.05 m as a constant value
z_decay_bioturb: 0.01   # Decay scale of bioturbation diffusivity below z_const_bioturb [m] (default = 0.01 m, following
Soetaert and Middelburg, 2009)
K_O2s:      5.0    # Half-saturation constant for the effect of oxygen on bioturbation and bioirrigation [uM] (default =
5.0 uM)
# Bioturbation diffusivity and bioirrigation rate are modulated by a Michaelis-Menten function with
parameter K_O2s
bioturb_across_SWI: 1  # 1 to allow (interphase) bioturbation diffusion across the SWI (default = 1)
# Bioturbation across the SWI must be interphase mixing rather than the intraphase mixing assumed
within the sediments
```

```
#
#
##---Bioirrigation parameters-------------------------------------------------------------------
#
# Bioirrigation rate alpha = a1_bioirr*exp(-a2_bioirr*z_s), where z_s is depth below the SWI [m]
#
a1_bioirr:    0.0    # Maximum rate of bioirrigation in the sediments [s^-1] (default = 0.E-5)
#  Schluter et al. (2000) infer a range (0-5) d^-1 = (0-6)E-5 s^-1 for a1
#  This range is also broadly consistent with the profiles of alpha inferred by Miele et al. (2001)
a2_bioirr:    50.   # Decay rate with depth of bioirrigation rate [m^-1] (default = 50)
#  Schluter et al. (2000) infer a range (0-1) cm^1 = (0-100) m^-1 for a2
#  This range is also broadly consistent with the profiles of alpha inferred by Miele et al. (2001)
#
#
##---Sedimentation parameters-------------------------------------------------------------------
w_binf:        1.0E-10 # Particulate background burial velocity deep in the sediments where phi = phi_inf [m/s] (default =
1.0E-10 m/s = 0.3 cm/year, but note that true values are highly variable)
#  Soetaert et al. (1996) propose a regression model as a function of water depth:
#  w = 982*D^-1.548, where D is water depth in [m] and w is in cm/year, e.g. for D = 100 m, w = 0.8
20   cm/year = 2.5E-10 m/s
#  Note: Shallow particulate and solute burial velocities are inferred by assuming steady state compaction
(Boudreau, 1997)
dynamic_w_sed: 1      # 1 to enable time-dependent advective velocities in the sediments (default = 0)
#  This uses the modelled (reactive) particulate variables to correct the advective velocities in the
sediments (see calculate_sed)
#  w_binf and phi_inf then define constant background components of these velocities
#
#
##---Porosity parameters-------------------------------------------------------------------------
#
#  Porosity phi = phi_inf + (phi_0-phi_inf)*exp(-z_s/z_decay_phi), where z_s is depth below the SWI [m]
#
phi_0:        0.95   # Maximum porosity at the SWI (default = 0.95, following Soetaert et al., 1996)
phi_inf:      0.80   # Minimum porosity deep in the sediments (default = 0.80, following Soetaert et al., 1996)
z_decay_phi:   0.04   # Exponential decay scale for excess porosity in the upper sediments [m] (default = 0.04,
following Soetaert et al., 1996)
#
#
##---I/O options------------------------------------------------------------------------------------
input_type: 2                        # input forcing type: 0 for sinusoidal changes, 1 to read from ascii, 2 to read from netCDF
(default)
ncoutfile_name: BROM_Sleipner_out20.nc    # netCDF output file name
outfile_name: finish.dat              # ascii output file name
port_initial_state: 1                 # 0 to use FABM default (default), 1 to read from ascii file (icfile_name)
icfile_name: start19.dat              # ascii initial condition file name (needed if port_initial_state = 1)
#The following are only used if reading input from netCDF (input_type = 2)
#Note: NetCDF variables (temperature, salinity, diffusivity) must have either two dimensions (depth, time) or four
dimensions ((latitude, longitude, depth , time) or (longitude, latitude, depth, time))
nc_set_k_wat_bbl: 1                   # 1 (default) to set the no. water column layers to agree with netCDF input
# 0 to use the value k_wat_bbl set above by subsampling the netCDF input
# Note that in both cases the water layer thickness is determined by the netCDF input,
overriding water_layer_thickness above
nc_staggered_grid: 1                  # 1 (default) to assume a staggered input grid, (t,s) at layer midpoints, kz on layer
interfaces (e.g. ROMS, GOTM)
nc_bottom_to_top: 1                   # 1 (default) if netCDF variables are stored with vertical index increasing from
bottom to top (e.g. ROMS, GOTM)
nc_z_increasing_upward: 1             # 1 if netCDF depth variables are increasing upward (e.g. if "depth" is negative)
(default = 0)
ncinfile_name: nns_annual.nc          # netCDF input file name
ncintime_name: time                   # netCDF time dimension name [units since nc_year0-01-01 00:00:00]
nc_year0: 1998                        # reference year for netCDF time variable (default = 1970) WARNING: This MUST be
correctly specified
ncinz_name: z                         # netCDF depth dimension name for layer midpoints (rho points) [m]
```

```
ncinz2_name: z1                          # netCDF depth dimension name for layer interfaces (w points) [m]
ncinlat_name: lat                        # netCDF latitude dimension name (needed if reading 4D variables)
ncinlon_name: lon                        # netCDF longitude dimension name (needed if reading 4D variables)
ncinlat_sel: 1                  # Chosen latitude index (1,2,...,nlat) (needed if reading from 4D variables with nlat > 1)
ncinlon_sel: 1                  # Chosen longitude index (1,2,...,nlon) (needed if reading from 4D variables with nlon >
1)
#
#Below we specify the names of variables in netCDF input files
#Format is <BROM internal name>: <netCDF input name>
#Can also specify a constant scale factor "fac", e.g. to convert units, or correct bias.
#BROM internal variable = fac * netCDF input variable (BROM assumes fac = 1 if not specified here)
#This factor can also be used to apply a simple stoichiometric assumption in lieu of nutrient variable data
#E.g. ncinSis_name: NO3s                  # netCDF input surface silicate variable name [uM] - here using nitrate
#    ncinSis_fac: 1.5                     # scale factor for netCDF input surface silicate - here assuming "extended Redfield
ratio" Si:N = 1.5 mol Si / mol N
#
#2D physical variables used for setting BROM forcings
#These must be arrays of size [no. water column layers (= k_wat_bbl) * no. of days for all available years]
ncint_name: temp                     # netCDF input temperature variable name [degC]
ncins_name: salt                     # netCDF input salinity variable name [psu]
ncinkz_name: nus                     # netCDF input vertical diffusivity variable name [m2/s]
ncinkz_fac: 1.0                      # scale factor for netCDF input vertical diffusivity (default = 1.0)
#
#1D physical variables used for setting BROM forcings
#These must be arrays of size [no. of days for all available years]
ncinEair_name: Eair                  # netCDF input shortwave irradiance in air at water surface [W/m2] (only used if
use_Eair = 1)
ncinEair_fac: 1.0                    # scale factor for netCDF input shortwave irradiance (default = 1.0) (only used if
use_Eair = 1)
ncinhice_name: hice                  # netCDF input ice thickness variable name [m] (only used if use_hice = 1)
ncinhice_fac: 1.0                    # scale factor for netCDF input ice thickness (default = 1.0) (only used if use_hice = 1)
#
#Biogeochemical variables used for setting Dirichlet BCs at surface or bottom (bctype = 3)
#These must be arrays of size [1 * no. of days in repeated period (= days_in_yr)]
ncinNH4s_name: NH4s                  # netCDF input surface ammonium variable name [uM]
ncinNH4s_fac: 1.0                    # scale factor for netCDF input surface ammonium (default = 1.0)
ncinNO3s_name: NO3s                  # netCDF input surface nitrate variable name [uM]
ncinNO3s_fac: 1.0                    # scale factor for netCDF input surface nitrate (default = 1.0)
ncinPO4s_name: PO4s                  # netCDF input surface phosphate variable name [uM]
ncinPO4s_fac: 1.0                    # scale factor for netCDF input surface phosphate (default = 1.0)
ncinSis_name: Sis                    # netCDF input surface silicate variable name [uM]
ncinSis_fac: 1.0                     # scale factor for netCDF input surface silicate (default = 1.0)
ncinAlks_name: ATs                   # netCDF input surface alkalinity variable name [uM]
ncinAlks_fac: 1.0                    # scale factor for netCDF input surface alkalinity (default = 1.0)
#
#Biogeochemical variables used for setting horizontal mixing fluxes
#NOTE: These must be arrays of size [no. water column layers (= k_wat_bbl) * no. of days in repeated period (=
days_in_yr)]
#NOTE: The depth indexing must agree with temperature and salinity inputs
#NOTE: The layer index of the mixing variable is the layer with which it mixes in the internal BROM grid
#    This is does not necessarily reflect the actual depth of the mixing variable in its external location
#NOTE: This information is only used if hmix_<variable name> is > 0, see above
ncinNH4hmix_name: NH4_N              # netCDF input horizontal mixing ammonium variable name [uM]
ncinNH4hmix_fac: 1.0                 # scale factor for netCDF input horizontal mixing ammonium (default = 1.0)
ncinNO3hmix_name: NO3_N              # netCDF input horizontal mixing nitrate variable name [uM]
ncinNO3hmix_fac: 1.0                 # scale factor for netCDF input horizontal mixing nitrate (default = 1.0)
ncinPO4hmix_name: PO4_N              # netCDF input horizontal mixing phosphate variable name [uM]
ncinPO4s_fac: 1.0                    # scale factor for netCDF input horizontal mixing phosphate (default = 1.0)
ncinSihmix_name: NO3_N               # netCDF input horizontal mixing silicate variable name [uM]
ncinSihmix_fac: 1.5                  # scale factor for netCDF input horizontal mixing silicate (default = 1.0)
ncinO2hmix_name: O2_N                # netCDF input horizontal mixing oxygen variable name [uM]
ncinO2hmix_fac: 1.0                  # scale factor for netCDF input horizontal mixing oxygen (default = 1.0)
#
```

```
#Horizontal mixing rates
#NOTE: This must be an array of size [no. water column layers (= k_wat_bbl) * no. of days in repeated period (=
days_in_yr)]
#NOTE: The depth indexing must agree with temperature and salinity inputs
#NOTE: This information is only used if hmix_<variable name> is > 0, see above
ncinhmix_rate_name: hmix_rate          # netCDF input horizontal mixing rates [day^-1]
ncinhmix_rate_fac: 1.0                # scale factor for netCDF input horizontal mixing rate (default = 1.0)
#
#
##---Options for run-time output to screen---------------------------------------------------------
show_maxmin: 0                     # 1 to show the profile maximum and minimum of each variable at the end of each
12  day (default = 0)
show_kztCFL: 0                     # 1/2 to show the max/profile of total vertical diffusivity and associated Courant-
Friedrichs-Lewy number at the end of each day (default = 0)
show_wCFL: 0                       # 1/2 to show the max/profile of vertical advection and associated Courant-Friedrichs-
Lewy number at the end of each day (default = 0)
show_nan: 0                        # 1 to show the profile concentration output on NaN-termination for the offending
variable (default = 1)
show_nan_kztCFL: 2                 # 1/2 to show the max/profile of total vertical diffusivity and associated Courant-
Friedrichs-Lewy number on NaN-termination (default = 1)
show_nan_wCFL: 1                   # 1/2 to show the max/profile of vertical advection and associated Courant-
Friedrichs-Lewy number on NaN-termination (default = 1)
#
#
## References
# Boudreau B.P., 1997. Diagenetic Models and Their Implementation, Springer-Verlag, Berlin.
# Holzbecher, E., 2002. Advective flow in sediments under the influence of compaction. Hydrological Sciences 47(4), 641-
649.
# Link JS, Griswold CA, Methratta ET, Gunnard J, Editors. 2006. Documentation for the Energy Modeling and Analysis
eXercise (EMAX). US Dep. Commer., Northeast Fish. Sci. Cent. Ref. Doc. 06-15; 166 p.
# Meile, C., Koretsky, C.M., Cappellen, P.V., 2001. Quantifying bioirrigation in aquatic sediments: An inverse modeling
approach. Limnol. Oceanogr. 46(1), 164–177.
# Meire, L., Soetaert, K.E.R., Meysman, F.J.R, 2013. Impact of global change on coastal oxygen dynamics and risk of
hypoxia. Biogeosciences 10, 2633–2653.
# Mobley, C.D., Boss, E.S., 2012. Improved irradiances for use in ocean heating, primary production, and photo-oxidation
calculations. Applied Optics 51(27), 6549-6560.
# Schluter, M., Sauter, E., Hansen, H., Suess, E., 2000. Seasonal variations of bioirrigation in coastal sediments: Modelling
of field data. Geochimica et Cosmochimica Acta 64(5), 821–834.
# Soetaert, K., Herman, P.M.J., Middelburg, J.J., 1996. A model of early diagenetic processes from the shelf to abyssal
depths. Geochimica et Cosmochimica Acta 60(6), 1019-1040.
# Soetaert, K., Middelburg, J.J., 2009. Modeling eutrophication and oligotrophication of shallow-water marine systems: the
importance of sediments under stratified and well-mixed conditions. Hydrobiologia 629:239–254.
# Wimbush, M., 2012: The Physics of The Benthic Boundary Layer, in The Benthic Boundary Layer, edited by I. McCave.
```

**Appendix D: run-time input file for BROM-biogeochemistry (fabm.yaml)**

```
# IMPORTANT !!!! _ <TAB> is NOT allowed here, used <Space> only !!!!
# Each entry must have 6 spaces before the parameter name
require_initialization: true
instances:
#-------------------------------------------------------------------------
niva_brom_eqconst:
#-------------------------------------------------------------------------
niva_brom_carb:
initialization:
Alk: 2200.
DIC: 2100.
coupling:
Kc0: niva_brom_eqconst/Kc0
Kc1: niva_brom_eqconst/Kc1
```

```
Kc2: niva_brom_eqconst/Kc2
Kw: niva_brom_eqconst/Kw
Kb: niva_brom_eqconst/Kb
Kp1: niva_brom_eqconst/Kp1
Kp2: niva_brom_eqconst/Kp2
Kp3: niva_brom_eqconst/Kp3
Knh4: niva_brom_eqconst/Knh4
Kh2s: niva_brom_eqconst/Kh2s
#    Kh2s2: niva_brom_eqconst/Kh2s2
KSi: niva_brom_eqconst/KSi
kso4: niva_brom_eqconst/kso4
kflu: niva_brom_eqconst/kflu
tot_free: niva_brom_eqconst/tot_free
#   Constants calculated: Kc0 (Weiss, 1974), Kc1, Kc2 (Roy et al., 1993), Kw, Kp1,Kp2,Kp3 (DOE, 2004),
#                 Kb (Dickson,1990), KSi(Millero,1995), Knh4, Kh2s1(Luff et al, 2001), Kh2s2(Volkov 1984)
#                 dissociation for B, F according to (Dickson et al., 2007), more references in the code.
PO4: niva_brom_bio/PO4
NH4: niva_brom_bio/NH4
DON: niva_brom_bio/DON
Si: niva_brom_redox/Si
H2S: niva_brom_redox/H2S
Mn3: niva_brom_redox/Mn3
Mn4: niva_brom_redox/Mn4
Fe3: niva_brom_redox/Fe3
SO4: niva_brom_redox/SO4
#-------------------------------------------------------------------------
niva_brom_bio:
initialization:
O2: 200.
Phy: 0.01
Het: 0.01
PON: 0.01
DON: 0.0
NO3: 5.
PO4: 1.
NH4: 0.0
parameters:
# ---- Phy ----------
K_phy_gro: 4.7        # Maximum specific growth rate (1/d) =0.9-1.3 (Savchuk, 2002), =3.(Gregoire, Lacroix, 2001) >!0.5
worked for Berre!<
Iopt: 25.          # Optimal irradiance (W/m2) =50 (Savchuk, 2002)
bm: 0.12            # Coefficient for growth dependence on t
43       cm: 1.4            # Coefficient for growth dependence on t
K_phy_mrt: 0.20      # Specific rate of mortality, (1/d) =0.3-0.6 (Savchuk, 2002), =0.05 (Gregoire, Lacroix, 2001)
K_phy_exc: 0.10      # Specific rate of excretion, (1/d) =0.01 (Burchard et al., 2006)
# ----Het -----------
K_het_phy_gro: 1.1   #! Max.spec. rate of grazing of Zoo on Phy, (1/d), =0.9 (Gregoire, Lacroix, 2001), =1.5 (Burchard
et al., 2006)
K_het_phy_lim: 0.5   #! Half-sat.const.for grazing of Zoo on Phy for Phy/Zoo ratio
K_het_pom_gro: 0.50  #! Max.spec.rate of grazing of Zoo on POP and bacteria, (1/d), =1.2 (Burchard et al., 2006)
K_het_pom_lim: 0.05  #! Half-sat.const.for grazing of Zoo on POP for POP/Zoo  ratio
K_het_res: 0.02      #! Specific respiration rate =0.02 (Yakushev et al., 2007)
K_het_mrt: 0.05      #! %! Maximum specific rate of mortality of Zoo (1/d) =0.05 (Gregoire, Lacroix, 2001)
Uz: 0.5           #! Food absorbency for Zoo (nd) =0.5-0.7 (Savchuk, 2002)
55       Hz: 0.5           #! Ratio betw. diss. and part. excretes of Zoo (nd), =0.5 (Gregoire, Lacroix, 2001)
limGrazBac: 2.      #! Limiting parameter for bacteria grazing by Zoo, =2. (Yakushev et al., 2007)
# ----N -------------
K_nox_lim: 0.1       #! Half-sat.const.for uptake of NO3+NO2 (uM) =0.5 (Gregoire, Lacroix, 2001)
K_nh4_lim: 0.02      #! Half-sat.const.for uptake of NH4 (uM) =0.2 (Gregoire, Lacroix, 2001)
K_psi: 1.46          #! Strength of NH4 inhibition of NO3 uptake constant (uM-1) =1.46_rk (Gregoire, Lacroix, 2001)
K_nfix: 0.4          #! Maximum specific rate of N-fixation (1/d) =0.5 (Savchuk, 2002)
# ----P -----------
K_po4_lim: 0.012     #! Half-sat. constant for uptake of PO4 by Phy
```

```
# ----Si------------
K_si_lim: 0.1        #! Half-sat. constant for uptake of Si_lim by Phy
# ----Sinking-------
Wsed: 5.0        #! Rate of sinking of detritus (m/d), =0.4 (Savchuk, 2002), =5. (Gregoire, Lacroix, 2001), =1-370
(Alldredge, Gotschalk, 1988)
Wphy: 0.2        #! Rate of sinking of Phy (m/d), =0.1-0.5 (Savchuk, 2002)
Whet: 1.        #! Rate of sinking of Het (m/d), =1. (Yakushev et al., 2007)
# ---- Stoichiometric coefficients ----
r_n_p: 16.0        #! N[uM]/P[uM]
r_o_n: 6.625        #! O2[uM]/N[uM]
r_c_n: 8.0        #! C[uM]/N[uM]
r_si_n: 1.0        #! Si[uM]/N[uM]
coupling:
NO2: niva_brom_redox/NO2
H2S: niva_brom_redox/H2S
Baan: niva_brom_redox/Baan
Baae: niva_brom_redox/Baae
Bhae: niva_brom_redox/Bhae
Bhan: niva_brom_redox/Bhan
Si: niva_brom_redox/Si
Sipart: niva_brom_redox/Sipart
DIC: niva_brom_carb/DIC
Alk: niva_brom_carb/Alk
Hplus: niva_brom_carb/Hplus
Kp1: niva_brom_eqconst/Kp1
Kp2: niva_brom_eqconst/Kp2
Kp3: niva_brom_eqconst/Kp3
Knh4: niva_brom_eqconst/Knh4
KSi: niva_brom_eqconst/KSi
#-------------------------------------------------------------------------
niva_brom_redox:
initialization:
Mn2: 0.0
Mn3: 0.0
Mn4: 0.0
MnS: 0.0
MnCO3: 0.0
Fe2: 0.0
Fe3: 0.0
FeS: 0.0
FeCO3: 0.0
NO2: 0.0
Si: 0.0
Sipart: 0.0
H2S: 0.0
S0:  0.0
S2O3: 0.0
SO4: 25000.
Baae: 0.01
Bhae: 0.01
Baan: 0.01
Bhan: 0.01
CaCO3: 5.0
CH4: 0.001
FeS2: 0.0
parameters:
# ---- Model parameters ------
Wbact: 0.4        #! Rate of sinking of bacteria (Bhae,Baae,Bhan,Baan) (1/d), (Yakushev et al.,2007)
59       Wm: 7.0        #! Rate of accelerated sinking of particles with settled metal hydroxides (1/d), (Yakushev et al.,2007)
# specific rates of biogeochemical processes
#---- Mn---------
K_mn_ox1: 0.1        #! Specific rate of oxidation of Mn2 to Mn3 with O2 (1/d).
K_mn_ox2: 0.2        #! Specific rate of oxidation of Mn3 to Mn4 with O2 (1/d)
```

K_mn_rd1: 0.5      #! Specific rate of reduction of Mn4 to Mn3 with H2S (1/d)
K_mn_rd2: 1.0      #! Specific rate of reduction of Mn3 to Mn2 with H2S (1/d)
K_mns: 1500.      #! Conditional equilibrium constant for MnS from Mn2 with H2S (M)
K_mns_diss: 0.0005   #! Specific rate of dissolution of MnS to Mn2 and H2S (1/d)
K_mns_form: 0.00001 #! Specific rate of formation of MnS from Mn2 with H2S (1/d)
K_mnco3: 1.      #! Conditional equilibrium constant % 1.8e-11 (M) (Internet)    1 uM2 for Mn2+CO3->MnCO3 (Meysman,2003)
K_mnco3_diss: 7.e-7 #! Specific rate of dissolution of MnCO3 (1/d) =6.8e-7 (2.5 X 10-1 yr-1 (Van Cappellen, Wang, 1996) !1x10-4 yr-1) (Hunter et al, 98)
K_mnco3_form: 0.1e-4 #! Specific rate of formation of MnCO3 (1/d) =2.7e-7  (1. X 10-4 yr-1 (Van Cappellen, Wang, 1996)!1x10-4 yr-1) (Hunter et al, 98))
K_mnco3_ox: 0.0027  #! Specific rate of oxidation of MnCO3 with O2 (1/d)=0.0027  ( 1x10^(-6) M/yr (Van Cappellen, Wang, 1996).
K_DON_mn: 0.001     #! Specific rate of oxidation of DON with Mn4 (1/d)
K_PON_mn: 0.001     #! Specific rate of oxidation of PON with Mn4 (1/d)
s_mnox_mn2: 0.01   #! threshold of Mn2 oxidation (uM Mn) (Yakushev et al.,2007)
s_mnox_mn3: 0.01   #! threshold of Mn3 oxidation (uM Mn) (Yakushev et al.,2007)
s_mnrd_mn4: 0.01   #! threshold of Mn4 reduciton (uM Mn) (Yakushev et al.,2007)
s_mnrd_mn3: 0.01   #! threshold of Mn3 reduciton  (uM Mn) (Yakushev et al.,2007)
#---- Fe---------
K_fe_ox1: 0.5      #!Specific rate of oxidation of Fe2 to Fe3  with O2 (1/d), =4. (Konovalov et al., 2006)
K_fe_ox2: 0.001      #!0.1! Specific rate of oxidation of Fe2 to Fe3  with MnO2 (1/d) =0.74 (Konovalov et al., 2006); 3x10^6 1/(M yr) is estimated in Van Cappellen-Wang-96
K_fe_rd: 1.2      #!0.5! Specific rate of reduction of Fe3 to Fe2  with H2S (1/day) *=0.05 (Konovalov et al., 2006)
K_fes: 2510.0      #!FeS equilibrium constant (Solubility Product Constant) (uM)=2510 ( 2.51x10-6 mol cm-3, Bektursuniva,11)
K_fes_form: 5.e-4    #!Specific rate of precipitation of FeS from Fe2 with H2S (1/day)=1.e-5 (4x10-3 1/yr, Bektursunova,11)
K_fes_diss: 1.e-6      #!Specific rate of dissollution of FeS to Fe2 and H2S  (1/day)=3.e-6 (1x10-3 1/yr, Bektursunova,11)
K_fes_ox: 0.001      #!Specific rate of oxidation of FeS with O2 (1/day)=0.001(3x10^5 1/(M yr),(Van Cappellen, Wang, 1996)
K_DON_fe: 0.00005     #!-0.0003 ! %  Specific rate of oxidation of DON with Fe3 (1/day)
K_PON_fe: 0.00001      #!-0.0001 ! %  Specific rate of oxidation of PON with Fe3 (1/day)
K_fes2_form: 1.e-6    #!specific rate of FeS2 formation by FeS oxidation by H2S (1/day)=0.000009 (10^(-4) L/mol/s (Rickard-97)
K_fes2_ox: 4.38e-4    #!specific rate of pyrite oxidation by O2  (1/uM/d)=4.38x10^(-4) 1/micromolar/day (Wijsman et al -2002).
s_feox_fe2: 0.001      #!threshold of Fe2 reduciton
s_ferd_fe3: 0.01      #!threshold of Fe3 reduciton  (uM Fe)
K_feco3: 15.      #!10. !2.e-2 ! Conditional equilibrium constant % 1.8e-11 (M) (Internet) 1 uM2 for Mn2+CO3->FeCO3 (Meysman,2003)
K_feco3_diss: 7.e-4  #!Specific rate of dissolution of FeCO3 (1/day)=6.8e-7  !2.5 X 10-1 yr-1 (Van Cappellen, Wang, 1996) !1x10-4 yr-1 (Hunter et al, 98)
K_feco3_form: 3.e-4  #!Specific rate of formation of FeCO3 (1/day)=2.7e-7  !! 1. X 10-4 yr-1(Van Cappellen, Wang, 1996)!1x10-4 yr-1 (Hunter et al, 98)
K_feco3_ox: 0.0027    #!Specific rate of oxidation of FeCO3 with O2 (1/day)=0.0027  ( 1x10^(-6) M/yr (Van Cappellen, Wang, 1996).
#---- S---------
K_hs_ox: 0.5      #! Specific rate of oxidation of H2S to S0  with O2 (1/d),  =0.1 (Gregoire, Lacroix, 2001)
K_s0_ox: 0.02      #! 0.02 Specific rate of oxidation of S0 with O2 (1/d), (Yakushev, Neretin,1997)
K_s2o3_ox: 0.01      #! Specific rate of oxidation of S2O3 with O2 (1/d), (Yakushev, Neretin,1997)
K_so4_rd: 5.e-6      #! Specific rate of OM sulfate reduction with sulfate (1/d), (Yakushev, Neretin,1997)
K_s2o3_rd: 0.001      #! Specific rate of OM sulfate reduction with thiosulfate (1/d) (Yakushev, Neretin,1997)
K_s0_disp: 0.001      #! Specific rate of S0 dispropotionation (1/d) (Yakushev,2013)
K_s0_no3: 0.9      #! Specific rate of oxidation of S0 with NO3 (1/d) (Yakushev,2013)
K_s2o3_no3: 0.01      #! Specific rate of oxidation of S2O3 with NO3 (1/d) (Yakushev,2013)
K_mnrd_hs: 1.0      #! half sat. of Mn reduction (uM S) (Yakushev,2013)
K_ferd_hs: 1.0      #! half sat. of Fe reduction (uM S) (Yakushev,2013)
#---- N---------!
K_DON_ox: 0.05      #! Specific rate of oxidation of DON with O2 (1/d) = 0.1(Savchuk, 2002)
K_PON_ox: 0.002      #! Specific rate of oxidation of PON with O2 (1/d) =0.002 (Savchuk, 2002), =0.07 (Gregoire, Lacroix, 2001)
Tda: 13.0      #! Temperature control coefficient for OM decay (Burchard et al., 2006)

beta_da: 20.0      #! Temperature control coefficient for OM decay (Burchard et al., 2006)
K_omox_o2: 1.0      #! Half sat. of o2 for OM mineralization (uM) (Yakushev,2013)
K_PON_DON: 0.1       #! Specific rate of Autolysis of PON to DON (1/d), =0.02 (Burchard et al., 2006)
K_nitrif1: 0.01     #! Spec.rate of 1st st. of nitrification, (1/d), =0.01 (Yakushev,2013) =0.1(Savchuk, 2002) =0.1
(Gregoire, Lacroix, 2001)
K_nitrif2: 0.1      #! Spec.rate of 2d st. of nitrification, (1/d), =0.1 (Yakushev,2013)
K_denitr1: 0.16      #! Spec.rate of 1 stage of denitrif =0.16 (Yakushev, Neretin,1997),= 0.5(Savchuk, 2002),=
0.015(Gregoire, Lacroix, 2001)
K_denitr2: 0.25      #! Spec.rate of 2 stage of denitrif =0.22 (Yakushev, Neretin,1997)
K_omno_no3: 0.001     #! Half sat. of no3 for OM denitr. (uM N) (Yakushev,2013)
K_omno_no2: 0.001     #! Half sat. of no2 for OM denitr. (uM N) (Yakushev,2013)
K_hs_no3: 0.8      #! Spec.rate of thiodenitrification (1/d), =.015 (Gregoire, Lacroix, 2001)
K_annamox: 0.8      #! Spec.rate of Anammox (1/d),  (Gregoire, Lacroix, 2001)
#---- O2--------!
O2s_nf: 5.       #! threshold of O2  saturation for nitrification, (uM), =10. (Gregoire, Lacroix, 2001)
O2s_dn: 10.0       #! threshold of O2 for denitrification, anammox, Mn reduction (uM O2), =40 (0.72 mgO2/l)
(Savchuk, 2002)
s_omox_o2: 0.01    #! threshold of o2 for OM mineralization (uM O2) (Yakushev,2013)
s_omno_o2: 25.0     #! threshold of o2 for OM denitrification (uM O2) (Yakushev,2013)
s_omso_o2: 25.0     #! threshold of o2 for OM sulfate reduction (uM O2) (Yakushev,2013)
s_omso_no3: 5.0     #! threshold of noX for OM sulfate reduction (uM O2) (Yakushev,2013)
K_mnox_o2: 2.0     #! half sat. of Mn oxidation (uM O2) (Yakushev,2013)
#---- C--------!
K_caco3_diss: 3.0    #! CaCO3 dissollution rate constant (1/d) (wide ranges are given in (Luff et al., 2001))
K_caco3_form: 0.0002  #! CaCO3 precipitation rate constant (1/d) (wide ranges are given in (Luff et al., 2001))
K_DON_ch4: 0.00014    #! Specific rate of methane production from DON (1/d) (Lopes et al., 2011)
K_PON_ch4: 0.00014    #! Specific rate of methane production from PON (1/d) (Lopes et al., 2011)
K_ch4_o2: 0.14      #! Specific rate of oxidation of CH4 with O2 (1/d) =0.14 (Lopes et al., 2011)
K_ch4_so4: 0.0000274  #! Specific rate of oxidation of CH4 with SO4 (1/uM/day) (0.0274 m3 /mol-1 day-1 Lopes et al.,
2011)
s_omch_so4: 30.      #! threshold of of SO4 for methane production from OM (uM) (Lopes et al., 2011)
#---- Si-------!
K_sipart_diss: 0.080  #! Si dissollution rate constant (1/d), =0.008 (Popova, Srokosz, 2009)
#---- Bacteria-!
K_Baae_gro: 0.1    #!  Baae maximum specific growth rate (1/d) (Yakushev, 2013)
K_Baae_mrt: 0.005    #!  Baae specific rate of mortality (1/d) (Yakushev et al., 2013)
K_Baae_mrt_h2s: 0.899 #!  Baae increased specific rate of mortality due to H2S (1/d) (Yakushev et al., 2013)
limBaae: 2.0       #! Limiting parameter for nutrient consumprion by Baae (nd) (Yakushev, 2013)
K_Bhae_gro: 0.5     #! Bhae maximum specific growth rate (1/d) (Yakushev, 2013)
K_Bhae_mrt: 0.01     #! Bhae specific rate of mortality (1/d) (Yakushev, 2013)
K_Bhae_mrt_h2s: 0.799 #! Bhae increased specific rate of mortality due to H2S (1/d) (Yakushev, 2013)
limBhae: 5.0       #! Limiting parameter for OM consumprion by Bhae (nd) (Yakushev, 2013)
K_Baan_gro: 0.2     #! Baan maximum specific growth rate (1/d) (Yakushev, 2013)
K_Baan_mrt: 0.005     #! Baan specific rate of mortality (1/d) (Yakushev, 2013)
limBaan: 2.0       #! Limiting parameter for nutrient consumprion by Baan (nd) (Yakushev, 2013)
K_Bhan_gro: 0.15     #! Bhan maximum specific growth rate (1/d) (Yakushev, 2013)
K_Bhan_mrt: 0.01     #! Bhan specific rate of mortality (1/d) (Yakushev, 2013)
K_Bhan_mrt_o2: 0.899  #! Bhan increased specific rate of mortality due to O2 (1/d) (Yakushev, 2013)
limBhan: 2.0       #! Limiting parameter for OM consumprion by Bhan (nd) (Yakushev, 2013)
#---- Stoichiometric coefficients ----!
r_fe_n: 26.5       #! Fe[uM]/N[uM] (Boudreau, 1996)
r_mn_n: 13.25       #! Mn[uM]/N[uM] (Boudreau, 1996)
f: 0.66       #! conversion factor relating solid and dissolved species concentrations
r_fe3_p: 2.7      #! Fe[uM]/P[uM] partitioning coeff. for Fe oxide (Yakushev et al., 2007)
r_mn3_p: 0.67      #! Mn[uM]/P[uM] complex stoichiometric coeff. for Mn(III) (Yakusheve al., 2007)
r_fe3_si: 3.      #! Fe[uM]/Si[uM] partitioning coeff. for Fe oxide
coupling:
O2: niva_brom_bio/O2   # O2: niva_oxydep/oxy
NH4: niva_brom_bio/NH4
NO3: niva_brom_bio/NO3
PO4: niva_brom_bio/PO4
PON: niva_brom_bio/PON
DON: niva_brom_bio/DON

```
Wsed: niva_brom_bio/Wsed
Kp1: niva_brom_eqconst/Kp1
Kp2: niva_brom_eqconst/Kp2
Kp3: niva_brom_eqconst/Kp3
Knh4: niva_brom_eqconst/Knh4
Kh2s: niva_brom_eqconst/Kh2s
KSi: niva_brom_eqconst/KSi
Kc0: niva_brom_eqconst/Kc0
Alk: niva_brom_carb/Alk
DIC: niva_brom_carb/DIC
Hplus: niva_brom_carb/Hplus
Om_Ca: niva_brom_carb/Om_Ca
Om_Ar: niva_brom_carb/Om_Ar
CO3: niva_brom_carb/CO3
pCO2: niva_brom_carb/pCO2
Ca: niva_brom_carb/Ca
```

# REFERENCES:

# Alldredge, A.L. and Gotschalk, C., 1988. In situ settling behavior of marine snow. Limnology and Oceanography, 33(3), pp.339-351.

# Boudreau, B. P.: A method-of-lines code for carbon and nutrient diagenesis in aquatic sediments, Comput. Geosci., 22(5), 479–496, doi:10.1016/0098-3004(95)00115-8, 1996.

# Burchard H., Bolding K., Kühn W., Meister A., Neumann T. and Umlauf L., 2006. Description of a flexible and extendable physical–biogeochemical model system for the water column. Journal of Marine Systems, 61(3), 180-211.

# Dickson AG. 1990. Thermodynamics of the dissociation of boric acid in synthetic seawater from 273.15 to 318.15 K. Deep-Sea Research Part a-Oceanographic Research Papers. 37:755-766

# Dickson, A.G., Sabine, C.L. and Christian, J.R., 2007. Guide to Best Practices for Ocean CO2 Measurements.

# DOE (1994) Handbook of methods for the analysis of the various parameters of the carbon dioxide system in sea water; version 2, A. G. Dickson & C. Goyet, eds., ORNL/CDIAC-74.

# Gregoire M. and Lacroix G., 2001. Study of the oxygen budget of the Black Sea waters using a 3D coupled hydrodynamical–biogeochemical model. Journal of marine systems, 31(1), pp.175-202.

# Konovalov, S.K., Murray, J.W., Luther, G.W., Tebo, B.M., 2006. Processes controlling the Redox budget for oxic/anoxic water column of the Black Sea. Deep Sea Research (II) 53: 1817-1841.

# Link JS, Griswold CA, Methratta ET, Gunnard J, Editors. 2006. Documentation for the Energy Modeling and Analysis eXercise (EMAX). US Dep. Commer., Northeast Fish. Sci. Cent. Ref. Doc. 06-15; 166 p.

# Millero FJ , 1995 Thermodynamics of the carbon dioxide system in the oceans. Geochimica et Cosmochimica Acta 59 (4), 661-677

# Luff R., Haeckel M., and Wallmann K. 2001. Robust and fast FORTRAN and MATLAB libraries to calculate pH distributions in marine systems, Comput. Geosci., 27, 157–169

# Popova EE, Srokosz MA. 2009. Modelling the ecosystem dynamics at the Iceland-Faeroes Front: Biophysical interactions, J. Mar. Syst., 77(1-2), 182–196, doi:10.1016/j.jmarsys.2008.12.005, 2009.

# Savchuk O. 2002. Nutrient biogeochemical cycles in the Gulf of Riga: scaling up field studies with a mathematical model. J. Mar. Syst. 32: 253-280.

# Volkov II. 1984. Geokhimiya Sery v Osadkakh Okeana (Geochemistry of Sulfur in Ocean Sedi-ments), Nauka, Moscow, USSR.

# Van Cappellen P., Wang, Y. F. 1996: Cycling of iron and manganese in surface sediments: A general theory for the coupled transport and reaction of carbon, oxygen, nitrogen, sulfur, iron, and manganese, Am. J. Sci., 296(3), 197–243, doi:10.2475/ajs.296.3.197, 1996.

# Weiss RF. 1974 Carbon dioxide in water and seawater: the solubility of a non-ideal gas.Marine Chemistry 2:203-215.

# Yakushev E., Neretin L., 1997. One-dimensional modelling of nitrogen and sulphur cycles in the aphotic zones of the Black Sea and Arabian Sea. Global Biogeochem. Cycles 11 Ž3.,401–414.

# Yakushev EV, Pollehne F., Jost G., Kuznetsov I., Schneider B., Umlauf L. 2007. Analysis of the water column oxic/anoxic interface in the Black and Baltic seas with a numerical model, Mar. Chem., 107(3), 388–410.

# Yakushev E. 2013. RedOx Layer Model: A Tool for Analysis of the Water Column Oxic/Anoxic Interface Processes. In: E.V.Yakushev (ed.) Chemical Structure of Pelagic Redox Interfaces: Observation and Modeling, Hdb Env Chem (2013) 22: 203–234, DOI 10.1007/698_2012_145, Springer-Verlag Berlin Heidelberg

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

**Table 1. State variables of BROM**

| N | Notation | Name | Units | N | Notation | Name | Units |
|---|---|---|---|---|---|---|---|
| | **N** | **Nitrogen** | | | **O** | **Oxygen** | |
| 1 | $NH_4$ | Ammonia | μM N | 19 | $O_2$ | Dissolved oxygen | μM $O_2$ |
| 2 | $NO_2$ | Nitrite | μM N | | **S** | **Sulfur** | |
| 3 | $NO_3$ | Nitrate | μM N | 20 | $H_2S$ | Hydrogen sulfide | μM S |
| 4 | PON | Particulate organic nitrogen | μM N | 21 | $S^0$ | Total elemental sulfur | μM S |
| 5 | DON | Dissolved organic nitrogen | μM N | 22 | $S_2O_3$ | Thiosulfate and sulfites | μM S |
| | **P** | **Phosphorus** | | 23 | $SO_4$ | Sulfate | μM S |
| 6 | $PO_4$ | Phosphate | μM P | | **C** | **Carbon** | |
| | **Si** | **Silicon** | | 24 | DIC | Dissolved inorganic carbon | μM C |
| 7 | Si | Dissolved silicon | μM Si | 25 | $CH_4$ | Methane | μM C |
| 8 | Si_part | Particulate silicon | μM Si | 26 | $CaCO_3$ | Calcium carbonate | μM Ca |
| | **Mn** | **Manganese** | | | | **Alkalinity** | |
| 9 | $Mn^{2+}$ | Dissolved bivalent manganese | μM Mn | 27 | Alk | Total alkalinity | μM |
| 10 | $Mn^{3+}$ | Dissolved trivalent manganese | μM Mn | | | | |
| 11 | $Mn^{4+}$ | Particulate quadrivalent manganese | μM Mn | | | **Ecosystem parameters** | |
| 12 | MnS | Manganese sulfide | μM Mn | 28 | Phy | Phototrophic producers | μM N |
| 13 | $MnCO_3$ | Manganese carbonate | μM Mn | 29 | Het | Pelagic and benthic heterotrophs | μM N |
| | **Fe** | **Iron** | | 30 | Bhae | Aerobic heterotrophic bacteria | μM N |
| 14 | $Fe^{2+}$ | Dissolved bivalent iron | μM Fe | 31 | Baae | Aerobic autotrophic bacteria | μM N |
| 15 | $Fe^{3+}$ | Particulate trivalent iron | μM Fe | 32 | Bhan | Anaerobic heterotrophic bacteria | μM N |
| 16 | FeS | Iron monosulfide | μM Fe | 33 | Baan | Anaerobic autotrophic bacteria | μM N |
| 17 | $FeS_2$ | Pyrite | μM Fe | | | | |
| 18 | $FeCO_3$ | Ferrous Carbonate | μM Fe | | | | |

1    **Table 2.  Parameterization of the biogeochemical processes**

2    **2.1. Nutrients**

| Name of Process, reference, reaction | Parameterization in the model |
|---|---|
| **Nitrogen** ||
| Autolysis (Savchuk and Wulff, 1996) | Autolysis = K_PON_DON***PON** |
| Mineralization at oxic conditions  (Richards, 1965) $(CH_2O)_{106}(NH_3)_{16}H_3PO_4 + 106O_2 \square$ $106CO_2 + 16NH_3 + H_3PO_4 + 106H_2O$ | $DcDM\_O2 = K\_DON\_ox * \mathbf{DON} * \dfrac{\mathbf{O_2}}{\mathbf{O_2} + K\_omox\_o2} * (1 + beta\_da\ \dfrac{t^2}{t^2 + tda^2})$ $DcPM\_O2 =\ K\_PON\_ox\ * \mathbf{PON} * \dfrac{\mathbf{O_2}}{\mathbf{O_2} + K\_omox\_o2} * (1 + beta\_da\ \dfrac{t^2}{t^2 + tda^2})$ |
| Nitrification  1 stage (Canfield et al., 2005): $NH_4^+ + 1.5O_2 \rightarrow NO_2^- + 2H^+ + H_2O$ | $Nitrif1 = K\_nitrif1 * \mathbf{NH_4} * \mathbf{O_2} * 0.5 * (1.+\tanh(\mathbf{O_2} - O2s\_nf))$ |
| Nitrification 2 stage (Canfield et al., 2005): $NO_2^- + 0.5\ O_2 \rightarrow NO_3^-$ | $Nitrif2 =\ K\_nitrif2 * \mathbf{NO_2} * \mathbf{O_2} * 0.5 * (1.+\tanh(\mathbf{O_2} - O2s\_nf))$ |
| Anammox (Canfield et al., 2005): $NO_2^- + NH_4^+ \square\ N_2 + 2H_2O$ | $Anammox =\ K\_anammox * \mathbf{NO_2} * \mathbf{NH_4} * \left(1 - 0.5 * \left(1 + \tanh(\mathbf{O_2} - O2s\_dn)\right)\right)$ |
| POM denitrification 1st stage: (Anderson et al., 1982) $0.5CH_2O + NO_3^- \rightarrow NO_2^- + 0.5H_2O + 0.5CO_2$ 2d stage: (Anderson et al., 1982) $0.75CH_2O + H^+ + NO_2^- \rightarrow 0.5N_2 + 1.25H_2O + 0.75CO_2$ | $Denitr1\_PM = K\_denitr1 * F\_dnox * \dfrac{\mathbf{NO_3}}{\mathbf{NO_3} + K\_omno\_no3} * \mathbf{PON}$ $Denitr2\_PM = K\_denitr2 * F\_dnox * \dfrac{\mathbf{NO_2}}{\mathbf{NO_2} + K\_omno\_no2} * \mathbf{PON}$ where $F\_dnox = 1 -\ 0.5 * \left(1 + \tanh(\mathbf{O_2} - O2s\_dn)\right)$ $DcPM\_NOX\ = \dfrac{16}{212} * Denitr1\_PM + \dfrac{16}{141.3} * Denitr2\_PM$ |

| | |
|---|---|
| DOM denitrification<br>(Anderson et al., 1982) | $\text{Denitr1\_DM} = \text{K\_denitr1} * \text{F\_dnox} * \dfrac{\mathbf{NO_3}}{\mathbf{NO_3} + \text{K\_omno\_no3}} * \mathbf{DON}$<br><br>$\text{Denitr2\_DM} = \text{K\_denitr2} * \text{F\_dnox} * \dfrac{\mathbf{NO_2}}{\mathbf{NO_2} + \text{K\_omno\_no2}} * \mathbf{DON}$<br><br>where $\text{F\_dnox} = 1 - 0.5 * \left(1 + \tanh(\mathbf{O_2} - \text{O2s\_dn})\right)$<br><br>$\text{DcDM\_NOX} = \dfrac{16}{212} * \text{Denitr1\_DM} + \dfrac{16}{141.3} * \text{Denitr2\_DM}$ |
| **Phosphate** | |
| Complexation with Mn(III)<br>(Yakushev et al., 2007): | mn_p_compl= (mn_ox2+mn_rd2-mn_ox1-mn_rd1)/ r_mn_p |
| Complexation with Fe(III)<br>(Yakushev et al., 2007): | fe_p_compl= (fe_rd-fe_ox1-fe_ox2+4.*DcDM_Fe+4.*DcPM_Fe)/r_fe_p |
| **Silicate** | |
| Dissolution of particulate Si<br>(Popova and Srokosz, 2009): | $\text{sipart\_diss} = \mathbf{Si\_part} * \text{K\_sipart\_diss}$ |
| Complexation with Fe(III): | fe_si_compl= (fe_rd-fe_ox1-fe_ox2+4.*DcDM_Fe+4.*DcPM_Fe)/r_fe_si |

1  **2.2. Redox metals and sulfur**

| Name of Process, reference, reaction | Parameterization in the model |
|---|---|
| **Manganese** | |
| Manganese(II) oxidation (Canfield et al., 2005) <br> $4Mn^{2+} + O_2 + 4H^+ \square\ 4Mn^{3+} + 2H_2O$ | $mn\_ox1 = 0.5 * \left(1 + \tanh(\mathbf{Mn^{2+}} - s\_mnox\_mn2)\right) * K\_mn\_ox1 * \mathbf{Mn^{2+}} * \dfrac{\mathbf{O_2}}{(\mathbf{O_2} + K\_mnox\_o2)}$ |
| Manganese (III) oxidation (Tebo et al., 1997) <br> $2Mn^{3+} + 3H_2O + 0.5O_2\ \square\ 2MnO_2 + 6H^+$ | $mn\_ox2 = 0.5 * (1 + \tanh(\mathbf{Mn^{3+}} - s\_mnox\_mn2)) * K\_mn\_ox2 * \mathbf{Mn^{3+}} * \dfrac{O_2}{(O_2 + K\_mnox\_o2)}$ |
| Manganese (IV) reduction (Canfield et al., 2005) <br> $2MnO_2 + 7H^+ + HS^-\square 2Mn^{3+} + 4H_2O + S^0$ | $mn\_rd1 = 0.5 * \left(1 + \tanh(\mathbf{Mn^{4+}} - s\_mnrd\_mn4)\right) * K\_mn\_rd1 * \mathbf{Mn^{4+}} * \dfrac{\mathbf{H_2S}}{(\mathbf{H_2S} + K\_mnrd\_hs)}$ |
| Manganese (III) reduction <br> $2Mn^{3+} + HS^-\square\ 2Mn^{2+} + S^0 + H^+$ | $mn\_rd2 = 0.5 * \left(1 + \tanh(\mathbf{Mn^{3+}} - s\_mnrd\_mn3)\right) * K\_mn\_rd2 * \mathbf{Mn^{3+}} * \dfrac{\mathbf{H_2S}}{(\mathbf{H_2S} + K\_mnrd\_hs)}$ |
| MnS formation/dissollution (Davison, 1993) : <br> $Mn^{2+} + HS^- \leftrightarrow MnS\ +\ H^+$ | $mns\_form = K\_mns\_form * \max(0, \left(\dfrac{\mathbf{H_2S} * \mathbf{Mn^{2+}}}{K\_mns * \mathbf{H^+}} - 1\right))$ <br><br> $mns\_diss =\ K\_mns\_diss * \mathbf{MnS} * \max(0, \left(1 - \dfrac{\mathbf{H_2S} * \mathbf{Mn^{2+}}}{K\_mns * \mathbf{H^+}}\right))$ |
| MnCO$_3$ precipitation/dissolution <br> (Van Cappellen and Wang, 1996): <br> $Mn^{2+} + CO_3^{2-} \leftrightarrow \square\ \square nCO_3$ | $mnco3\_prec = K\_mnco3\_pres * \max(0, \left(\dfrac{\mathbf{Mn^{2+}} * \mathbf{CO_3}}{K\_mnco3} - 1\right))$ <br><br> $mnco3\_diss =\ K\_mnco3\_diss * \mathbf{MnCO_3} * \max(0, \left(1 - \dfrac{\mathbf{Mn^{2+}} * \mathbf{CO_3}}{K\_mnco3}\right))$ |
| MnCO$_3$ oxidation by O$_2$ (Morgan, 2000): <br> $2\ MnCO_3 + O_2 + 2H_2O\ \square\ 2\ MnO_2 + 2HCO_3^- + 2H^+$ | $mnco3\_ox =\ K\_mnco3\_ox * \mathbf{MnCO_3} * \mathbf{O_2}$ |
| Manganese reduction for PON (Boudreau, 1996): <br> $(CH_2O)_{106}(NH_3)_{16}H_3PO_4 + 212MnO_2 + 318CO_2 + 106H_2O \rightarrow$ | $DcPM\_Mn = K\_PON\_mn * \mathbf{PON} * \dfrac{\mathbf{Mn^{4+}}}{\mathbf{Mn^{4+}} + 0.5} * (1 - 0.5 * (1 + \tanh(\mathbf{O_2} - O2s\_dn\ )))$ |

| | |
|---|---|
| $424HCO_3^- + 212\ Mn^{2+} + 16NH_3 + H_3PO_4$ | |
| Manganese reduction for DON  (Boudreau, 1996): | $DcDM\_Mn = K\_DON\_mn * \mathbf{DON} * \dfrac{\mathbf{Mn^{4+}}}{\mathbf{Mn^{4+}} + 0.5} * (1 - 0.5 * (1 + \tanh(\mathbf{O_2} - O2s\_dn\ ))$ |
| **Iron** | |
| Fe (II) oxidation with $O_2$ (Van Cappellen and Wang, 1996): <br> $4Fe^{2+} + O_2 + 10H_2O\ \square\rightarrow\square\square Fe(OH)_3 + 8H^+$ | $fe\_ox1 = 0.5 * (1 + \tanh(\mathbf{Fe^{2+}} - s\_feox\_fe2)) * K\_fe\_ox1 * \mathbf{O_2} * \mathbf{Fe^{2+}}$ |
| Fe (II) oxidation with Mn oxide (Van Cappellen and Wang, 1996): <br> $2Fe^{2+} + MnO_2 + 4H_2O \rightarrow\square\square Fe(OH)_3 + Mn^{2+} + 2\ H^+$ | $fe\_ox2 = 0.5 * \left(1 + \tanh(\mathbf{Fe^{2+}} - s\_feox\_fe2)\right) * K\_fe\_ox2 * \mathbf{Mn^{4+}} * \mathbf{Fe^{2+}}$ |
| Fe (III) reduction (Volkov, 1984): <br> $2Fe(OH)_3 + HS^- + 5H^+\square\ 2Fe^{2+} + S^0 + 6H_2O$ | $fe\_rd = 0.5 * (1. + \tanh(\mathbf{Fe^{3+}} - s\_feox\_fe3)) * K\_fe\_rd * \mathbf{Fe^{3+}} * \dfrac{\mathbf{H_2S}}{\mathbf{H_2S} + K\_ferd\_hs}$ |
| FeS formation/dissolition <br>  (Bektursunova and L'Heureux, 2011) : <br> $Fe^{2+} + HS^- \leftrightarrow FeS + H^+$ | $fes\_prec = K\_fes\_form * \max\left(0, \left(\dfrac{\mathbf{H_2S} * \mathbf{Fe^{2+}}}{K\_fes * \mathbf{H^+}} - 1\right)\right)$ <br><br> $fes\_diss = K\_fes\_diss * \mathbf{FeS} * \max\left(0, \left(1 - \dfrac{\mathbf{H_2S} * \mathbf{Fe^{2+}}}{K\_fes * \mathbf{H^+}}\right)\right)$ |
| FeS oxidation (Soetaert et al., 2007): <br> $FeS + 2.25O_2 + 2.5H_2O \rightarrow\square Fe\,(OH)_3 + 2H^+ + SO_4^{2-}$ | $fes\_ox = K\_fes\_ox * \mathbf{O_2} * \mathbf{FeS}$ |
| Pyrite formation (Rickard and Luther, 1997; Soetaert et al., 2007): $FeS + H_2S\ \square\square FeS_2\ + H_2$ | $fes2\_form = K\_fes2\_form * \mathbf{H_2S} * \mathbf{FeS}$ |
| Pyrite oxidation by $O_2$ (Wijsman et al., 2002): <br> $FeS_2 + 3.5O_2 + H_2O\square Fe^{2+} + 2SO_4^{2-} + 2H^+$ | $fes2\_ox = K\_fes2\_ox * \mathbf{FeS_2} * \mathbf{O_2}$ |
| $FeCO_3$ precipitation/dissolution (Van Cappellen and Wang, 1996): | $feco3\_form = K\_feco3\_form * \max\left(0, \left(\dfrac{\mathbf{Fe^{2+}} * \mathbf{CO_3}}{K\_feco3} - 1\right)\right)$ |

| | |
|---|---|
| $Fe^{2+}+CO_3^- \leftrightarrow \square FeCO_3$ | $feco3\_diss = K\_feco3\_diss * \mathbf{FeCO_3} * \max\left(0, \left(1 - \dfrac{\mathbf{Fe^{2+}} * \mathbf{CO_3}}{K\_feco3}\right)\right)$ |
| $FeCO_3$ oxidation by $O_2$ (Morgan, 2000): <br> $2\ FeCO_3 + O_2 + 2H_2O \square\ 2\ FeO_2 + 2HCO_3^- + 2H^+$ | $feco3\_ox = K\_feco3\_ox * \mathbf{FeCO_3} * \mathbf{O_2}$ |
| Iron reduction for DON (Boudreau, 1996): <br> $(CH_2O)_{106}(NH_3)_{16}H_3PO_4 + 424\ Fe(OH)_3 + 742CO_2 \rightarrow$ <br> $848HCO_3^- + 424\ Fe^{2+} + 318\ H_2O + 16NH_3 + H_3PO_4$ | $DcDM\_Fe = K\_DON\_fe * \mathbf{DON} * \mathbf{Fe^{3+}} * \left(1. -0.5 * \left(1 + \tanh(\mathbf{O_2} - O2s\_dn)\right)\right)$ |
| Iron reduction for PON (Boudreau, 1996): | $DcPM\_Fe = K\_PON\_fe * \mathbf{PON} * \mathbf{Fe^{3+}} * \left(1. -0.5 * \left(1 + \tanh(\mathbf{O_2} - O2s\_dn)\right)\right)$ |
| **Sulfur** | |
| $S^0$ disproportionation (Canfield et al., 2005): <br> $4S^0 + 3H_2O \rightarrow 2H_2S + S_2O_3^{2-} + 2H^+$ | $s0\_disp = K\_s0\_disp * \mathbf{S^0}$ |
| Sulphide oxidation with $O_2$ (Volkov, 1984): <br> $2H_2S + O_2 \rightarrow 2S^0 + 2H_2O$ | $hs\_ox = K\_hs\_ox * \mathbf{H_2S} * \mathbf{O_2}$ |
| $S^0$ oxidation with $O_2$ (Volkov, 1984): <br> $2S^0 + O_2 + H_2O \rightarrow S_2O_3^{2-} + 2H^+$ | $s0\_ox = K\_s0\_ox * \mathbf{S^0} * \mathbf{O_2}$ |
| $S^0$ oxidation with $NO_3$ (Kamyshny et al., 2013): <br> $4S^0 + 3NO_3^- + 7H_2O \rightarrow 4SO_4^{2-} + 3NH_4^+ + 2H^+$ | $s0\_no3 = K\_s0\_no3 * \mathbf{NO_3} * \mathbf{S^0}$ |
| $S_2O_3$ oxidation with $O_2$: (Volkov, 1984): <br> $S_2O_3^{2-} + 2O_2 + 2OH^- \rightarrow 2SO_4^{2-} + H_2O$ | $s2o3\_ox = K\_s2o3\_ox * \mathbf{S_2O_3} * \mathbf{O_2}$ |
| $S_2O_3$ oxidation with $NO_3$: (Kamyshny et al., 2013) <br> $S_2O_3^{2-} + NO_3^- + 2H_2O \rightarrow 2SO_4^{2-} + NH_4^+$ | $s2o3\_no3 = K\_s2o3\_no3 * \mathbf{NO_3} * \mathbf{S_2O_3}$ |

| | |
|---|---|
| Thiodenitrification:<br>(Schippers and Jorgensen, 2002; Volkov, 1984) $5H_2S+8NO_3^-$ $+2OH^- \rightarrow 5SO_4^{2-} +4N_2 + 6H_2O$ | $hs\_no3 = K\_hs\_no3 * \mathbf{H_2S} * \mathbf{NO_3}$ |
| POM sulfate reduction 1st and 2d stages (Boudreau, 1996):<br>$(CH_2O)_{106}(NH_3)_{16}H_3PO_4+ 53SO_4^{2-} \rightarrow 106HCO_3^- + 16NH_3 +$ $H_3PO_4 + 53H_2S$ | $so4\_rd\_PM = K\_so4\_rd * F\_sox * F\_snx * \mathbf{SO_4} * \mathbf{PON}$<br>$s2o3\_rd\_PM = K\_s2o3\_rd * F\_sox * F\_snx * \mathbf{S_2O_3} * \mathbf{PON}$<br>$F\_sox = 1 - 0.5 * (1.+\tanh(\mathbf{O_2} - s\_omso\_o2))$<br>$F\_snx = 1 - 0.5 * (1.+\tanh(\mathbf{NO_3} - s\_omso\_no3))$<br>$DcPM\_SO4 = \frac{16}{53} * (so4\_rd\_PM + s2o3\_rd\_PM)$ |
| DOM sulfate reduction 1st and 2d stages (Boudreau, 1996): | $so4\_rd\_DM = K\_so4\_rd * F\_sox * F\_snx * \mathbf{SO_4} * \mathbf{DON}$<br>$s2o3\_rd\_DM = K\_s2o3\_rd * F\_sox * F\_snx * \mathbf{S_2O_3} * \mathbf{DON}$<br>$DcDM\_SO4 = \frac{16}{53} * (so4\_rd\_PM + s2o3\_rd\_PM)$ |

2     **2.3. Carbon and Alkalinity**

| Name of Process, reference, reaction | Parameterization in the model |
|---|---|
| $CaCO_3$ formation/dissolution (Luff et al., 2001):<br>$Ca^{2+} + CO_3^2 \leftrightarrow CaCO_3$ | $caco3\_form = K\_caco3\_form * \max\left(0, \left(\frac{\mathbf{Ca^{2+}} * \mathbf{CO_3}}{K\_caco3} - 1\right)\right)$<br><br>$caco3\_diss = K\_caco3\_diss * \mathbf{CaCO_3} * \max\left(0, \left(1 - \frac{\mathbf{Ca^{2+}}*\mathbf{CO_3}}{K\_caco3}\right)\right)^{4.5}$ |
| $CH_4$ formation from PON, methanogenesis (Boudreau, 1996) :<br>$(CH_2O)_{106}(NH_3)_{16}H_3PO_4 \square$<br>$53CO_2 + 53CH_4 + 16NH_3 + H_3PO_4$ | $DcPM\_CH4 = K\_PON\_ch4 * F\_sox * F\_snx * F\_ssx * \mathbf{CH_4} * \mathbf{PON}$<br>$F\_sox = 1 - 0.5 * (1.+\tanh(\mathbf{O_2} - s\_omso\_o2))$<br>$F\_snx = 1 - 0.5 * (1.+\tanh(\mathbf{NO_3} - s\_omso\_no3))$<br>$F\_ssx = 1 - 0.5 * (1.+\tanh(\mathbf{SO_4} - s\_omch\_so4))$ |
| $CH_4$ formation from DON, methanogenesis (Boudreau, 1996) | $DcDM\_CH4 = K\_DON\_ch4 * F\_sox * F\_snx * F\_ssx * \mathbf{CH_4} * \mathbf{DON}$ |

| | |
|---|---|
| $CH_4$ oxidation by $O_2$ (Boudreau, 1996) :<br>$CH_4 + 2O_2 + \rightarrow CO_2 + 2H_2O$ | $ch4\_o2 = K\_ch4\_o2 * \mathbf{CH_4} * \mathbf{O_2}$ |
| Alkalinity changes<br>(Dickson, 1992; Wolf-Gladrow et al., 2007) | $\begin{aligned} dAlk = {} & -\text{Nitrif1} + \text{Denitr2\_PM} + \text{Denitr2\_DM} + 2*(\text{so4}_{rd} + \text{s2o3}_{rd}) + \text{mn\_ox1} - 3 \\ & * \text{mn\_ox2} + 3*\text{mn\_rd1} - \text{mn\_rd2} - 2*\text{mns\_form} + 2*\text{mns\_diss} - 2 \\ & * \text{mnco3\_form} + 2*\text{mnco3\_diss} + 26.5*(\text{DcDM}_{Mn} + \text{DcPM}_{Mn}) - 2*\text{fe\_ox1} \\ & - \text{fe\_ox2} + 2*\text{fe\_rd} - \text{fes\_form} + \text{fes\_diss} - 2*\text{fes\_ox} - 2*\text{fes2\_ox} + 53 \\ & * (\text{DcDM}_{Fe} + \text{DcPM}_{Fe}) - 0.5*\text{Disprop} + \text{s0\_ox} - 0.5*\text{s0\_no3} - \text{s2o3\_ox} \\ & - 0.4*\text{hs\_no3} - 2*\text{caco3\_form} + 2*\text{caco3\_diss} + \text{GrowthPhy}*\left(\frac{\text{LimNO3}}{\text{LimN}}\right) \\ & - \text{GrowthPhy}*\left(\frac{\text{LimNH4}}{\text{LimN}}\right) \end{aligned}$ |

 **2.4. Ecosystem processes**

| Name of Process, reference, reaction | Parameterization in the model |
|---|---|
| **Phytoplankton** | |
| Influence of the irradiance on photosynthesis | $\text{LimLight} = (^{\text{Iz}}/_{\text{Iopt}}) * e^{(1-\text{Iz}/\text{Iopt})}$ |
| Influence of temperature on photosynthesis | $\text{LimT} = e^{\,(\text{bm}*\text{t}-\text{cm})}$ |
| Dependence of photosynthesis on P | $\text{LimP} = \dfrac{(\mathbf{PO_4}/\mathbf{Phy})^2}{(\text{K\_po4\_lim}*\text{r\_n\_p})^2 + (\mathbf{PO_4}/\mathbf{Phy})^2}$ |
| Dependence of photosynthesis on NO$_3$ | $\text{LimNO}_3 = \dfrac{((\mathbf{NO_3}+\mathbf{NO_2})/\mathbf{Phy})^2}{\text{K\_nox\_lim}^2 + ((\mathbf{NO_3}+\mathbf{NO_2})/\mathbf{Phy})^2}\exp(-\text{K\_psi}\dfrac{(\mathbf{NH_4}/\mathbf{Phy})^2}{\text{K\_nh4\_lim}^2+(\mathbf{NH_4}/\mathbf{Phy})^2})$ |
| Dependence of photosynthesis on NH$_4$ | $\text{LimNH}_4 = \dfrac{\left(\frac{\mathbf{NH_4}}{\mathbf{Phy}}\right)^2}{\text{K\_nh4\_lim}^2+\left(\frac{\mathbf{NH_4}}{\mathbf{Phy}}\right)^2}(1 - \exp(-\text{K\_psi}\dfrac{(\mathbf{NH_4}/\mathbf{Phy})^2}{\text{K\_nh4\_lim}^2+(\mathbf{NH_4}/\mathbf{Phy})^2}))$ |
| Influence of N on photosynthesis | $\text{LimN} = \text{LimNO}_3 + \text{LimNH}_4$ |
| Growth of phytoplankton | $\text{GrowthPhy} = \text{K\_phy\_gro} * \text{LimLight} * \text{LimT} * \min(\text{LimP}, \text{LimN}) * \mathbf{Phy}$ |
| Excretion rate of phytoplankton | $\text{ExcrPhy} = \text{K\_phy\_exc} * \mathbf{Phy}$ |
| Phytoplankton mortality rate | $\text{MortPhy} = (\text{K\_phy\_mrt} + 0.45 * (0.5 - 0.5 * \tanh(\mathbf{O_2} - 60)) + 0.45 * (0.5 - 0.5 * \tanh(\mathbf{O_2} - 20))) * \mathbf{Phy}$ |
| **Heterotrophs** | |
| Grazing of Heterotrophs | $\text{Grazing} = \text{GrazPhy} + \text{GrazPOP} + \text{GrazBact}$ |
| Grazing of Het. on phytoplankton | $\text{GrazPhy} = \text{K\_het\_phy\_gro} * \mathbf{Het} * \dfrac{(\mathbf{Phy}/(\mathbf{Het} + 10^{-4}))^2}{\text{K\_het\_phy\_lim}^2 + (\mathbf{Phy}/(\mathbf{Het} + 10^{-4}))^2}$ |

| | |
|---|---|
| Grazing of Het. on detritus | $$\text{GrazPOP} = \text{K\_het\_pom\_gro} * \textbf{Het} * \frac{(\frac{\textbf{PON}}{\textbf{Het} + 10^{-4}})^2}{\text{K\_het\_pom\_lim}^2 + (\frac{\textbf{PON}}{\textbf{Het} + 10^{-4}})^2}$$ |
| Grazing of Het. on bacteria | $\text{GrazBact} = \text{GrazBaae} + \text{GrazBaan} + \text{GrazBhae} + \text{GrazBhan}$ |
| Grazing of Het. on bacteria autotrophic aerobic | $$\text{GrazBaae} = \text{K\_het\_pom\_gro} * \textbf{Het} * \frac{(\textbf{Baae}/(\textbf{Het} + 10^{-4}))^2}{\text{limGrazBac}^2 + (\textbf{Baae}/(\textbf{Het} + 10^{-4}))^2}$$ |
| Grazing of Het. on bacteria autotrophic anaerobic | $$\text{GrazBaan} = 0.5 * \text{K\_het\_pom\_gro} * \textbf{Het} * \frac{(\textbf{Baan}/(\textbf{Het} + 10^{-4}))^2}{\text{limGrazBac}^2 + (\textbf{Baan}/(\textbf{Het} + 10^{-4}))^2}$$ |
| Grazing of Het. on bacteria heterotrophic aerobic | $$\text{GrazBhae} = \text{K\_het\_pom\_gro} * \textbf{Het} * \frac{(\textbf{Bhae}/(\textbf{Het} + 10^{-4}))^2}{\text{limGrazBac}^2 + (\textbf{Bhae}/(\textbf{Het} + 10^{-4})^2}$$ |
| Grazing of Het. on bacteria heterotrophic anaerobic | $$\text{GrazBhan} = 1.3 * \text{K\_het\_pom\_gro} * \textbf{Het} * \frac{(\textbf{Bhan}/\textbf{Het} + 0.0001)^2}{\text{limGrazBac}^2 + (\textbf{Bhan}/\textbf{Het} + 10^{-4})^2}$$ |
| Respiration rate of Het. | $\text{RespHet} = \text{K\_het\_res} * \textbf{Het} * (0.5 + 0.5 * \tanh(\textbf{O}_2 - 20))$ |
| Mortality of Het. | $$\text{MortHet} = \textbf{Het} * \begin{pmatrix} 0.25 + 0.3 * (0.5 - 0.5 * \tanh(\textbf{O}_2 - 20)) \\ + 0.45 * (0.5 + 0.4 * \tanh(\textbf{H}_2\textbf{S} - 10)) \end{pmatrix}$$ |
| **Bacteria** | |
| Growth rate of Bacteria aerobic autotrophic | $(\text{ChemBaae} = \text{Nitrif1} + \text{Nitrif2} + \text{mn\_ox1} + \text{fe\_ox1} + \text{s2o3\_ox} + \text{s0\_ox} + \text{anammox}) * \text{k}_{\text{Baae}_\text{gro}} * \textbf{Baae}$ $$* \min(\frac{(\textbf{NH}_4 / ((\textbf{Baae} + 10^{-4})^2}{\text{limBaae}^2 + (\textbf{NH}_4 / (\textbf{Baae} + 10^{-4}))^2}, \frac{(\textbf{PO}_4 / (\textbf{Baae} + 10^{-4}))^2}{\text{limBaae}^2 + (\textbf{PO}_4 / (\textbf{Baae} + 10^{-4}))^2})$$ |
| Rate of mortality of Bacteria aerobic autotrophic | $\text{MortBaae} = \text{K\_Baae\_mrt} + \text{K\_Baae\_mrt\_h2s} * 0.5 * (1 - \tanh(1 - \textbf{H}_2\textbf{S})) * \textbf{Baae}^2$ |

| | |
|---|---|
| Growth rate of Bacteria aerobic heterotrophic | $\text{HetBhae} = (\text{DcPM\_O2} + \text{DcDM\_O2}) * \text{K\_Bhae\_gro} * \textbf{Bhae} * \frac{(\textbf{DON}/(\textbf{Bhae}+10^{-4}))^2}{\text{limBhae}^2 + (\textbf{DON}/(\textbf{Bhae}+10^{-4}))^2}$ |
| Rate of mortality of Bacteria aerobic heterotrophic | $\text{MortBhae} = \text{K\_Bhae\_mrt} + \text{K\_Bhae\_mrt\_h2s} * \textbf{Bhae} * 0.5 * (1 - \tanh(1 - \textbf{H}_2\textbf{S}))$ |
| Growth rate of Bacteria anaerobic autotrophic | $\text{ChemBaan} = (\text{mn\_rd1} + \text{mn\_rd2} + \text{fe\_rd} + \text{hs\_ox} + \text{hs\_no3}) * \text{K\_Baan\_gro} * \textbf{Baan}$ $* \min(\frac{(\textbf{NH4}/(\textbf{Baan}+10^{-4}))^2}{\text{limBaan}^2 + (\textbf{NH4}/(\textbf{Baan}+10^{-4}))^2}$ |
| Rate of mortality of Bacteria anaerobic autotrophic | $\text{MortBaan} = \text{K\_Baan\_mrt} * \textbf{Baan}$ |
| Growth rate of Bacteria anaerobic heterotrophic | $\text{HetBhan} = (\text{DcPM\_NOX} + \text{DcDM\_NOX} + \text{DcDM\_Mn} + \text{DcPM\_Mn} + \text{DcDM\_Fe} + \text{DcPM\_Fe} + \text{DcDM\_SO4}$ $+ \text{DcPMSO4} + \text{DcDM\_CH4} + \text{DcPM\_CH4}) * \text{K\_Bhan\_gro} * \textbf{Bhan}$ $* \frac{(\textbf{DON}/(\textbf{Bhan}+10^{-4}))^2}{\text{limBhan}^2 + (\textbf{DON}/(\textbf{Bhan}+10^{-4}))^2}$ |
| Rate of mortality of Bacteria anaerobic heterotrophic | $\text{MortBhan} = \text{K\_Bhan\_mrt} + \text{K\_Bhan\_mrt\_o2} * \textbf{Bhan} * (0.5 + 0.5 * \tanh(1 - \textbf{O}_2))$ |
| Summarized OM mineralization | $\text{Dc\_OM\_total} = \text{DcDM\_O2} + \text{DcPM\_O2} + \text{DcPM\_NOX} + \text{DcDM\_NOX} + \text{DcDM\_Mn} + \text{DcPM\_Mn} + \text{DcDM\_Fe} + \text{DcPM\_Fe} + \text{DcDM\_SO4} + \text{DcPM\_SO4} + 0.5 * (\text{DcDM\_CH4} + \text{DcPM\_CH4})$ |

**Table 3. Parameters names, notations, values and units of the coefficients used in the model**

**Table 3.1. Nutrients and oxygen**

| Parameter | Notation | Units | Value | Reference ranges |
|---|---|---|---|---|
| **Nitrogen** | | | | |
| Specific rate of DON oxidation of with $O_2$ | K_DON_ox | $d^{-1}$ | $1*10^{-2}$ | 0.1 (Savchuk, 2002) |
| Specific rate of PON oxidation of with $O_2$ | K_PON_ox | $d^{-1}$ | $2*10^{-3}$ | 0.002 (Savchuk, 2002) |
| Temperature control threshold coefficient for OM decay | Tda | $^{o}C$ | 13 | 13 (Burchard et al., 2006) |
| Temperature control coefficient for OM decay | beta_da | - | 20 | 20 (Burchard et al., 2006) |
| Half-saturation constant of $O_2$ for OM mineralization | K_omox_o2 | µM | 1 | 1 (Yakushev, 2013) |
| Specific rate of autolysis, PON to DON | K_PON_DON | $d^{-1}$ | 0.1 | 0.02 (Burchard et al., 2006) |
| Half saturation constant for uptake of $NO_3+NO_2$ | K_nox_lim | µM | 0.12 | 0.5 (Gregoire and Lacroix, 2001) |
| Half saturation constant for uptake of $NH_4$ | K_nh4_lim | µM | $2*10^{-2}$ | 0.2 (Gregoire and Lacroix, 2001) |
| Strength of NH4 inhibition of NO3 uptake constant | K_psi | - | 1.46 | 1.46 (Gregoire and Lacroix, 2001) |
| Specific rate of the 1st stage of nitrification | K_nitrif1 | $d^{-1}$ | $1*10^{-2}$ | 0.01 (Yakushev, 2013) |
| Specific rate of the 2d stage of nitrification | K_nitrif2 | $d^{-1}$ | 0.1 | 0.1 (Yakushev, 2013) |
| Specific rate of 1st stage of denitrification | K_denitr1 | $d^{-1}$ | 0.16 | 0.16 (Yakushev and Neretin, 1997), 0.5 (Savchuk, 2002) |
| Specific rate of 2d stage of denitrification | K_denitr2 | $d^{-1}$ | 0.25 | 0.22 (Yakushev, Neretin, 1997) |
| Half-saturation of $NO_3$ for OM denitrification | k_omno_no3 | µM N | $1*10^{-3}$ | $1*10^{-3}$ (Yakushev, 2013) |
| Half-saturation of $NO_2$ for OM denitrification | k_omno_no2 | µM N | $1*10^{-3}$ | $1*10^{-3}$ (Yakushev, 2013) |
| Specific rate of thiodenitrification | K_hs_no3 | $µM^{-1}\ d^{-1}$ | 0.8 | 0.8 (Yakushev and Neretin, 1997), 0.015 (Gregoire and Lacroix, 2001) |
| Specific rate of anammox | K_annamox | $d^{-1}$ | 0.8 | 0.8 (Gregoire and Lacroix, 2001), 0.03 (Yakushev et al., 2007) |
| **Oxygen** | | | | |
| Half-saturation constant for nitrification | O2s_nf | µM | 5. | 10 (Gregoire and Lacroix, 2001) |
| Half-saturation constant for denitrification anammox, Mn reduction | O2s_dn | µM | 10 | 40 (Savchuk, 2002) |
| Threshold value of $O_2$ for OM mineralization | s_omox_o2 | µM | $1*10^{-2}$ | $1*10^{-2}$ (Yakushev, 2013) |
| Threshold value of $O_2$ for OM denitrification | s_omno_o2 | µM | 25 | 25 (Yakushev, 2013) |
| Threshold value of $O_2$ for OM sulfate reduction | s_omso_o2 | µM | 25 | 25 (Yakushev, 2013) |
| Threshold value of NO for OM sulfate reduction | s_omso_no3 | µM | 5 | 5 (Yakushev, 2013) |

| Stoichiometric coefficients | | | | |
|---|---|---|---|---|
| N/P | r_n_p | - | 16 | (Richards, 1965) |
| O/N | r_o_n | - | 6.625 | (Richards, 1965) |
| C/N | r_c_n | - | 8 | (Richards, 1965) |
| Si/N | r_si_n | - | 1 | (Richards, 1965) |
| Fe/N | r_fe_n | - | 26.5 | (Boudreau, 1996) |
| Mn/N | r_mn_n | - | 13.25 | (Boudreau, 1996) |
| Phosphorus | | | | |
| Half-saturation constant for uptake of $PO_4$ by phytoplankton | K_po4_lim | μM | 0.02 | 0.01 (Yakushev et al., 2007) |
| Fe/P ratio in complexes with Fe oxides | r_fe_p | | 2.7 | (Yakushev et al., 2007) |
| Mn/P ratio in complexes with Mn(III) | r_mn_p | | 0.67 | (Yakushev et al., 2007) |
| Silicon | | | | |
| Specific rate of Si dissolution | K_sipart_diss | $d^{-1}$ | 0.008 | 0.008 (Popova, Srokosz, 2009) |
| Half-saturation constant for uptake of Si by phytoplankton | K_si_lim | - | 0.1 | 0.1 (Popova and Srokosz, 2009) |
| Fe/P ratio in complexes with Fe oxides | r_fe_si | | 2.7 | 2.7 (Yakushev et al., 2007) |

**Table 3.2. Redox metals and sulfur**

| Parameter | Notation | Units | Value | Reference ranges |
|---|---|---|---|---|
| Manganese | | | | |
| Specific rate of Mn(II) oxidation to Mn(III) with $O_2$ | K_mn_ox1 | $d^{-1}$ | 0.1 | 0.18-1.9 M/yr; (Tebo, 1991) 2 $d^{-1}$; (Yakushev et al., 2007) |
| Specific rate of Mn(IV) reduction to Mn(III) with $H_2S$ | K_mn_rd1 | $d^{-1}$ | 0.5 | 22 $d^{-1}$; (Yakushev et al., 2007) |
| Specific rate of Mn(III) oxidation to Mn(IV) with $O_2$ | K_mn_ox2 | $d^{-1}$ | 0.2 | 18 $d^{-1}$; (Yakushev et al., 2008) |
| Specific rate of Mn(III) reduction to Mn(II) with $H_2S$ | K_mn_rd2 | $d^{-1}$ | 1 | 0.96-3.6 M/yr; (Tebo, 1991) 2 $d^{-1}$; (Yakushev et al., 2007) |
| Specific rate of formation of MnS from Mn(II) and $H_2S$ | K_mns_form | $d^{-1}$ | $1*10^{-5}$ | |
| Specific rate of dissolution of MnS to Mn(II) and $H_2S$ | K_mns_diss | $d^{-1}$ | $5*10^{-4}$ | |
| Solubility product for MnS | K_mns | M | 1500 | |
| Solubility product for $MnCO_3$ | K_mnco3 | M | 1 | |
| Specific rate of $MnCO_3$ formation | K_mnco3_form | $d^{-1}$ | $3*10^{-4}$ | $10^{-4}$ – $10^{-2}$ mol/g yr; (Wersin, 1990); (Wollast, 1990) |

| | | | | |
|---|---|---|---|---|
| Specific rate of MnCO$_3$ dissolution | K_mnco3_diss | d$^{-1}$ | 7*10$^{-4}$ | 10$^{-2}$ – 10$^3$ yr$^{-1}$; (Wersin, 1990; Wollast, 1990) |
| Specific rate of MnCO3 oxidation | K_mnco3_ox | d$^{-1}$ | 27*10$^{-4}$ | |
| Specific rate of DON Oxidation with Mn(IV) | K_DON_Mn | d$^{-1}$ | 1*10$^{-3}$ | 1*10$^{-3}$ (Yakushev et al., 2007) |
| Specific rate of PON Oxidation with Mn(IV) | K_PON_Mn | d$^{-1}$ | 1*10$^{-3}$ | 1*10$^{-3}$ (Yakushev et al., 2007) |
| Threshold value of Mn(II) oxidation | s_mnox_mn2 | µM Mn | 1*10$^{-2}$ | 1*10$^{-2}$ (Yakushev et al., 2007) |
| Threshold value of Mn(III) oxidation | s_mnox_mn3 | µM Mn | 1*10$^{-2}$ | 1*10$^{-2}$ (Yakushev et al., 2007) |
| Threshold value of Mn(IV) reduction | s_mnrd_mn4 | µM Mn | 1*10$^{-2}$ | 1*10$^{-2}$ (Yakushev et al., 2007) |
| Threshold value of Mn(III) reduction | s_mnrd_mn3 | µM Mn | 1*10$^{-2}$ | 1*10$^{-2}$ (Yakushev et al., 2007) |
| Half saturation constant of Mn oxidation | K_mnox_o2 | µM O$_2$ | 2 | 2 (Yakushev et al., 2007) |
| **Iron** | | | | |
| Specific rate of Fe(II) to Fe(III) oxidation with O$_2$ | K_fe_ox1 | d$^{-1}$ | 0.5 | 2*10$^9$ M/yr; (Boudreau, 1996); 4 d$^{-1}$; (Yakushev et al., 2007) |
| Specific rate of Fe(II) to Fe(III) oxidation with MnO$_2$ | K_fe_ox2 | d$^{-1}$ | 1*10$^{-3}$ | 10$^4$–10$^8$ M/yr; (Boudreau, 1996); 1 d$^{-1}$; (Yakushev et al., 2007) |
| Specific rate of Fe(III) to Fe(II) reduction with H$_2$S | K_fe_rd | d$^{-1}$ | 0.5 | 1*10$^4$ M/yr;(Boudreau, 1996); 0.05d$^{-1}$; (Yakushev et al., 2007) |
| Solubility product for FeS | K_fes | µM | 2510 | |
| Specific rate of FeS formation from Fe(II) and H$_2$S | K_fes_form | d$^{-1}$ | 5*10$^{-4}$ | 5*10$^{-6}$–10$^{-3}$ M/yr; (Boudreau, 1996; Hunter et al., 1998); (Bektursunova and L'Heureux, 2011) |
| Specific rate of FeS dissolution to Fe(II) and H$_2$S | K_fes_diss | d$^{-1}$ | 1*10$^{-6}$ | 1*10$^{-3}$ yr$^{-1}$ (Hunter et al., 1998); (Bektursunova and L'Heureux, 2011) |
| Specific rate of FeS oxidation with O$_2$ | K_fes_ox | d$^{-1}$ | 1*10$^{-3}$ | 2*10$^7$–3*10$^5$ M/yr; (Boudreau, 1996); (Van Cappellen and Wang, 1996) |
| Specific rate of DON oxidation with Fe(III) | K_DON_fe | d$^{-1}$ | 5*10$^{-5}$ | 5*10$^{-5}$ (Yakushev et al., 2007) |
| Specific rate of PON oxidation with Fe(III) | K_PON_fe | d$^{-1}$ | 1*10$^{-5}$ | 1*10$^{-5}$ (Yakushev et al., 2007) |
| Specific rate of FeS$_2$ formation by reaction of FeS with H$_2$S | K_fes2_form | d$^{-1}$ | 1*10$^{-6}$ | 8.9*10$^{-6}$ M/day; (Rickard and Luther, 1997) |
| Specific rate of FeS$_2$ oxidation with O$_2$ | K_fes2_ox | d$^{-1}$ | 4.4*10$^{-4}$ | |
| Threshold value of Fe(II) reduction | s_feox_fe2 | µM Fe | 1*10$^{-3}$ | 1*10$^{-3}$ (Yakushev et al., 2007) |

| | | | | |
|---|---|---|---|---|
| Threshold value of Fe(III) reduction | s_ferd_fe3 | µM Fe | $1*10^{-2}$ | $1*10^{-2}$(Yakushev et al., 2007) |
| Solubility product for $FeCO_3$ | K_feco3 | $d^{-1}$ | 15 | |
| Specific rate of $FeCO_3$ dissolution | K_feco3_diss | $d^{-1}$ | $7*10^{-4}$ | $2.5*10^{-1}$–$10^{-2}$ $yr^{-1}$; (Wersin, 1990; Wollast, 1990) |
| Specific rate of $FeCO_3$ formation | K_feco3_form | $d^{-1}$ | $3.4*10^{-4}$ | $10^{-6}$–$10^{-2}$ mol/g yr; (Boudreau, 1996; Wersin, 1990; Wollast, 1990) |
| Specific rate of $FeCO_3$ oxidation with $O_2$ | K_feco3_ox | $d^{-1}$ | $2.7*10^{-3}$ | |
| **Sulfur** | | | | |
| Specific rate of $H_2S$ oxidation to $S^0$ of with $O_2$ | K_hs_ox | $d^{-1}$ | 0.5 | 0.5 (Yakushev et al., 2007) |
| Specific rate of $S^0$ oxidation of with $O_2$ | K_s0_ox | $d^{-1}$ | $2*10^{-2}$ | $2*10^{-2}$(Yakushev et al., 2007) |
| Specific rate of $S^0$ oxidation of with $NO_3$ | K_s0_no3 | $d^{-1}$ | 0.9 | 0.9 (Yakushev et al., 2007) |
| Specific rate of $S_2O_3$ oxidation with $O_2$ | K_s2o3_ox | $d^{-1}$ | $1*10^{-2}$ | $1*10^{-2}$(Yakushev et al., 2007) |
| Specific rate of $S_2O_3$ oxidation with $NO_3$ | K_s2o3_no3 | $d^{-1}$ | $1*10^{-2}$ | $1*10^{-2}$(Yakushev et al., 2007) |
| Specific rate of OM reduction with sulfate | K_so4_rd | $d^{-1}$ | $5*10^{-6}$ | $5*10^{-6}$(Yakushev et al., 2007) |
| Specific rate of OM reduction with thiosulfate | K_s2o3_rd | $d^{-1}$ | $1*10^{-3}$ | $1*10^{-3}$(Yakushev et al., 2007) |
| Specific rate of $S^0$ disproportionation | K_s0_disp | $d^{-1}$ | $1*10^{-3}$ | $1*10^{-3}$(Yakushev et al., 2007) |
| Half-saturation of Mn reduction | K_mnrd_hs | µM S | 1 | 1 (Yakushev et al., 2007) |
| Half-saturation of Fe reduction | K_ferd_hs | µM S | 1 | 1 (Yakushev et al., 2007) |

**Table 3.3. Carbon**

| Parameter | Notation | Units | Value | Reference ranges |
|---|---|---|---|---|
| Specific rate of $CaCO_3$ dissolution | K_caco3_diss | $d^{-1}$ | 3 | wide ranges are given in (Luff et al., 2001) |
| Specific rate of $CaCO_3$ formation | K_caco3_prec | $d^{-1}$ | $2*10^{-4}$ | wide ranges are given in (Luff et al., 2001) |
| Solubility product constant for CaCO3 | K_caco3 | | | Calculated as a function of T, S (Roy et al., 1993b) |
| Specific rate of $CH_4$ formation from DON | K_DON_ch4 | $d^{-1}$ | $5*10^{-5}$ | (Lopes et al., 2011) |
| Specific rate of $CH_4$ formation from PON | K_PON_ch4 | $d^{-1}$ | $1*10^{-5}$ | (Lopes et al., 2011) |
| Specific rate of $CH_4$ oxidation with $O_2$ | K_ch4_o2 | $uM^{-1}d^{-1}$ | 0.14 | 0.14 (Lopes et al., 2011) |
| Specific rate of $CH_4$ oxidation with $SO_4$ | K_ch4_so4 | $uM^{-1}d^{-1}$ | 0.0000274 | (0.0274 m3 /mol-1 day-1 (Lopes et al., 2011) |

**Table 3.4. Ecosystem parameters**

| Parameter | Notation | Units | Value | Reference ranges |
|---|---|---|---|---|
| **Bacteria** | | | | |
| Baae maximum specific growth rate | K_Baae_gro | $d^{-1}$ | $2*10^{-2}$ | $2*10^{-2}$ (Yakushev et al., 2007) |
| Baae specific rate of mortality | K_Baae_mrt | $d^{-1}$ | $5*10^{-3}$ | $5*10^{-3}$ (Yakushev et al., 2007) |
| Baae increased specific rate of mortality due to $H_2S$ | K_Baae_mrt_ h2s | $d^{-1}$ | 0.899 | 0.899 (Yakushev et al., 2007) |
| Bhae maximum specific growth rate | K_Bhae_gro | $d^{-1}$ | 0.5 | 0.5  (Yakushev et al., 2007) |
| Bhae specific rate of mortality | K_Bhae_mrt | $d^{-1}$ | $2*10^{-2}$ | $2*10^{-2}$  (Yakushev et al., 2007) |
| Bhae increased specific rate of mortality due to $H_2S$ | K_Bhae_mrt_h2s | $d^{-1}$ | 0.799 | 0.799 (Yakushev et al., 2007) |
| Baan maximum specific growth rate | K_Baan_gro | $d^{-1}$ | 0.12 | 0.12 (Yakushev et al., 2007) |
| Baan specific rate of mortality | K_Baan_mrt | $d^{-1}$ | $1.2*10^{-2}$ | $1.2*10^{-2}$ (Yakushev et al., 2007) |
| Bhan maximum specific growth rate | K_Bhan_gro | $d^{-1}$ | 0.19 | 0.19 (Yakushev et al., 2007) |
| Bhan specific rate of mortality | K_Bhan_mrt | $d^{-1}$ | $7*10^{-3}$ | $7*10^{-3}$ (Yakushev et al., 2007) |
| Bhan increased specific rate of mortality due to $O_2$ | K_Bhan_mrt_o2 | $d^{-1}$ | 0.899 | 0.899 (Yakushev et al., 2007) |
| Limiting parameter for bacteria grazing by Het | limGrazBac | - | 2 | 2 (Yakushev et al., 2007) |
| Limiting parameter for bacteria anaerobic heterotrophic | limBhan | - | 2 | 2 (Yakushev et al., 2007) |
| Limiting parameter for bacteria aerobic heterotrophic | limBhae | - | 5 | 5 (Yakushev et al., 2007) |
| Limiting parameter for bacteria anaerobic autotrophic | limBaan | - | 2 | 2 (Yakushev et al., 2007) |
| Limiting parameter for nutrient consumption by Baae | limBaae | - | 2 | 2 (Yakushev et al., 2007) |
| **Phytoplankton** | | | | |
| Maximum specific growth rate | K_phy_gro | $d^{-1}$ | 4.8 | 0.9-1.3 (Savchuk, 2002), 3.0 (Gregoire and Lacroix, 2001) |
| Optimal irradiance | Iopt | $W\ m^{-2}$ | 25 | 50 (Savchuk, 2002) |
| 1st coefficient for growth dependence on t | bm | $°C^{-1}$ | 0.12 | |
| 2d coefficient for growth dependence on t | cm | - | 1.4 | |
| Specific rate of mortality | K_phy_mrt | $d^{-1}$ | 0.15 | 0.3-0.6 (Savchuk, 2002), 0.05 (Gregoire and Lacroix, 2001) |
| Specific rate of excretion | K_phy_exc | $d^{-1}$ | 0.05 | 0.01 (Burchard et al., 2006) |
| **Heterotrophs** | | | | |
| Maximum specific rate of grazing of Het on Phy | K_het_phy_gro | $d^{-1}$ | 1.0 | 0.9 (Gregoire and Lacroix, 2001), 1.5 |

| | | | | (Burchard et al., 2006) |
|---|---|---|---|---|
| Half-saturation constant for the grazing of Het on Phy for Phy/Het ratio | K_het_phy_lim | - | 1.1 | 1 (Yakushev et al., 2007) |
| Maximum specific rate of grazing of Het on POM | K_het_pom_gro | d$^{-1}$ | 0.7 | 1.2 (Burchard et al., 2006) |
| Specific respiration rate | K_het_res | d$^{-1}$ | 0.02 | 1 (Yakushev et al., 2007) |
| Half-saturation constant for the grazing of Het on POM in dependence to ratio POM/Het | K_het_pom_lim | - | 0.2 | 0.2 (Yakushev et al., 2007) |
| Maximum specific rate of mortality of Het | K_het_mrt | d$^{-1}$ | 0.05 | 0.05(Gregoire and Lacroix, 2001) |
| Food absorbency for Heterotrophs | Uz | - | 0.5 | 0.5-0.7 (Savchuk, 2002) |
| Ratio between dissolved and particulate excretes of Heterotrophs | Hz | - | 0.5 | 0.5 (Gregoire and Lacroix, 2001) |

**Table. 3.5. Sinking**

| Parameter | Notation | Units | Value | Reference ranges |
|---|---|---|---|---|
| Rate of sinking of Phy | Vphy | m d$^{-1}$ | 1 | 0.1-0.5 (Savchuk, 2002) |
| Rate of sinking of Het | Vhet | m d$^{-1}$ | 1 | 1 (Yakushev et al., 2007) |
| Rate of sinking of bacteria (Bhae, Baae, Bhan, Baan) | Vbact | m d$^{-1}$ | 0.4 | 0.5 (Yakushev et al., 2007) |
| Rate of sinking of detritus (PON) | Vsed | m d$^{-1}$ | 6 | 0.4 (Savchuk, 2002), 1-370 (Alldredge and Gotschalk, 1988) |
| Rate of sinking of inorganic particles (Fe and Mn hydroxides, carbonates) | Vm | m d$^{-1}$ | 8 | 6-18 (Yakushev et al., 2007) |

**Table 4.  Rates of biogeochemical production/consumption of the model compartments**

**Table 4.1. Nutrients and oxygen**

| Parameter | Rate |
|---|---|
| **O$_2$** | $R\,O_2 = (GrowthPhy - RespHet - DcDM\_O2 - DcPM\_O2) * r\_o\_n - 0.25 * mn\_ox1 - 0.25 * mn\_ox2 - 0.25 * fe\_ox1 - 0.5 * hs\_ox - 0.5 * s0\_ox - 0.5 * s2o3\_ox - 0.5 * mns\_ox - 1.5 * Nitrif1 - 0.5 * Nitrif2 - 2.25 * fes\_ox - 3.5 * fes2\_ox - 0.5 * mnco3\_ox + feco3\_ox - 2 * ch4\_o2$ |

| Particulate Organic Nitrogen (PON) | R PON = MortBaae + MortBaan + MortBhae + MortBhan + MortPhy + MortHet + Grazing $*$ $(1 - Uz)$ $*$ $(1 - Hz)$ − GrazPOP) − autolysis − DcPM_O2 − DcPM_NOX − DcPM_SO4 − DcPM_Mn − DcPM_Fe − 0.5 $*$ DcPM_CH4 |
|---|---|
| Dissolved Organic Nitrogen (DON) | R DON = autolysis − DcDM_O2 − DcDM_NOX − DcDM_SO4 − DcDM_Mn − DcDM_Fe − 0.5 $*$ DcPM_CH4 − HetBhae − HetBhan + ExcrPhy + Grazing $*$ $(1 - Uz)$ $*$ Hz |
| NH$_4$ | R NH$_4$ = Dc_OM_total − Nitrif1 − anammox + 0.75 $*$ s0_ox + s2o3_ox − ChemBaae − ChemBaan + RespHet − GrowthPhy $* \dfrac{\text{LimNH4}}{\text{LimN}}$ |
| NO$_2$ | R NO$_2$ = Nitrif1 − Nitrif2 + Denitr1 − Denitr2 − anammox − GrowthPhy $* \dfrac{\text{LimNO3}}{\text{LimN}} * \dfrac{\mathbf{NO_2}}{\mathbf{NO_2 + NO_3} + 10^{-5}}$ |
| NO$_3$ | R NO$_3$ = Nitrif2 − Denitr1 − 1.6 $*$ hs_no3 − 0.75 s0_ox − s2o3_ox − GrowthPhy $* \left(\dfrac{\text{LimNO}_3}{\text{LimN}}\right) ** \left(\dfrac{\mathbf{NO_3} + 10^{-5}}{\mathbf{NO_2 + NO_3} + 10^{-5}}\right)$ |
| PO$_4$ | R PO$_4$ = $\dfrac{\text{GrowthPhy} + \text{RespHet} + \text{Dc\_OM\_total} - \text{ChemBaae} - \text{ChemBaan}}{\text{r\_n\_p}}$ + fe__p__compl + mn__p__compl |
| Si | R Si= (ExcrPhy-GrowthPhy)*r_si_n +fe_si_compl |
| Si particulate | R Si part = − K_sipart_diss $*$ **Si part** + (MortPhy + GrazPhy) $*$ r_si_n) |

Table 4.2. Redox metals and sulfur

| Parameter | Rate |
|---|---|
| **Mn(II)** | R Mn2 = mn_rd2 − mn_ox1 + mns_diss − mns_form − mnco3_form + mnco3_diss + 0.5 $*$ fe_ox2 + (DcDM_Mn + DcPM_Mn) $*$ r_mn_n |
| **Mn(III)** | R Mn3 = mn_ox1 − mn_ox2 + mn_rd1 − mn_rd2 |
| **Mn(IV)** | RMn4 = mn_ox2 − mn_rd1 − 0.5 $*$ fe_ox2 + mnco3_ox − (DcDM_Mn + DcPM_Mn) $*$ r_mn_n |
| **MnS** | R MnS = mns_form − mns_diss |
| **MnCO$_3$** | R MnCO$_3$ = mnco3_form − mnco3_diss − mnco3_ox |
| **Fe(II)** | R Fe2 = fe_rd − fes_form − fe_ox1 − fe_ox2 + fes_diss − feco3_form + feco3_diss + fes2_ox + 4 $*$ r_fe_n $*$ (DcDM_Fe + DcPM_Fe) |

| | |
|---|---|
| **Fe(III)** | $R\ Fe3\ =\ fe\_ox1 + fe\_ox2 - fe\_rd + fes\_ox + feco3\_ox - 4 * r\_fe\_n * (DcDM\_Fe + DcPM\_Fe)$ |
| **FeS** | $R\ FeS\ =\ fes\_form - fes\_diss - fes\_ox - fes2\_form$ |
| **FeS$_2$** | $R\ FeS_2\ =\ fes2\_form - fes2\_ox$ |
| **FeCO$_3$** | $R\ FeCO_3\ =\ feco3\_form - feco3\_diss - feco3\_ox$ |
| **H$_2$S** | $R\ H_2S\ =\ 0.5 * s0\_disp\ - hs\_no3 + s2o3\_rd - fes2\_form - 0.5 * mn\_rd1 - 0.5 * mn\_rd2 - 0.5 * fe\_rd - hs\_ox + fes\_diss - fes\_form$ $+ mns\_diss - mns\_form$ |
| **S$^0$** | $R\ S^0\ =\ hs\_ox + 0.5 * mn\_rd1 + 0.5 * mn\_rd2 + 0.5 * fe\_rd - s0\_ox - s0\_disp - s0\_no3$ |
| **S$_2$O$_3$** | $R\ S_2O_3\ =\ 0.5 * s0\_ox - s2o3\_ox + 0.25 * s0\_disp\ + 0.5 * so4\_rd - 0.5 * s2o3\_rd - s2o3\_no$ |
| **SO$_4$** | $R\ SO_4\ =\ hs\_no3 - so4\_rd + 0.5 * s2o3\_ox + s0\_no3 + 2 * s2o3\_no3 + fes\_ox + 2 * fes2\_ox$ |

**Table 4.3. Carbon and alkalinity**

| Parameter | Rate |
|---|---|
| **DIC** | $R\ DIC = caco3\_diss - caco3\_form\ - mnco3\_form + mnco3\_diss + mnco3\_ox - feco3\_form + feco3\_diss + feco3\_ox$ $+ (Dc\_OM\_total - ChemBaae - ChemBaan - GrowthPhy + RespHet) * r\_c\_n$ |
| **CaCO$_3$** | $R\ CaCO3 =\ caco3\_form - caco3\_diss$ |
| **CH$_4$** | $R\ CH4\ =\ ch4\_form - ch4\_ox$ |
| **Total alkalinity** | $R\ Alk =\ dAlk$ |

**Table 4.4. Ecosystem parameters**

| Parameter | Rate |
|---|---|
| **Phytoplankton** | $R\ Phy\ =\ GrowthPhy - MortPhy - ExcrPhy - GrazPhy$ |
| **Heterotrophs** | $R\ Het\ =\ Uz * Grazing - MortHet - RespHet$ |
| **Aerobic heterotrophic bact.** | $R\ Bhae\ =\ HetBhae - MortBhae\ - GrazBhae$ |

| Aerobic autotrophic bact. | $R_{Baae} = Chem_{Baae} - Mort_{Baae} - Graz_{Baae}$ |
|---|---|
| Anaerobic heterotrophic bact. | $R_{Bhan} = Het_{Bhan} - Mort_{Bhan} - Graz_{Bhan}$ |
| Anaerobic autotrophic bact. | $R_{Baan} = Chem_{Baan} - Mort_{Baan} - Graz_{Baan}$ |

.

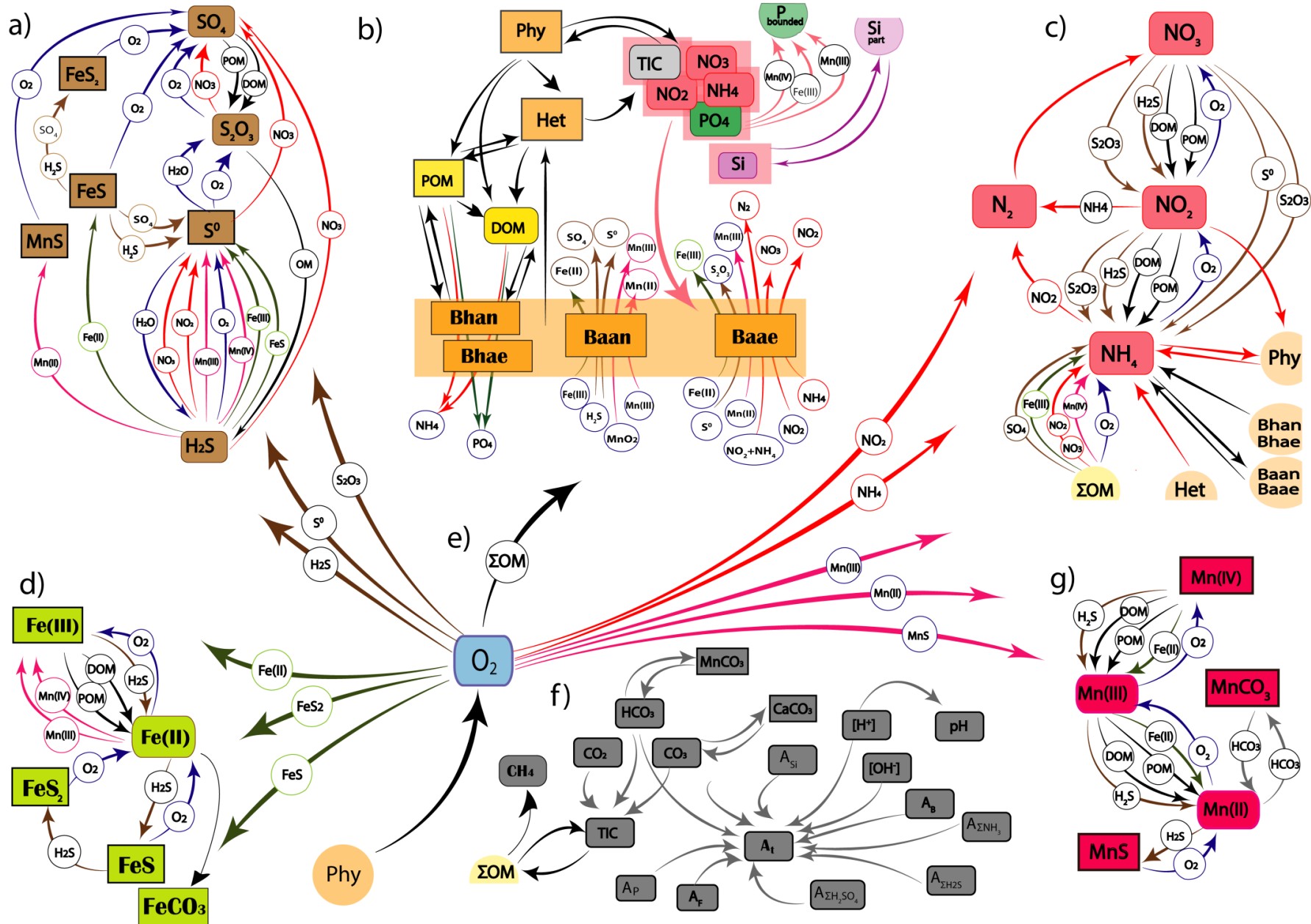

**Figure 1. Flow-chart of biogeochemical processes represented in the Benthic RedOx Model (BROM), showing the transformation of sulphur species (a), the ecological block (b), the transformation of nitrogen species (c), the transformation of iron species (d), the processes affecting dissolved oxygen (e), the carbonate system and alkalinity (f), and the transformation of manganese species (g).**

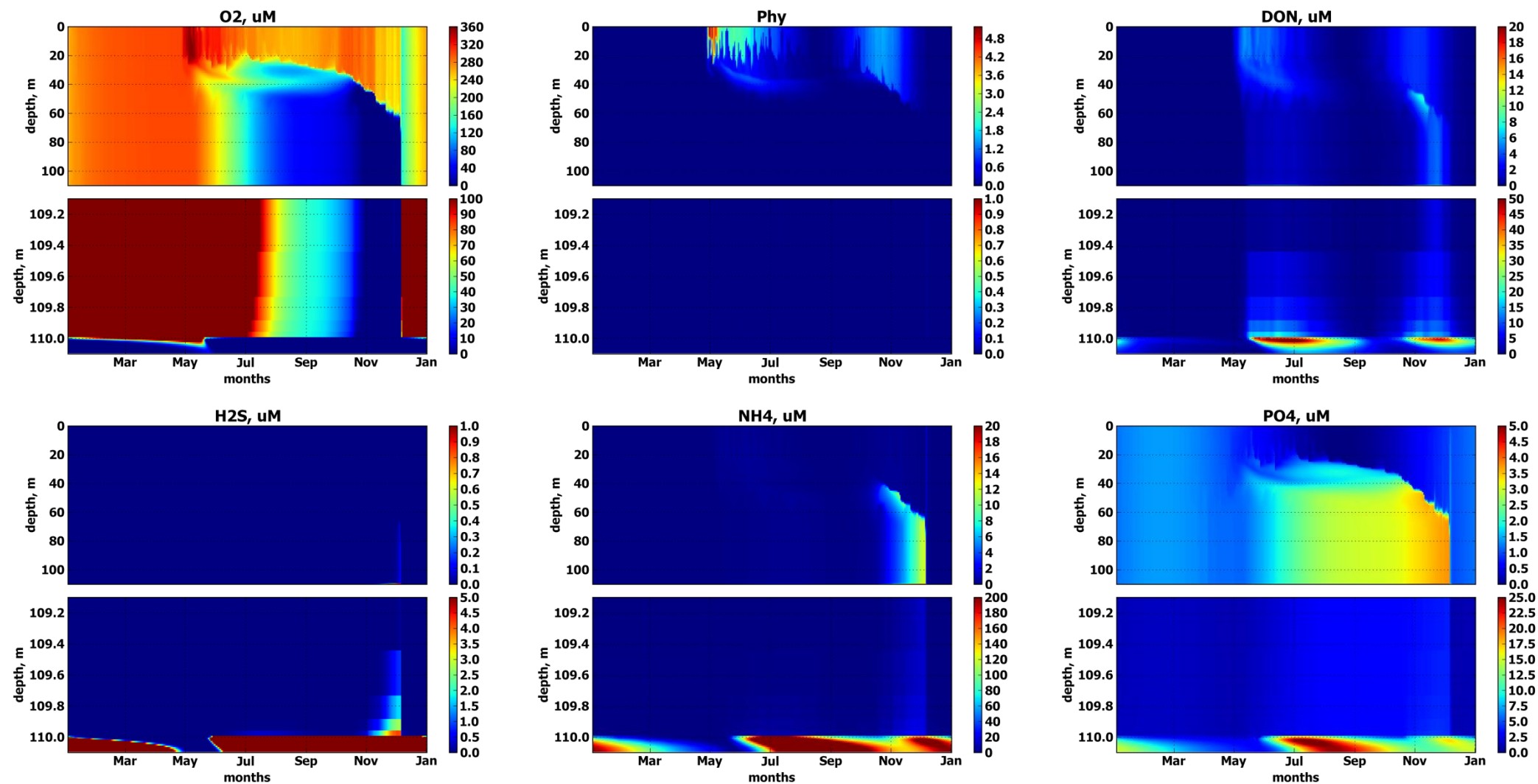

**Figure 2. Simulated seasonal variability of the selected modelled chemical parameters (μM), in the water column (top panels) and in the benthic boundary layer and sediments (bottom panels).**

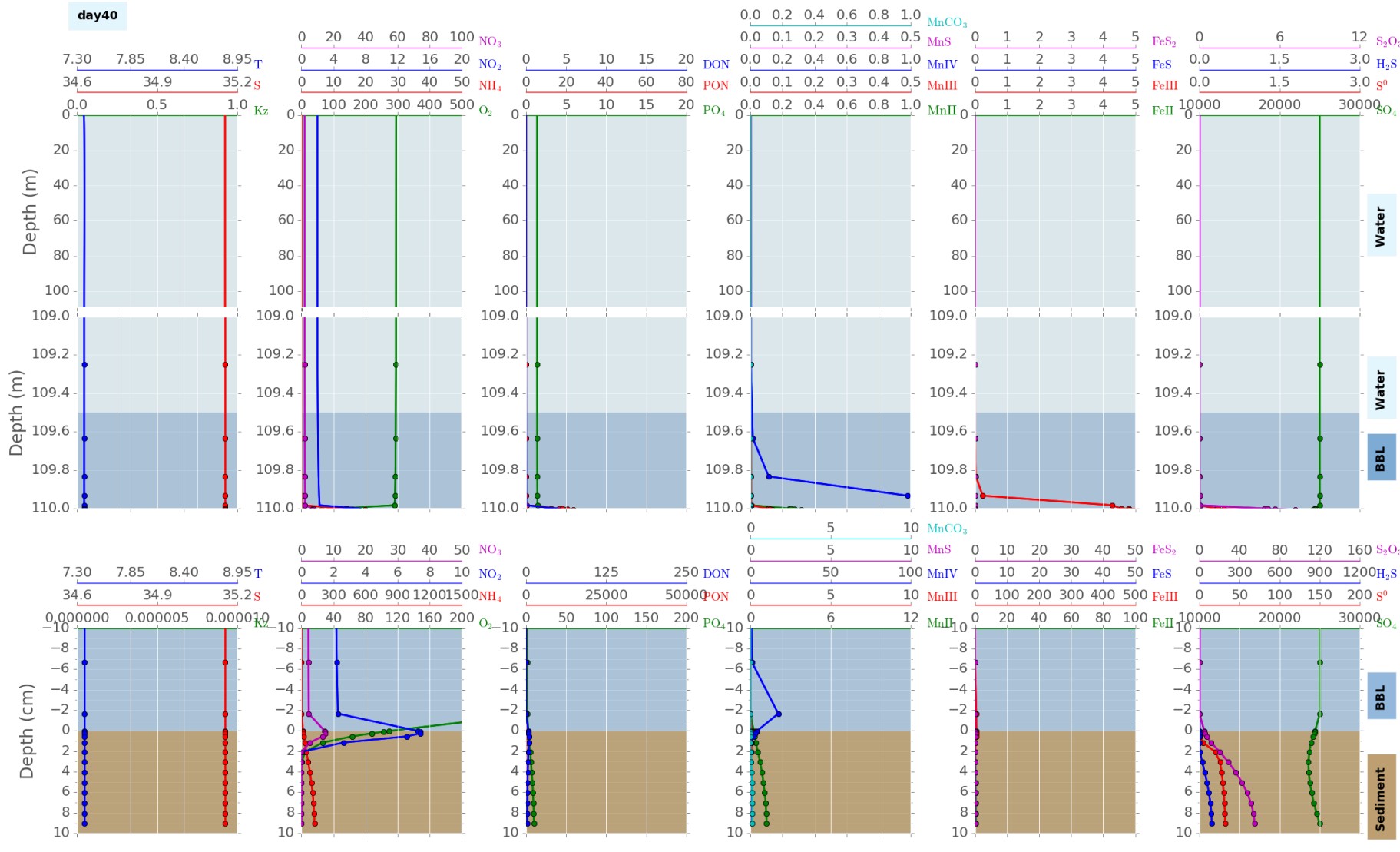

**Figure 3. Vertical distributions of the modelled chemical parameters (µM), biological parameters (µM N), temperature ($^o$C), salinity (PSU) and vertical diffusivity ($10^{-3}$m$^2$s$^{-1}$) during the winter period of well-mixed conditions, showing the water column (light blue), the benthic boundary layer (dark blue), and the sediments (light brown).**

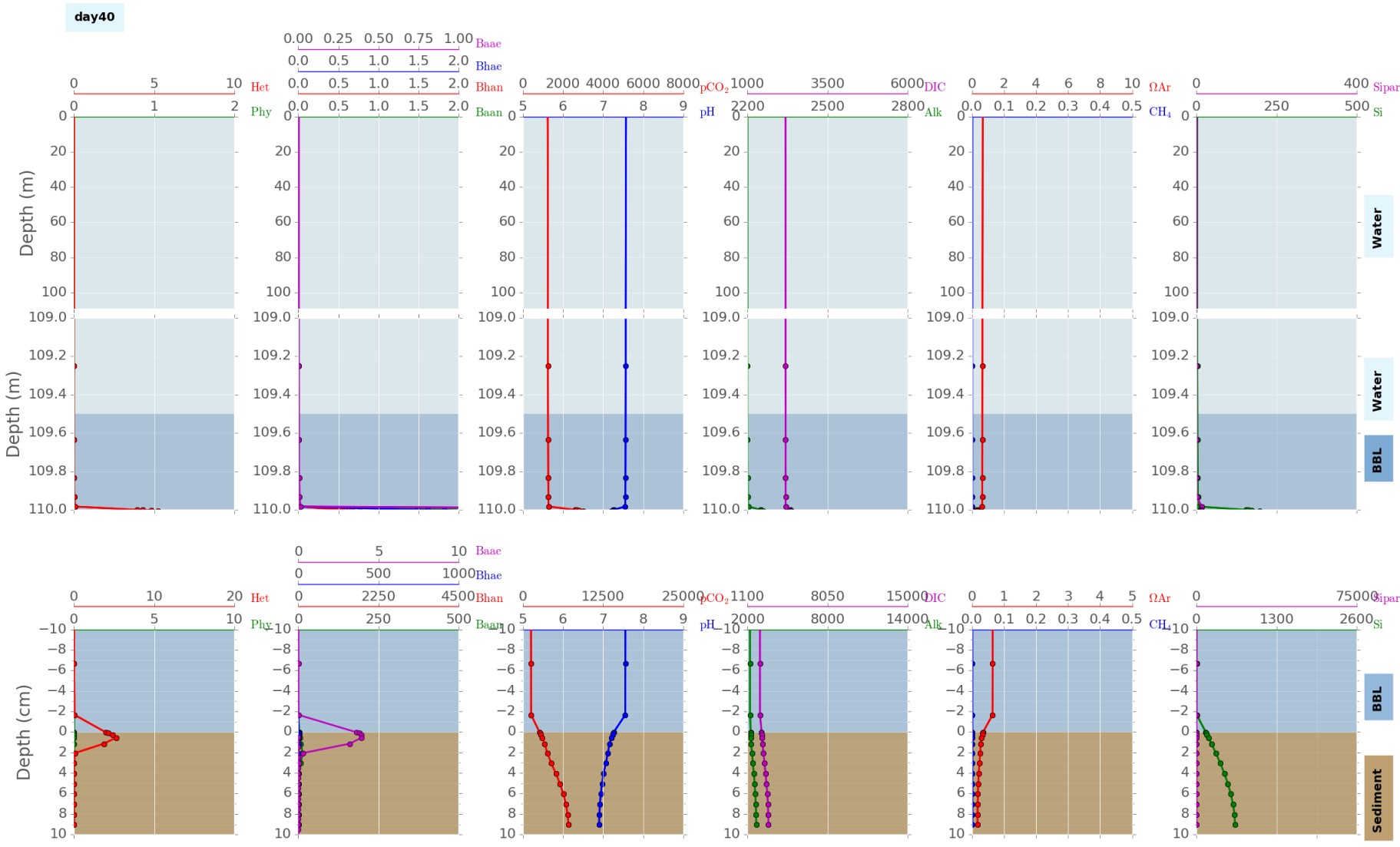

**Figure 3 contd. Vertical distributions of the modelled chemical parameters (μM) and biological parameters (μM N) during the winter period of well-mixed conditions, showing the water column (light blue), the benthic boundary layer (dark blue), and the sediments (light brown).**

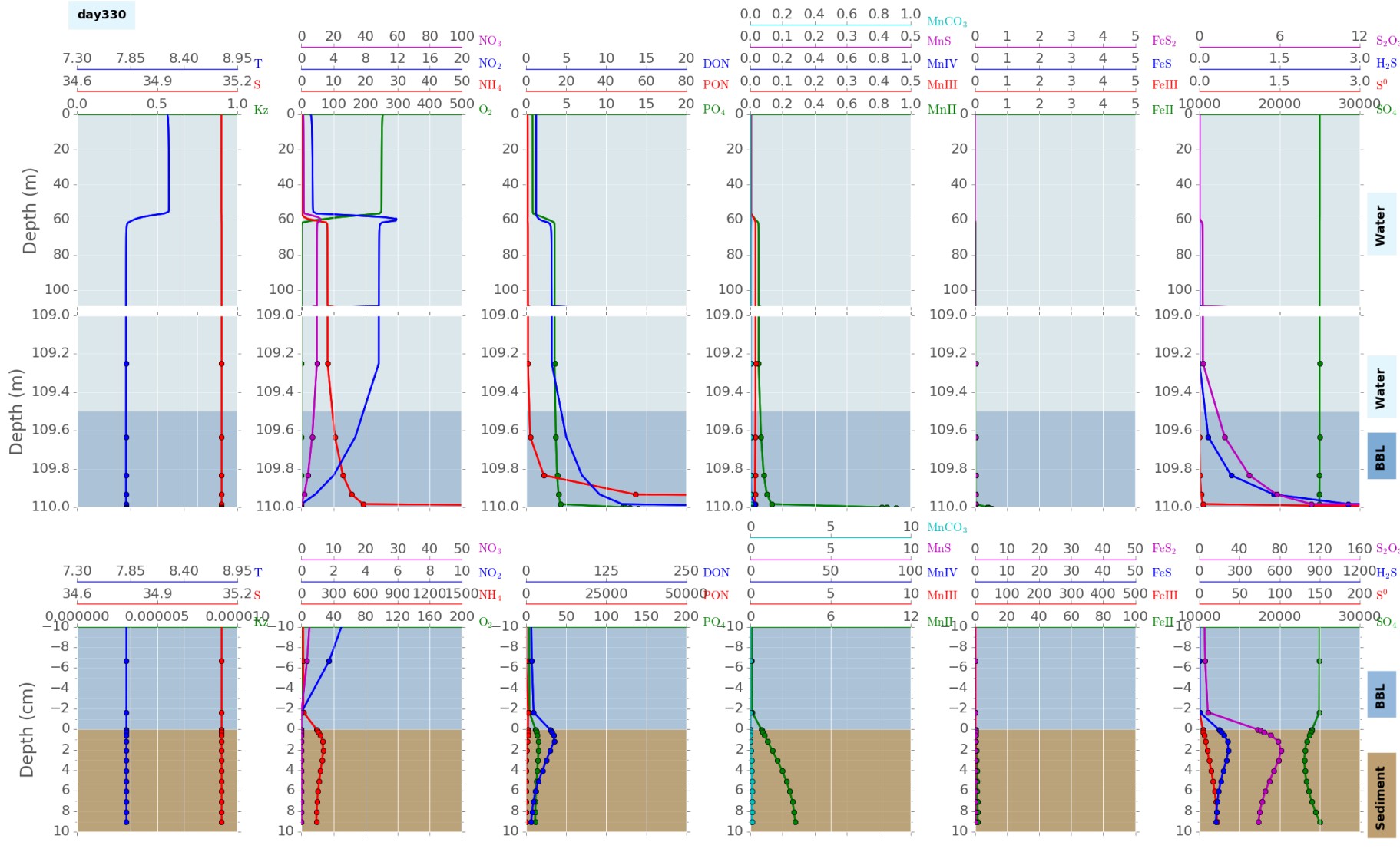

**Figure 4. Vertical distributions of the modelled chemical parameters (µM), biological parameters (µM N), temperature ($^o$C), salinity (PSU) and vertical diffusivity ($10^{-3}$m$^2$s$^{-1}$) during the period of bottom anoxia, showing the water column (light blue), the benthic boundary layer (dark blue), and the sediments (light brown).**

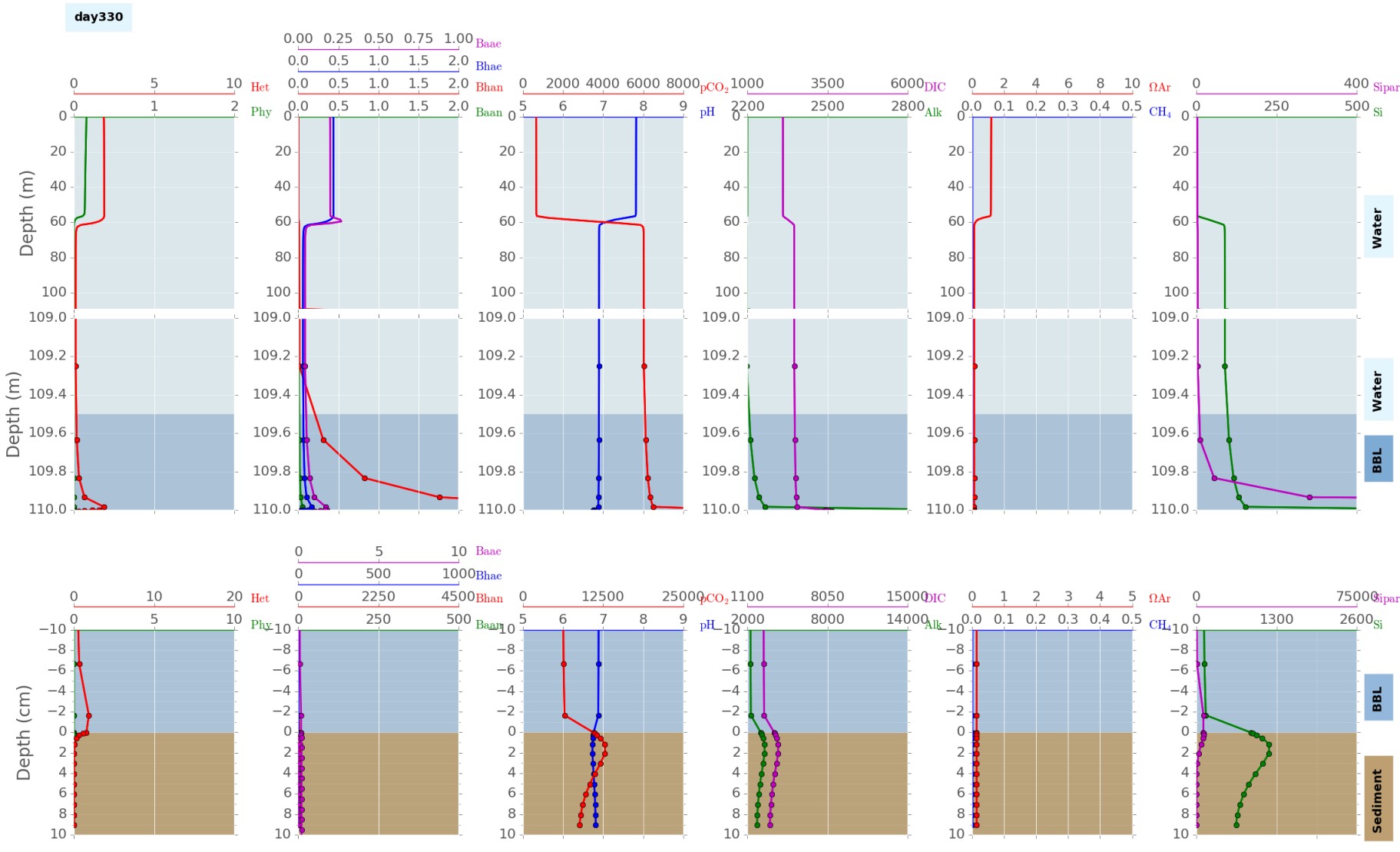

**Figure 4 contd. Vertical distributions of the modelled chemical parameters (µM) and biological parameters (µM N) during the period of bottom anoxia, showing the water column (light blue), the benthic boundary layer (dark blue), and the sediments (light brown).**

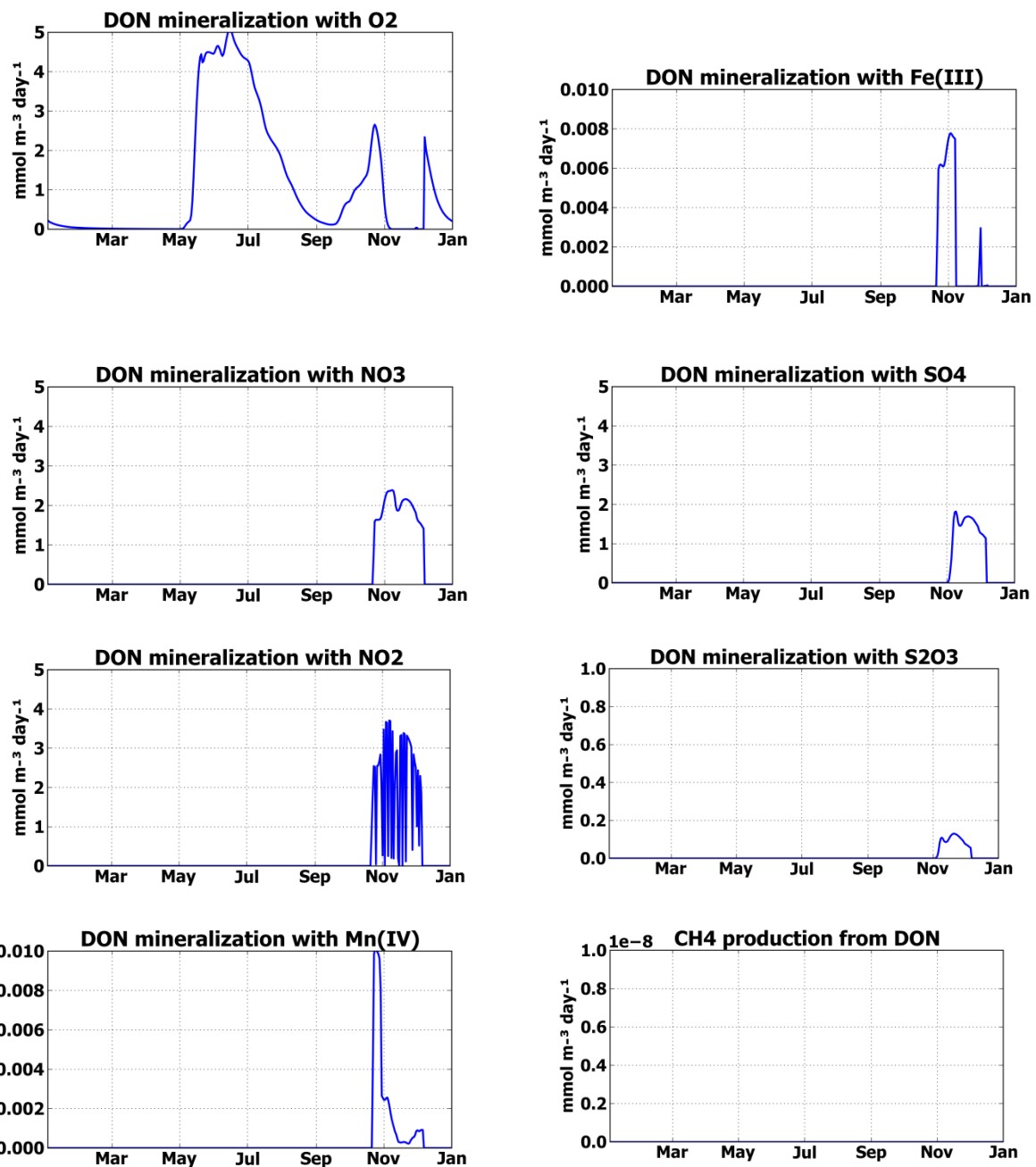

**Figure 5. Simulated seasonal variability of biogeochemical transformation rates just above the sediment water interface, showing the rates of DON mineralization with oxygen, nitrate, nitrite, Mn(IV), Fe(III), SO4, S2O3, and CH4 production from DON. Units are mmol m$^{-3}$ d$^{-1}$.**

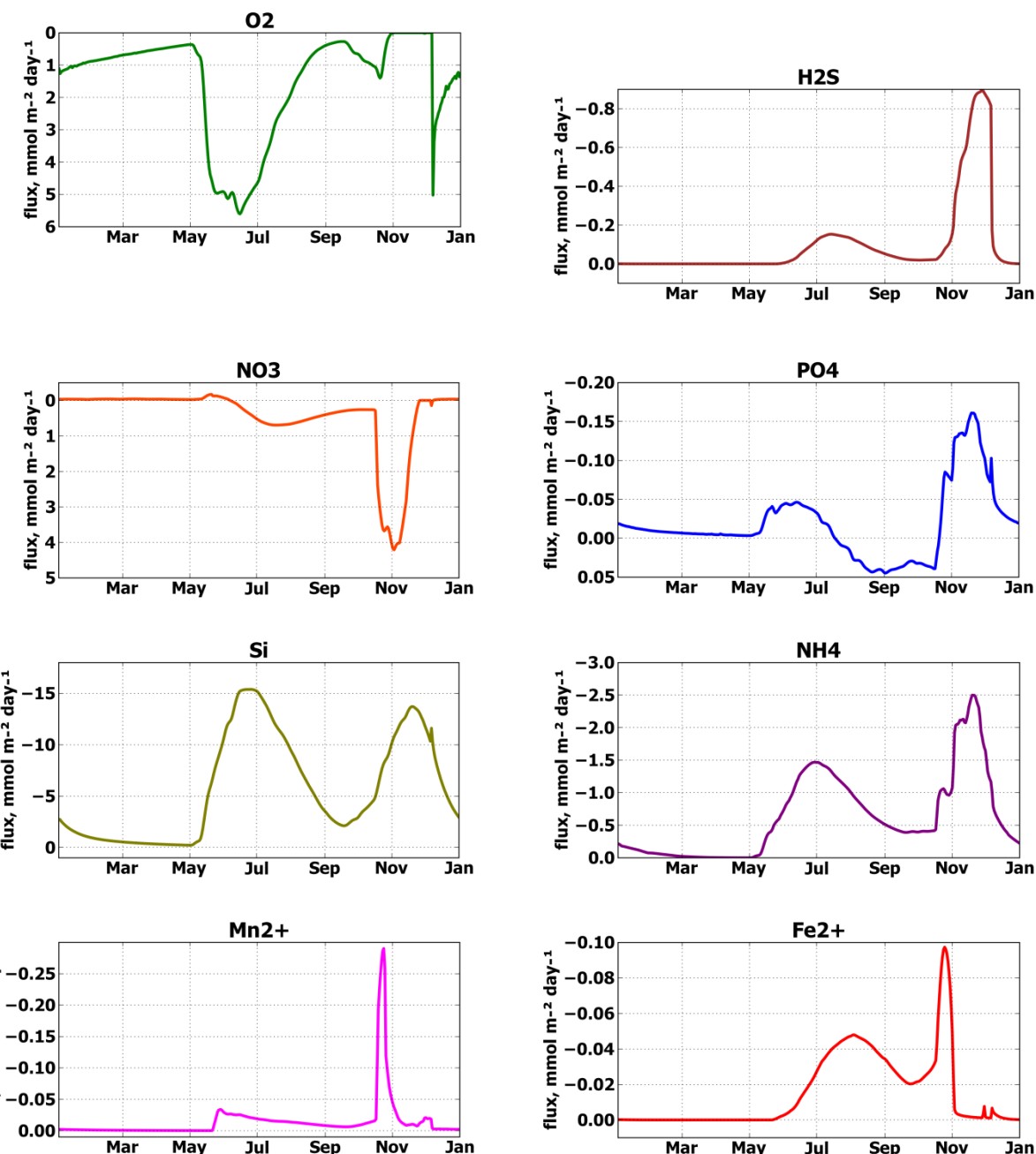

**Figure 6. Simulated seasonal variability of vertical diffusive fluxes from the benthic boundary layer to the sediments of oxygen, hydrogen sulphide, nitrate, silicate, ammonia, Mn(II) and Fe(II). Positive fluxes are downward and negative fluxes are upward. Units are mmol m$^{-2}$ d$^{-1}$.**

