# Peer review of "Bottom RedOx Model (BROM, v.1.1): a coupled benthic-pelagic model for simulation of water and sediment biogeochemistry"

_Geoscientific Model Development, 2015_

## Referee Comment (RC1) · J. Middelburg (Referee) · 10 Feb 2016

The authors present a rather complex model allowing simulation of biogeochemical processes in coastal systems subject to seasonal anoxia. The paper has a few strengths and many weaknesses. Most numerical biogeochemical models focus either on the water column or on the sediments and very few couple these domains. The presented BROM model does explicitly deal with the coupled system and is therefore of value.

Strengths of this paper include the (a) coupling of pelagic and benthic modules, (b) the explicit resolution of the benthic boundary layer (BBL), (c) the focus on seasonal hypoxia and (d) its linkage to/integration with the Framework for Aquatic Biogeochemi-

cal Modeling. These characteristics make this an interesting paper and the presented model is potentially useful.

However, there are many issues to be resolved before publication of this paper and model. 1. The paper is poorly written in terms of organization, flow and use of English. A few examples of the latter (line 1: seawater and benthic sediments, benthic systems or sediments are fine, but not benthic sediments; the use of the term protolithic processes: do you refer to stone age processes? or do you simply mean equilibrium processes, etc. etc.). There appear native speakers and/or UK/Canada based scientists among the authors: perhaps they should have another look at it. The text is also not prepared with utmost care: many typos, wrong equations etc. (see below in the list of detailed comments).

2. The model could be much better put in context and existing literature is poorly incorporated. I was missing references to other papers dealing with coupled benthic-pelagic models, the more simpler ones (Lancelot, Soetaert, Fennel, Reed and co-workers) and the highly complex ones from the ERSEM family. Soetaert and co-workers (Soetaert and Middelburg, 2009; Meire et al., 2013) have published on seasonal oxygen issues with coupled pelagic benthic models. There is also a large body of knowledge on the effect of oxygen in early diagenetic models; that literature is not covered. Extensive work on the role of sediments as moderating the timing of return fluxes (delay in return of N, P, Si after bloom) and the memory provided by stored reduced sulfur, iron and carbon in sediments is poorly covered (see work on Gulf of Mexico by Nancy Rabalais and co-workers).

3. The model description is often incomplete and imprecise (see below for a far from complete list). The documentation is not sufficient. Boundary conditions of the model are not clearly described. Details about the coupling of the models are insufficient: e.g. the grid size is very likely changing, yet not provided. It is unclear whether particulate organic matter is modeled explicitly. Is it transported by bioturbation in the sediment. It is unclear how bio-irrigation and solid-phase mixing are treated. Sometimes parame-
ters are introduced in the text, but appear as fixed value (hard-coded) in the tables.

4. The model is very complex and detailed (perhaps too much) in some aspects and very rudimentary in other aspects. Regarding the latter, many detailed Mn, Fe, S transformations are included, chemoautotrophy is resolved for aerobic and anaerobic microbes (or only bacteria?), but important processes such as methane generation and anaerobic methane oxidation coupled to sulfate reduction are ignored. Another example particle settling velocities is corrected for the formation of Mn-oxides in the water column but other carrier phases such as calcium carbonate are not resolved. Clearly, the presented model is a version 1.0 and represents a first step, but the priorities of the authors do not match those of the majority of the audience. At the minimum some motivation for their particular choice should be communicated to the audience.

5. Section 3.1 on model output discussion needs major revision. The link with the figures is unclear and the organization is suboptimal. You discuss the oxygenated winter period and then link to later periods or a few days later in the section on oxygenated winter period. There is no story. Try to limit yourself to a few findings and discuss those. The reader now has to digest all the computer output her or himself.

6. Section 3.2 is not useful or convincing. The link with data is very poor. This is indeed a difficult job, but here serious work has been done. A comparison with just three to four papers is made and the extensive databases on oxygen uptake, oxygen penetration depth etc are not consulted. A statement like line 10 on page 20: "further observations under anoxic and suboxic conditions are rare as field and experimental studies generally focus on oxic conditions" is close to unacceptable. Consider all the work done on the Eastern Pacific ocean margins (Washington and Oregon shelves, California basins, Mexico/Peru/Chile margin), Indian ocean and shelves and multiple European systems (Black Sea, Baltic Seas). There are nice seasonal time series in coastal systems with low-oxygen events during summer (Kiel bight, etc.)

Minor issues:

Page 3: - Line 2: benthic sediments? - Line 3: directly affect and are impacted by.. - Line 7: fuzzy writing, rewrite

Page 4: - Line 3: enrichment with OM or do you mean deposition of labile organic matter - Line 26-27: animals can die, migrate or change their behavior: revise text accordingly

Page 5: - Line 4-15: additional literature incorporation required. How does your model differ from those. Where are the improvements etc. - Line 22: the benthic boundary layer is a major strength. Introduce it better. Delineate the features, etc. - Line 24: at the BBL: do you exclude the sediments here?

Page 6: - line 9-10: it is unclear whether organic C is also modeled or is it just inorganic C. - Line 12: No nitrogen transformations?

Page 7: - Line 2: delete consists - Line 26 and all through: replace protolithic processes with equilibrium processes/reactions or acid-base reactions

Page 8: - Line 11: provide the number of state variables to the reader

Page 9: - Line 1: chemoautotrophy is resolved, but overall secondary production is ignored. There may be good reasons for this, but communicate this then to the reader - Line 8-10: why is methanogenesis excluded? This is probably related to the way you model organic matter. Conceptually most simple is to turn all labile organic matter remaining after depletion of all oxidants into methane and carbon dioxide. - Line 15-25: the alkalinity equations as given are wrong: the phosphate alkalinity term should include a $H_3PO_4$ term, the ammonium alkalinity term should not include $NH_4^+$, etc. Please check whether you have implemented it correctly into your model. Page 10: -Line 2: I guess you mean Atom was set to zero and not TOM.

Page 11: -Line 1-10: quite a number of the reactions are not balanced and inconsistent with Table 2: e.g. denitrification with hydrogen sulfide and the line above represent two reactions of which the latter misses a two before OH-. Check carefully. -Line 11: Table

2 not 3.

Page 12: middle of page: a distinction is made between settling velocity of particulate matter and of particles with Fe and Mn oxides. Why not write particle settling velocities (w) as a sum of various contributing terms. Why the focus on Fe and Mn? Just a Black Sea model heritage?

Page 13: - Line 8: the eddy diffusion coefficient was assumed constant in the BBL. As a first coupled model that resolves the BBL it may be done like this, but given the lognormal velocity profiles, would one not expect a depth profile in Kz as well. This can be incorporated quite easily. - Line 15-20: the description of bioturbation/bioirrigation is difficult to follow. As written above are solid phase and solutes transported separately or together? This is unclear. Bioturbation depth are very shallow for fully oxic conditions.

Section 2.3. The description of boundary conditions needs more attention. It appears that you use flux boundary conditions for the gases and constant or fixed (at least imposed) boundary for the others. This may lead to mass balance issues. The boundary conditions at the bottom (depth 12 cm in sediments) are not described: no flux or no gradient or fixed concentration? The assumed sulfate concentration is either close to zero or do you mean 25 10-3M? Basically all external sources such as atmosphere and rivers are added to surface layer.

Section 3: I stop making detailed feedback because there are too many issues and the referee already spent double the amount of time normally needed for an evaluation.

p. 26, line 5-9: chemoautotrophy indeed involves CO2 consumption and thus has the potential to increase pH. However, the energy required for CO2 fixation is obtained from oxidation of reduced products: usually an acid producing process. With typical growth efficiencies one would produce more acid linked for the energy than consumption of acid by organic matter production. Cable-bacteria spatially disconnect half reactions and can therefore cause a real pH increase. Without detailed model investigations, I

[Figure]

suggest removing these sentences. The authors might be right because of the complexity of reactions and the many buffering reactions, but it is not convincing as presented here.

Table 1: it is stated that oxygen is presented in microM O, but sometimes it might be, at other places it is definitely in microM O2.

Table 2: - Aerobic respiration and denitrification are treated different than Fe, Mn and sulfate reduction regarding DON and PON separation. - For Mn reduction where does the 0.5 come from (half saturation constant hard-coded?) - There are multiple typos which complicate checking. - Where is the (1+ftD(t)) term coming from. ftD is not defined. - Page 41: I guess that NO3 dependence should depend on nitrate and not on ammonia?

Table 3: - I guess that K_Mn_rds should be K_Mnrd_HS? - There are many values assumed, some literature citations would be helpful. I guess that the model is rather insensitive to most of these parameters and their value should therefore be based on literature values. - Why did you choose 2.7 for the Fe/P ratio and not the conventional 10?

Table 4: check carefully: e.g. for phosphate you have hard-coded 2.7 for Fe/P and 0.67 for Mn/P rather than a parameter. Taking stoichiometry as a constant is fine, but do not present

Table 5: this table is unuseful and I doubt whether the fluxes are all in the right units.

Table 6: could be deleted.

Figures. All captions need more documentation. For instance it is not even mentioned why some concentrations are presented on two different scales. As written above, reconsider to focus on a few results and elaborate the model results in another paper. The figures as presented now appear more like raw model output.

---

## Referee Comment (RC2) · O.P. Savchuk (Referee) · 23 Feb 2016

Simulation of alternating oxic/anoxic conditions in coastal ecosystems on the fine spatio-temporal scales is useful for studies of specific questions, from an explicit description of the bottom boundary layer to a succession/alteration of multiple electron donor/acceptor agents to details of alkalinity composition and effects on the carbonate system, etc. Therefore the manuscript could be interesting to a wider audience and published also in the main body of Geoscientific Model Development papers. In that case, the manuscript demands a major revision, because both the form and content are rather sloppily observed and prepared. Many of specific issues and details of such revision have already been indicated by the first reviewer, Prof. J. Middelburg. I concur

with almost all of them.

However, while trying to further expand the list of questions, suggestions, and requests, I got substantial doubts in the suitability of this specific manuscript for this particular journal, based on the following:

1. Categorization of this manuscript as a "model description paper" requires a comprehensive model description, which internal consistency is verified by demonstration of its capacities, rather than a detailed validation of its implementation as would be expected from a "model evaluation paper". The ambiguity of the paper's goals is reflected in repeating expressions like "to develop a model AND analyse seasonal effects". As it looks now, the manuscript describes a specific model implemented for studies of some particular biogeochemical questions rather than presents some finished single product that can be relatively straightforwardly borrowed and used by interested colleagues.

2. Such ambiguity starts already from rather inconsistent definition of objectives. The title announces "coupled benthic-pelagic model for simulation of seasonal anoxia", the abstract indicates the goal as a capturing of "biogeochemical processes occurring at the bottom boundary layer (BBL) AND sediment-water interface (SWI)", the last sentence of "Background" Section indicates the goal as a capturing of "key biogeochemical processes occurring at the bottom boundary layer" only. Even farther, "the main goal of the model was to reproduce the biogeochemical mechanism of transformation of oxic conditions into anoxic in the sediment–water interface". Perhaps, such obscurity reflects also a story of development of BROM from ROLM by substantially expanding list of variables and their interactions. If, as it seems to me, the real focus and achievements lay in the "middle", then almost a sole goal of the water column and sediment parts is to generate consistent boundary conditions for interacting BBL and SWI. From the manuscript it is also unclear, why the focus is on seasonal dynamics and what prevents the reproduction of sporadic short-term alterations or long-term persisting states.

3. Then, for a further implementation in diverse geographical areas it should be

**[GMDD](https://www.geosci-model-dev-discuss.net/)**
stressed and clearly explained, where from should the user obtain the data about external inputs, internal dynamics and distribution on multiple forms of sulfur, manganese, iron, as well as on different functional groups of bacteria. At the least, recommendations should be given on some proxies that could be derived from the pelagic ecosystem models with less uncommon sets of variables and processes.

4. Furthermore, there are several ad hoc features and patches pertaining, perhaps, only for this implementation that should be explicitly indicated for a prospective users, for instance, holding sea surface concentrations constant results in non-conservation; prescription constant coefficient of vertical transport in BBL, while arbitrarily modifying it by assumed bioturbation in the sediments; extensive use of squared availabilities (Nutrient/Biomass)ˆ2 instead of concentrations N in nutrient limitation and trophic functions.

Fortunately, selected results, ideas and formulations can still be gratefully borrowed by interested colleagues with appropriate reference to the ever available discussion paper.

---

## Referee Comment (RC3) · G. Munhoven (Referee) · 4 Mar 2016

**1 General comments**

**1.1 Appreciation of the manuscript**

In this paper, E. V. Yakushev and co-authors present a highly complex model suitable to study the coupled biogeochemical processes at the bottom boundary layer, the sediment-water interface and the surface sediment. The model appears to provide an extremely complete description, considering all the processes and primary and

secondary chemical reactions that have been taken into account. It is integrated into the *Framework for Aquatic Biogeochemical Models*, FABM (Bruggeman and Bolding, 2014).

Although the model appears to have been skilfully designed and set up, the paper has, unfortunately, a number of weaknesses. It is not suitable for publication in *Geoscientific Model Development* in its current form – it should nevertheless be possible to **reconsider it after a major revision**.

This paper would definitely have benefited from another round or two of rereading and proofreading. Not even the name of the model is unambiguously given: in the title, the name is *Bottom RedOx Model*, in the model presentation (p. 2, ll. 2–3) it is *Bottom RedOx Layer Model*. The English of the paper needs some thorough revision. There are parts that are acceptable and others that are almost unsuitable for review. I am not going to point out all the English errors that I found – they are simply to numerous to key them all in here. There is one British co-author and two co-authors with affiliations to institutions in English-speaking countries or regions: could they please have a look at the manuscript and help to correct it and rewrite where necessary! There are errors (spelling, grammar, syntax, style) on nearly every single page, but sections 3.2.4 (Manganese) and 3.3 (Carbonate system) require particularly close attention.

The paper has been submitted as a "model description paper". Requirements for that type of paper are detailed in http://www.geoscientific-model-development.net/about/manuscript_types.html#item1. Quite some requirements are not met in this paper.

1. The model description is not well contextualized. The application presented is for a shallow-water environment, but one may ask where else it could be applicable, and which extensions or adaptations would be required or which simplifications would be possible. The authors mention, e. g., a possible coupling to NEMO,

which encompasses almost the complete range of marine environments that one can imagine.

2. The technical details of the implementation are incomplete, and therefore, the criterion of model reproducibility that the paper should aim for is not met. All to many details are not covered in the description.

3. The instructions about where to get the code are incomplete. Much guesswork is currently required to locate the relevant files inside the FABM distribution. This could easily be avoided by, say, three to five extra sentences.

4. There does not seem to exist a way to permanently access the precise model version described in this paper.

5. The limitations of the model and the fundamental software requirements are not given: if the model described here is really BROM-transport (this is not a name found in the paper, but it is the name of the only sensible source code collection that I could find), then the paper needs to state right away that:

   - the BROM source code can only be compiled with the Intel Fortran compiler for Windows
   - the current version can only use hydrodynamic conditions derived from GOTM (according to the Wiki at https://sourceforge.net/p/fabm/wiki/BROM_ FABM).

   Although it is reported on p. 16 (l. 5), that the model was run with the Intel compiler for Windows,[1] it is said nowhere that this is the only way to run it. This is obviously a extremely strongly limitation and I am wondering whether such a restriction is
* * *
[1]It would be useful to provide the version number of the compiler used. FABM and BROM require some specific Fortran 2003 features and the Intel compiler only offers full support for Fortran 2003 since version 15. However, a subset offered by earlier versions might be sufficient here.

fundamentally necessary. As far as I can see, FABM itself is written in standard-conforming Fortran 2003 in a portable manner (no hardcoded kind types, etc.) and does not seem to rely on a single compiler for a single platform.

I strongly encourage the authors to prepare a version of the source code that can be used on other platforms and with alternative compilers. It should be possible to do this quite rapidly by introducing a few pre-processor directives, which would switch off some extra functionality provided by the Intel Fortran compiler for Windows, but that is not fundamentally required for the model itself. This would increase the usefulness of BROM by orders of magnitude! Else, what is the point in emphasizing that the model code "[. . . ] uses modern software standards: it is coded in object-oriented Fortran 2003, [. . . ]" (p. 27, ll. 17–18) if in the end, it only compiles with one single compiler on one single platform.

The model itself seems to be carefully designed and set up. There are a few assumptions regarding the physical environment that may be debatable and that would benefit from a few extra words of explanation (see specific comments below). The set of processes and coupled chemical reactions and equilibria that have been taken into account is extremely complex. It is not obvious if such a high degree of complexity is truly necessary. The model indeed seems to allow a rather accurate simulation of the environment chosen. However, to what extent does it contribute to improve our understanding of the way the environment evolves? It would be interesting to know which are the dominant actors of the system. Unfortunately, the paper does not address this kind of question at all.

**2 Specific comments**

**2.1 Introduction**

The scope of the model, i. e., the bottom boundary layer (BBL), (also known as the benthic boundary layer, or are there differences between those two BBLs?) deserves to be presented in more detail. What is its typical thickness? What influences that thickness? How does it change throughout the global ocean? What are the typical gradients across the BBL? Please do not forget that *Geoscientific Model Development* has a broad lectureship.

I am surprised to read that "the Bottom Boundary Layer (BBL) [. . . ] is still understudied" (p. 5, l. 22). On my shelf I have the fine book *The Benthic Boundary Layer: Transport Processes and Biogeochemistry* (Boudreau and Jørgensen, 2001). It is nearly 15 years old and BBL research has certainly not come to a rest since that book got published. Please reconsider that statement and provide a fair representation of the existing literature.

**2.2 General model presentation**

**2.2.1 Scope of the model**

In the end, it is not entirely clear what the exact scope of BROM is. In the abstract, BROM is introduced as a model for the biogeochemical process in the bottom boundary layer; in the model description though, we read that "[t]he water column considered in our model spans the sea surface (upper boundary) down to user's defined sediment depth [. . . ]". This is to some extent contradicting as this domain largely exceeds the bottom boundary layer. Please clarify.

**2.2.2 Computational aspects**

It is stated that numerical integration was carried out with the Eulerian scheme (the explicit or the implicit variant? – the extremely short time-steps chosen make me guess it is the former, but it would be good to state this). Is the same Eulerian scheme used for both space and time dimensions? Please specify all the schemes used.

Details about the pH solving algorithm can only be looked up in the code. The text only says that "[. . . ] total pH was calculated using the Newton-Raphson method" (p. 11, ll. 20–21) and that "Carbonate system equilibration was parameterized using the standard approach (i.e. Lewis and Wallace, 1998)" (Table 2). This latter affirmation is meaningless: Lewis and Wallace (1998) neither provide information about the methods used in their program, nor do they define any standard approach. A few more details about how calculations are actually done in BROM would be of order here.

In general, the text really ought to be more complete and informative about numerical aspects of the model. This is what *Geoscientific Model Development* readers expect.

**2.2.3 Rate law expressions**

The tables that list all the processes considered in BROM and their rate laws, and that collect the different parameter values are among the most informative parts of the paper. They clearly represent one of its major strengths. Unfortunately no references are given for the parameter values presented in Table 3. There is a large variety of kinetic rate laws that are used in the model (Monod laws, squared Monod laws, laws in tanh, . . . ). I think it would be good to have a few words of explanation about the choices operated. Please also complete the references where missing (Table 2, on pp. 41–43 and Table 3, throughout).

As mentioned in the general appreciation already, I really wonder if all that complexity is really necessary, or, put the other way around: which minimalist set of process would be sufficient to obtain realistic results?

**2.2.4 Miscellanea**

Denitrification is considered, and nitrification, but I could not find anything about how nitrogen fixation is dealt with. I would expect that this process is required to avoid an unrealistic drift in the nitrogen inventory.

**2.3 Total alkalinity**

This part of the paper (p. 9) is one of the most disappointing ones. It is very approximate, completely overloaded with information that is ignored in the end. It furthermore contains several errors.

For clarity, it would be best to provide immediately the approximation actually used in the model, and not a hypothetical one, that could have been used. Alkalinity contributions that are not included or that are set to zero should be omitted. The text will be considerably simplified.

Whatever the expression chosen for total alkalinity, it will anyway always remain only an approximation. But even approximations need to be factually correct. Unlike written in the paper, ...

- ... $H_3PO_4$ is also part of alkalinity and $A_{TPO_4} = [HPO_4^{2-}] + 2[PO_4^{3-}] - [H_3PO_4]$ — interestingly this is correct in Table 6 (except for a typo) and also in the code;

- ... $NH_4^+$ is not part of alkalinity (it is the zero-level species) and thus $A_{TNH3} = [NH_3]$;

- ... it is the total borate concentration that is estimated from salinity and not $[B(OH)_4^-]$ — $[B(OH)_4^-]$ is calculated from the state variables just like to others (this is correctly done in the code, fortunately);

- ... $F^-$ is not part of alkalinity, only HF, so that $A_{THF} = [HF]$ — this is also wrong in Table 6 (at 68 $\mu$M, it would be barely negligible), but I suggest to discard the $A_{THF}$ term from the alkalinity expression anyway, as it is not included in the model.

Although it is specified later on that the stoichiometric constants of Roy et al. (1993) are used for the carbonate system, references for the other constants (e. g., dissociation constants for boric, phosphoric and silicic acids) required to solve the total alkalinity-pH equation are missing. Please provide references for those as well.

Finally, the pH scale used in the paper turns out to be the total scale. This should be stated more clearly than it is currently done (at my third reading, I discovered on p. 11 (l. 20) that "total pH was calculated". Please state this more obviously.

**2.4 Physical environment**

**2.4.1 Porosity**

Variable porosity is not included in the current version of BROM. The affirmation that "[...] its effect on [the] vertical transport is incorporated in[to] the values of $K_z$ and $K_{z_{bio}}$, [...]" (p. 13, l. 17) is rather obscure. $K_{z_{mol}}$ is actually constant so it is not clear how it could take porosity variations into account. I am furthermore not certain that this simplification is really necessary, given the complexity and detailed representation of the rest of the model. Variable porosity should not significantly increase the model's compleity.

Furthermore, it appears that a tortuosity corresponding to the porosity value of 90%

was used, with reference to a "value from Boudreau, 1997" (p. 13, l. 22). This is not very meaningful. Boudreau (1997) lists eight theoretically based tortuosity-porosity relationships and three empirical ones. Please specify which one was used here and then cite the original reference.

**2.4.2 Molecular diffusion**

BROM uses a species-independent molecular diffusion coefficient. This considerably simplifies the advection-diffusion-reaction equations, as the total concentrations a, such as DIC and alkalinity can be transported directly. The reported value $K_{z_{mol}} = 1 \times 10^{-11}\,\mathrm{m^2s^{-1}}$ is, however, almost two orders of magnitude lower than those for typical ions: e. g., from Boudreau (1997, Table 4.8), we may calculate diffusion coefficient values of $0.781 \times 10^{-9}\,\mathrm{m^2s^{-1}}$ for $HCO_3^-$, $0.632 \times 10^{-9}\,\mathrm{m^2s^{-1}}$ for $CO_3^{2-}$ and even $1.313 \times 10^{-9}\,\mathrm{m^2s^{-1}}$ for $HS^-$ (each one for $t = 10\,°C$). These are infinite dilution diffusion coefficients, but correcting them for tortuosity and for the dynamic viscosity of seawater does not reduce these values by more than 15–20%. How would results change if these much higher values would be used?

**2.4.3 Bioturbation**

Biotubation is parametrized as a diffusive process, as is common usage. For the biodiffusion coefficient, it is only stated that it takes a constant value over the top 2 cm and that it decreases exponentially afterwards. However, I have not been able to find the length scale of this decrease anywhere in the text. Now, one may ask whether it is realistic to consider any bioturbation at all in anoxic parts of the sediment, the more since the text already indicates that the maximal bioturbation depth was only 0.5–2.2 cm (p. 13, ll. 15–16). How would this change your conclusions?

**GMDD**

**2.4.4   Bioirrigation**

BROM takes the important process of bioirrigation into account. It is, however, represented as a purely diffusive process. Boudreau (1997) and Aller (2001) make a strong case that it would be more appropriate to represent bioirrigation as a non-local exchange process instead.

**2.5   Code**

On p. 7 (ll. 24–25), it is said that BROM consists of three modules. I did not want to download and install the complete FABM, but nevertheless wanted to inspect the BROM code, to find out more about the technical details that were missing from the paper. This was, however, not entirely straightforward.

**2.5.1   Accessibility**

After having opened http://fabm.org (which redirects to the FABM project page on SourceForge), I started to search for references to BROM. After some searching around, I detected the first trace of BROM under the "Wiki" tab: section 7 of chapter 2 of the User's Guide has the title "BROM-transport + FABM". BROM-transport is most probably the transport model mentioned in the paper (p. 7, l. 2), but that is not clear, since the paper always mentions BROM only. That section provides at least the first useful hint about where to find the BROM biogeochemical modules: under `src/models/niva/brom` in FABM. Proceeding to the "Code" tab then allowed me to browse to the relevant files (under the indicated directory tree). BROM-transport, however, is not with FABM and must be retrieved from a different repository, located at https://github.com/e-yakushev/BROM-transport, not mentioned in the paper.

I suggest that the authors give accurate and comprehensive instructions in the paper

about where the actual BROM source code files are located, both the biogeochemical ones and the main driver. And, please include also information about the license under which the code is distributed.

**2.5.2 Code quality**

The code is obviously "work in progress" and appears to undergo continuous changes. There are many lines of code that are commented out, some of them might be important. It is of not clear if they were also commented out when the results described in the paper were calculated.

I detected a few coding choices that put portability at risk. While `REAL`s in the three biogeochemistry related modules are declared in a portable way with `REAL(rk)`, where `rk` is an `INTEGER` parameter whose value gets derived from an appropriate `SELECTED_REAL_KIND(...)` call, there are some `INTEGER(4)` declarations that may lead to problems. In BROM-transport, there are numerous `REAL(8)` declarations, in different source code files. Kind type values – such as the '4' of the `INTEGER(4)` or the '8' of the `REAL(8)` declarations – are not standardized and may differ from one compiler to another. Programmers may not assume that they are equal to the expected byte length and for portability reasons kind type values must therefore not be hard-coded.[2] Portable and reliable code would consistently follow the FABM approach, with the `rk` parameter derived from `SELECTED_REAL_KIND(...)`

I have come across a few peculiarities or short-cuts in the code that may lead to serious confusion: e. g., in the subroutine `phIter` in `brom_carb.F90`, the `INTENT(IN)` argument `Sit_` (the total silicate concentration) is overridden by a local variable `Sit`,
* * *
[2]I know of one compiler where `DOUBLE PRECISION` is not `REAL(8)` but `REAL(3)`.

which is set to zero, thus making the code ignore silicate alkalinity. The paper does, however, not state that silicate alkalinity is ignored.

The pH calculation routine is neither safeguarded nor does it include diagnostics for possible convergence failures or for early convergence: it simply executes 100 Newton-Raphson iterations, starting from a preset fixed starting value, that furthermore seems to require manual modification from time to time. No diagnostic is included, neither for possible convergence failures nor for early convergence. (Why carry out 100 iterations if convergence is reached after five of them already?)

There are now reliable methods to solve the alkalinity-pH equation, which are guaranteed to converge under any physically meaningful conditions, howsoever exotic, and usually in less than six iterations (Munhoven, 2013). These would be particularly recommended in the environments that BROM has been developed for, with complex alkalinity compositions and unusual total concentrations.

Carbonate solubility constants do not take any pressure correction into account (the relevant lines are present, but commented out).

Finally, the comments in the code are not always correct, which also creates unnecessary confusion (e. g., the phosphoric alkalinity is not [H2PO4-] + 2.*[HPO4--] + 3.*[PO4---] as stated in a comment, but [HPO4--] + 2.*[PO4---] - [H3PO4]. Fortunately it is the latter that is implemented in the code.

**2.5.3 Permanent access to the code for model version 1.0**

As mentioned in the general appreciation, for model description papers there should exist a way to permanently access the precise model version described in the paper. The GitHub repository for BROM-transport includes a `Ver. 1.0` directory, so for the

transport model, this seems to be conceivable. The biogeochemical modules that are hosted in the FABM repository are however not clearly tied to version 1.0 of BROM.

It would thus be necessary to provide somehow tagged versions of the source code files for the model version 1.0 described here, or to provide copies of those files as a supplement to the paper.

**2.6   Tables**

The tables contain a wealth of information and represent one of the most useful parts of the paper (with the exception of Table 6, which could be deleted without loss). Unfortunately, Tables 1 and 4 are nearly unreadable because of the small font size. They would clearly benefit from a reorganization of their contents. Table 2 currently spans eight pages, Table 3 six pages. It would be useful to split them into smaller parts, with dedicated captions. While Table 2 still contains extensive references, Table 3 does not contain a single one. Readers ought to know where the adopted parameter values come from or how they have been derived.

The second column of the row "Alkalinity changes" in Table 2 is completely overloaded. Please reorganize this information.

Table 6 is not essential for the paper and I suggest to delete it altogether. It also contains errors and except for Canfield et al. (2005), none of the references cited is in the reference list. $A_{\text{THF}}$ is certainly not $68\,\mu$M, else it would not be negligible.

**3   Technical comments**

Throughout the paper: change "protolithic" to "protolitic" or "equilibrium" (depending on the context)

Throughout the paper: change "connected with" to "related to"

Throughout the paper: please check the usage of the word "parameterized" and "parameterization". For example, in Table 2, it is said that the carbonate system equilibration was parameterized. It were rather the stoichiometric constants that were parameterized, as a function of temperature, salinity and pressure, but the carbonate system equilibration (it would be more correct to say speciation) was calculated.

p. 4, l. 26: "death or flight"? "death or migration" would perhaps be more appropriate

p. 7, l. 15: "changeable" is not appropriate in this context. Perhaps "varying"?

p. 9, ll. 20–25: it is common usage to speak about bor*ate*, phosph*ate* and silic*ate* alkalinity (as with *carbonate alkalinity*) and to reserve the terms bor*ic*, phosphor*ic* and silic*ic* for the corresponding acids (as in *carbonic acid*).

p. 11, l. 20: change "Roy's constants" to "the set of constants of Roy et al." – the co-authors will appreciate

p. 16, ll. 4–5: change "FORTRAN" to "Fortran 2003" (spelling and standard) and change "Intel FORTRAN for Windows Compiler" to "Intel Fortran Compiler for Windows", which is the name of the product.

p. 16, l. 6: what is meant by "balanced distribution"?

pp. 21–26 (section 3.2.4 – section 3.4): please check for the English and rewrite where necessary.

p. 39, rows 10 and 11: "sulfatereduction" should read "sulfate reduction"

p. 40, second-last row, right-hand column: should the "$CaCO_3$" on the last linee not read "caco3_diss−caco3_prec"?

p. 41, row 7: there is probably some "$NO_3$"-"$NH_3$" mismatch here

p. 41, rows 7 and 8: the two trailing '2's in exponent seem to be misplaced (they probably belong to the second term in the denominator each time)

p. 46, in the first row relative to a half-saturation for OM denitrification, "$NO_2$" should probably read "$NO_3$"

Table 6: "$[PO_4^{2-}]$" should read "$[PO_4^{3-}]$"

**References**

Aller, R. C.: Transport and reactions in the bioirrigated zone, in: The Benthic Boundary Layer : Transport Processes and Biogeochemistry, edited by Boudreau, B. P. and Jørgensen, B. B., chap. 11, pp. 269–301, Oxford University Press, New York (NY), 2001.
Boudreau, B. P.: Diagenetic Models and Their Implementation, Springer-Verlag, Berlin, 1997.
Boudreau, B. P. and Jørgensen, B. B., eds.: The Benthic Boundary Layer : Transport Processes and Biogeochemistry, Oxford University Press, New York (NY), 2001.

Bruggeman, J. and Bolding, K.: A general framework for aquatic biogeochemical models, Environ. Model. Softw., 61, 249–265, doi:10.1016/j.envsoft.2014.04.002, 2014.

Lewis, E. and Wallace, D.: Program developed for $CO_2$ system calculations, Tech. Rep. 105, Carbon Dioxide Analysis Center, Oak Ridge National Laboratory, Oak Ridge (TN), available at http://cdiac.ornl.gov/oceans/co2rprt.html, 1998.

Munhoven, G.: Mathematics of the total alkalinity-pH equation – pathway to robust and universal solution algorithms: the SolveSAPHE package v1.0.1, Geosci. Model Dev., 6, 1367–1388, doi:10.5194/gmd-6-1367-2013, 2013.

Roy, R. N., Roy, L. N., Vogel, K. M., Porter-Moore, C., Pearson, T., Good, C. E., Millero, F. J., and Campbell, D. M.: The dissociation constants of carbonic acid in seawater at salinities 5 to 45 and temperatures 0 to 45 °C, Mar. Chem., 44, 249–267, doi:10.1016/0304-4203(93)90207-5, 1993.

---

## Author Comment (AC1) · 19 Jun 2016

Simulation of alternating oxic/anoxic conditions in coastal ecosystems on the fine spatio-temporal scales is useful for studies of specific questions, from an explicit description of the bottom boundary layer to a succession/alteration of multiple electron donor/acceptor agents to details of alkalinity composition and effects on the carbonate system, etc. Therefore the manuscript could be interesting to a wider audience and published also in the main body of Geoscientific Model Development papers. In that case, the manuscript demands a **major revision,** because both the form and content are rather sloppily observed and prepared. Many of specific issues and details of such revision have already been indicated by the first reviewer, Prof. J. Middelburg. I concur with almost all of them.

However, while trying to further expand the list of questions, suggestions, and requests, I got substantial doubts in the suitability of this specific manuscript for this particular journal, based on the following:

1.          Categorization of this manuscript as a "model description paper" requires a **comprehensive model description**, which internal consistency is verified by **demonstration of its capacities**, **rather than a detailed validation** of its implementation as would be ex- pected from a "model evaluation paper". The ambiguity of the paper's goals is reflected in repeating expressions like "to develop a model AND analyse seasonal effects". As it looks now, the manuscript describes a specific model implemented for studies of some particular biogeochemical questions rather than **presents some finished single product** that can be relatively straightforwardly borrowed and used by interested colleagues.

The text was extensively modified to become a comprehensive model description rather than a validation. We use an example of calculations to demonstrate the model capacity (this part was significantly reduced). The code was re-written in many parts and commented to facilitate its use by interested colleagues.

2.          Such ambiguity starts already from rather inconsistent definition of objectives. The title announces "coupled benthic-pelagic model for simulation of seasonal anoxia", the abstract indicates the goal as a capturing of "biogeochemical processes occurring at the bottom boundary layer (BBL) AND sediment-water interface (SWI)", the last sentence of "Background" Section indicates the goal as a capturing of "key biogeochemical processes occurring at the bottom boundary layer" only. Even farther, "the main goal of the model was to reproduce the biogeochemical mechanism of transformation of oxic conditions into anoxic in the sediment–water interface". Perhaps, such obscurity reflects also a story of development of BROM from ROLM by substantially expanding list of variables and their interactions. If, as it seems to me, the real focus and achievements lay in the "middle", then

almost a sole goal of the water column and sediment parts is to generate consistent boundary conditions for interacting BBL and SWI. From the manuscript it is also unclear, why the focus is on seasonal dynamics and what prevents the reproduction of sporadic short-term alterations or long-term persisting states.

The title and formulations of the goals in the abstract and text have been harmonized. A focus on seasonal dynamics was also deleted from the title following the Reviewer's suggestions. For the example calculations we focus on a seasonal cycle because much of the strongest biogeochemical variability (including deoxygenation) typically occurs on this time scale. However, we are clear in the revised text that BROM can be applied to study variations on a broad range of time scales.

Then, for a further implementation in diverse geographical areas it should be stressed and clearly explained, where from should the user obtain the data about external inputs, internal dynamics and distribution on multiple forms of sulfur, manganese, iron, as well as on different functional groups of bacteria. At the least, recommendations should be given on some proxies that could be derived from the pelagic ecosystem models with less uncommon sets of variables and processes.

A step-by-step guide to applying the model in given geographical area has been added to the text (Appendix A: Running BROM step-by-step). This guide includes recommendations on where necessary model inputs may be found (i.e. observations data, models output, databases, literature estimates). The issue of missing model inputs/data is now clearly confronted in the General Description (section 2.1.1):

"We acknowledge that for many of these additional variables, site-specific estimates of associated model parameters and initial/boundary conditions may be difficult or impossible to obtain, and may in practice require some crude assumptions and approximations (e.g. universal default parameter values, no-flux boundary conditions, initial conditions from a steady annual cycle). Nevertheless, we believe that for many applications this will be a price worth paying for the additional process resolution/realism provided by BROM for important biogeochemical processes in the BBL and sediments."

Besides, we made references in the Tables 2 and 3 regarding the processes formulations and the coefficients values

Furthermore, there are several ad hoc features and patches pertaining, perhaps, only for this implementation that should be explicitly indicated for a prospective users, for instance, holding sea surface concentrations constant results in non-conservation; prescription constant coefficient of vertical transport in BBL, while arbitrarily modifying it by assumed bioturbation in the sediments; extensive use of squared availabilities (Nutrient/Biomass)^2 instead of concentrations N in nutrient limitation and trophic functions.

In the modified submission we have improved the flexibility of the model code and clarified

the use of simplifying assumptions, including references where possible in the model code and input .yaml files. The BROM-transport model now allows the use of 4 different types of boundary conditions: 1) no flux (Neumann) except where surface fluxes are parameterized (Robin), 2) fixed constant (Dirichlet), 3) fixed sinusoidal variability (Dirichlet), 4) fixed arbitrary variability read from netCDF (Dirichlet). Choices 2-4 will result in a non-conservative simulation in the sense that material is added or removed from the model domain without an explicit treatment of the boundary flux; however, this may in many cases be more realistic than assuming no flux or a fixed constant flux. Regarding vertical diffusivity, the variation in the BBL can now be parameterized in two ways: 1) as a constant value, motivated by simplicity, and 2) as a linear variation, corresponding to a logarithmic layer for velocity (Holtappels and Lorke, 2011). These are both, of course, simplifying assumptions but we believe that they are justifiable as a first approximation. In the sediments, bioturbation can be the dominant process contributing to vertical transport and so should not generally be neglected. In the present version we have a 3-parameter model for the vertical variation of bioturbation in the sediments, including a constant maximum level near the surface followed by an exponential decay below a certain depth. We believe this should be flexible enough for most users.

Regarding the use of squared availabilities an explanation has been added to the text:

"The redox-dependent switches are preferably based on hyperbolic functions that improve system stability compared with discrete switches. The nutrient limitation and trophic functions are preferably based on squared Monod laws for Nutrient/Biomass ratio, which also stabilizes the system compared with Michaelis-Menten and Ivlev formulations."

Fortunately, selected results, ideas and formulations can still be gratefully borrowed by interested colleagues with appropriate reference to the ever available discussion paper.

---

## Author Response (AR1)

*Response to the review of J. Middelburg*

The authors present a rather complex model allowing simulation of biogeochemical processes in coastal systems subject to seasonal anoxia. The paper has a few strengths and many weaknesses. Most numerical biogeochemical models focus ei- ther on the water column or on the sediments and very few couple these domains. The presented BROM model **does explicitly deal with the coupled system** and is therefore of value.

Strengths of this paper include the (a) coupling of pelagic and benthic modules, (b) the explicit resolution of the benthic boundary layer (BBL), (c) the focus on seasonal hypoxia and (d) its linkage to/integration with the Framework for Aquatic Biogeochemical Modeling. These characteristics make this an interesting paper and the presented model is potentially useful.

However, there are many issues to be resolved before publication of this paper and model. 1. The paper is poorly written in terms of organization, flow and **use of English**. A few examples of the latter (line 1: seawater and benthic sediments, benthic sys- tems or sediments are fine, but not benthic sediments; the use of the term protolithic processes: do you refer to stone age processes? or do you simply mean equilibrium processes, etc. etc.). There appear native speakers and/or UK/Canada based scientists among the authors: perhaps they should have another look at it. The text is also not prepared with utmost care: many typos, wrong equations etc. (see below in the list of detailed comments).

We apologize for the condition of the original submitted manuscript, and thank the Reviewer for nevertheless providing a detailed and constructive review which has contributed to a major improvement (in our opinion) in the model code and description. The new submission has been thoroughly revised to improve structure, language, and accuracy of the equations.

The model could be much better put in context and **existing literature** is poorly incorporated. I was missing references to other papers dealing with coupled benthic-pelagic models, the more simpler ones (Lancelot, Soetaert, Fennel, Reed and co-workers) and the highly complex ones from the ERSEM family. Soetaert and co-workers (Soetaert and Middelburg, 2009; Meire et al., 2013) have published on seasonal oxygen issues with coupled pelagic benthic models. There is also a large body of knowledge on the effect of oxygen in early diagenetic models; that literature is not covered. Extensive work on the role of sediments as moderating the timing of return fluxes (delay in return of N, P, Si after bloom) and the memory provided by stored reduced sulfur, iron and carbon in sediments is poorly covered (see work on Gulf of Mexico by Nancy Rabalais and co-workers).

We thank the reviewer for drawing our attention to this work, which is indeed highly relevant. The Background section has been extended to provide a more thorough summary of existing literature, including all of the models cited above. It remains, however, a summary and not an in-depth review; the latter is beyond our intended scope.

The model **description** is often incomplete and imprecise (see below for a far from complete list). The documentation is not sufficient. Boundary conditions of the model are not clearly described. Details about the coupling of the models are insufficient: e.g. the grid size is very likely changing, yet not provided. It is unclear whether particulate organic matter is modeled explicitly. Is it transported by bioturbation in the sediment. It is unclear how bio-irrigation and solid-phase mixing are treated.

1    Sometimes parameters are introduced in the text, but appear as fixed value (hard-coded) in the tables.

3    We acknowledge and apologize for these shortcomings.  The new submission offers a much more
4    thorough description.   Boundary conditions are now described in a dedicated section 2.2.5. The
5    BROM-transport grid, which combines water column and sediment subgrids, is now described in a
6    dedicated section 2.2.3. Dead particulate organic matter is explicitly modeled, as is now stated clearly
7    in section 2.1.2.  Particulate variables are diffused in the sediment by bioturbation – this is now
8    clarified in section 2.2.1 which describes the BROM-transport model formulation.  In BROM-transport,
9    bio-irrigation is treated as a non-local exchange process following (Boudreau, 1997; Schluter et al.,
10   2000; Meile et al., 2001) (see section 2.2.1).  Mixing of solid phase constituents is only by bioturbation
11   in the sediments.  Mixing of the solid phase as a whole (interphase mixing) in BROM-transport may
12   occur only by bioturbation at the sediment-water interface.  These processes are now clearly
13   described in the new section 2.2.1.

15   The model is very complex and detailed (perhaps too much) in some aspects and very rudimentary
16   in other aspects. Regarding the latter, many detailed Mn, Fe, S transformations are included,
17   chemoautotrophy is resolved for aerobic and anaerobic  microbes (or only bacteria?), but important
18   processes such as methane generation and  anaerobic methane oxidation coupled to sulfate reduction
19   are ignored. Another example particle settling velocities is corrected for the formation of Mn-oxides in
20   the water  column but other carrier phases such as calcium carbonate are not resolved. Clearly, the
21   presented model is a version 1.0 and represents a first step, but the priorities of  the authors do not
22   match those of the majority of the audience. At the minimum some  motivation for their particular choice
23   should be communicated to the audience.

25   The motivation for the complexity of BROM is discussed in the new text (section 2.1.1):

26   "The model has 33 state variables ($C_i$), described in Table 1. This includes frequently measured
27   components such as hydrogen sulfide (H2S) and phosphate (PO4), as  well as rarely measured
28   variables such as elemental sulfur (S0), thiosulfate (S2O3), trivalent manganese species Mn(III), and
29   bacteria. Variables of the latter category were included because their contribution to biogeochemical
30   transformations is believed to be substantial. For instance, bacteria play an important role in many
31   modelled processes and can consume or release nutrients in organic and inorganic forms (Canfield et
32   al., 2005; Kappler et al., 2005). We acknowledge that for many of these additional variables, site-
33   specific estimates of associated model parameters and initial/boundary conditions may be difficult or
34   impossible to obtain, and may in practice require some crude assumptions and approximations (e.g.
35   universal default parameter values, no-flux boundary conditions, initial conditions from a steady annual
36   cycle).   Nevertheless, we believe that for many applications this will be a price worth paying for the
37   additional process resolution/realism provided by BROM for important biogeochemical processes in the
38   BBL and sediments."

39   The definition of the "bacteria" model compartment is made precise in the new text:

40   "We divide all the living OM (biota) into Phy (photosynthetic biota), Het (non-microbial heterotrophic
41   biota), and 4 groups of "bacteria" which may be considered to include microbial fungi."

42   The processes of methanogenesis and methane oxidation with oxygen have been added to BROM-
43   biogeochemistry.

44   The effect of accelerated particle settling velocities has in fact been removed in the new code, and the
45   text has been adjusted accordingly.

Section 3.1 on model output **discussion** needs major revision. The link with the figures is unclear and the organization is suboptimal. You discuss the oxygenated winter period and then link to later periods or a few days later in the section on oxygenated winter period. There is no **story**. Try **to limit yourself to a few findings and discuss those**. The reader now has to digest all the computer output her or himself.

Section 3 was shortened and re-structured. Following the reviewers' recommendations we now just focus on describing the ability of BROM to simulate changes in the redox conditions and illustrating the rates of processes and transport fluxes.

**Section 3.2** is not useful or convincing. The link with data is very poor. This is indeed a difficult job, but here serious work has been done. A comparison with just three to four papers is made and the extensive databases on oxygen uptake, oxygen penetration depth etc are not consulted. A statement like line 10 on page 20: "further observations under anoxic and suboxic conditions are rare as field and experimental studies generally focus on oxic conditions" is close to unacceptable. Consider all the work done on the Eastern Pacific ocean margins (Washington and Oregon shelves, California basins, Mexico/Peru/Chile margin), Indian ocean and shelves and multiple European systems (Black Sea, Baltic Seas). There are nice seasonal time series in coastal systems with low-oxygen events during summer (Kiel bight, etc.)

Section 3.2. was removed from the text.

Minor issues:

Page 3: - Line 2: benthic sediments? –

Modified to "benthic systems"

Line 3: directly affect and are impacted by.. –

Sentence modified to:

"Benthic fluxes of chemical elements (C, N, P, O, Si, Fe, Mn, S) alter redox state and acidification (i.e. pH and carbonate saturation), which in turn affect the functioning of benthic and pelagic ecosystems."

Line 7: fuzzy writing, rewrite

Sentence modified to:

"The redox state of the near bottom layer in many regions can change with time, responding to the supply of organic matter, physical regime and coastal discharge."

Page 4: - Line 3: enrichment with OM or do you mean deposition of labile organic matter –

Sentence modified to:

"The sediment generally consumes oxygen due to deposition of labile OM and presence of reduced forms of chemical elements"

Line 26-27: animals can die, migrate or change their behavior: revise text accordingly

Modified to:

"This may lead to death, migration, or changed behavior of the benthic macro- and meiofaunal organisms responsible for bioturbation and bioirrigation…"

Page 5: - Line 4-15: **additional literature** incorporation required. How does your model differ from those. Where are the improvements etc.

Additional literature has been incorporated in the new Background section. The ways in which BROM differs from existing models are now explicitly listed in the final paragraph of this section.

- Line 22: the benthic boundary layer is a major strength. Introduce it better. Delineate the features, etc. –

The BBL is now better introduced in the second last paragraph of the Background:

"The BROM model described herein is a fully coupled benthic-pelagic model with a special focus on deoxygenation and redox biogeochemistry in the sediments and Benthic Boundary Layer (BBL). The BBL is "the part of the marine environment that is directly influenced by the presence of the interface between the bed and its overlying water" (Dade et al., 2001). Physical scientists tend to prefer the term "bottom boundary layer", but this is largely synonymous with the BBL (Thorpe, 2005). Within BROM, the term BBL is used to refer to the lower parts of the fluid bottom boundary layer where bottom friction strongly inhibits current speed and vertical mixing, hence including the viscous and logarithmic sublayers up to at most a few metres above the sediment. This calm-water layer plays a critical role in mediating the interaction of the water column and sediment biogeochemistry and in determining e.g. near-bottom oxygen levels, yet it remains poorly resolved in most physical circulation models. For BROM we have developed an accompanying offline transport module "BROM-transport" that uses output from hydrodynamic water column models but solves the advection-diffusion-reaction equations for a "full" grid including both water column and sediments. BROM-transport uses greatly increased spatial resolution near to the SWI, and thereby provides explicit spatial resolution of the BBL and sediments."

Line 24: at the BBL: do you exclude the sediments here?

Our BBL definition does exclude the sediments (see above), but the scope of BROM does not. This scope or goal is now better defined:

"The goal of this work was to develop a model that captures key biogeochemical processes in the water and sediment and to analyze the changes occurring in the BBL and SWI."

Page 6: - line 9-10: it is unclear whether organic C is also modeled or is it just inorganic

C. - Line 12: No nitrogen transformations?

Only inorganic C is explicitly modeled (state variable name DIC). Organic matter (dissolved and particulate) is modeled only in nitrogen currency (variable names DON and PON) so to derive organic C estimates from model output would require use of a stoichiometric ratio C:N. Nitrogen transformations are modelled. The new manuscript reads:

1  "BROM considers interconnected transformations of species of (N, P, Si, C, O, S, Mn, Fe) and
2  resolves OM in nitrogen currency. OM dynamics include parameterizations of OM production (via
3  photosynthesis and chemosynthesis) and OM decay via oxic mineralization, denitrification, metal
4  reduction, sulfate reduction and methanogenesis."

6  Page 7: - Line 2: delete consists

7  Done

9  - Line 26 and all through: replace protolithic processes  with **equilibrium processes/reactions or acid-**
10  **base reactions**

11  Done

13  Page 8: - Line 11: provide the number of state variables to the reader

14  Done (33).

16  Page 9: - Line 1: chemoautotrophy is resolved, but overall **secondary production** is ignored.
17  There may be good reasons for this, but communicate this then to the reader

18  Secondary production is resolved.  The state variable 'Het' represents all non-microbial heterotrophs.
19  These graze the phytoplankton as well as bacteria and detritus, and they reach significant
20  concentrations in both pelagic and benthic parts of the model domain.  The new section 2.1.2 reads:

21  "…We divide all the living OM (biota) into Phy (photosynthetic biota), Het (non-microbial heterotrophic
22  biota), and 4 groups of "bacteria" which may be considered to include microbial fungi.  These latter
23  are: Baae (aerobic chemoautotrophic bacteria), Baan (anaerobic chemoautotrophic bacteria), Bhae
24  (aerobic heterotrophic bacteria), and Bhan (anaerobic heterotrophic bacteria).  OM is produced
25  photosynthetically by Phy and chemosynthetically by bacteria, specifically by Baae in oxic conditions
26  and by Baan in anoxic conditions. Growth of heterotrophic bacteria is tied to mineralization of OM,
27  favouring Bhae in oxic conditions and Bhan in anoxic conditions. Secondary production is represented
28  by Het which consumes phytoplankton as well as all types of bacteria and detritus…"

30  - Line 8-10: why is methanogenesis excluded? This is probably related to the way you model
31  organic matter. Conceptually most simple is to turn all labile organic matter remaining after depletion of
32  all oxidants into methane and carbon dioxide.

33  Methanogenesis is included in the new, modified version

36  - Line 15- 25: the alkalinity equations as given are wrong: the phosphate alkalinity term should
37  include a H3PO4 term, the ammonium alkalinity term should not include NH4+, etc. Please check
38  whether you have implemented it correctly into your model.

39  Checked and corrected.

2  Page 10:-Line 2: I guess you mean Atom was set to zero and not TOM.

3  Correct. In the new text, the TOM alkalinity is removed.

5  Page 11: -Line 1-10: quite a number of the reactions are not balanced and inconsistent with Table 2: e.g.
6  denitrification with hydrogen sulfide and the line above represent two reactions of which the latter
7  misses a two before OH-. Check carefully. -Line 11: Table 2 not 3.

8  Apologies for our sloppy editing. All equations have now been carefully checked and corrected.

10  Page 12: middle of page: a distinction is made between settling velocity of particulate matter and of
11  particles with Fe and Mn oxides. Why not write particle settling velocities (w) as a sum of various
12  contributing terms. Why the focus on Fe and Mn? Just a Black Sea model heritage?

13  In the new version, all the inorganic particles sink at the same constant velocity, and this velocity is
14  larger than all organic matter sinking velocities.

16  Page 13: - Line 8: the eddy diffusion coefficient was assumed constant in the BBL. As a first
17  coupled model that resolves the BBL **it may be done like this**, but given the **lognormal velocity**
18  **profiles,** would one not expect a depth profile in Kz as well. This can be incorporated quite easily.

19  A good suggestion, thanks. In the new version, Kz (now D, to be more conventional) can have a
20  linear depth variation either if it is treated dynamically or assuming a static log layer. This is now
21  described in section 2.2.7:

23  "The vertical diffusivity needs a more careful treatment as it is the main defining characteristic of the
24  pelagic vs. BBL vs. sediment environments. Within the water column, the total vertical diffusivity D =
25  Dm + De for solutes and D = De for particulates, where Dm is a constant molecular diffusivity at
26  infinite dilution, and De is the eddy diffusivity read from the input file for the pelagic water column. For
27  the BBL, De can be defined as "dynamic", in which case it is linearly interpolated for each day
28  between the deepest input forcing value above the SWI and zero at a depth hDBL above the SWI,
29  where hDBL is the diffusive boundary layer (DBL) thickness (default value 0.5 mm). This option is
30  likely appropriate for shallow water applications where De may be strongly time-dependent within the
31  user-defined BBL (default thickness 0.5 m). Alternatively, a static, fixed profile DeBBL(z) may be
32  more appropriate for deep water BBLs, where time dependence may be weak and deepest values
33  from hydrodynamic models may be relatively far above the SWI. In this case, BROM-transport offers
34  two options for DeBBL(z): 1) a constant value, dropping to zero in the DBL, or 2) a linear variation
35  between a fixed value at the top of the BBL and zero at the top of the DBL. Option 1) defines a
36  simplest-possible assumption, while option 2) corresponds to the assumption of a log layer for the
37  current speed (e.g. Boudreau and Jorgensen, 2001). Eddy diffusivity is strictly zero in the DBL, on the
38  SWI, and within the sediments. Diffusivity in the sediments is due to molecular diffusion and
39  bioturbation and is parameterized as described in section 2.2.1."

- Line 15-20: the description of bioturbation/bioirrigation is difficult to follow. As written above are solid phase and solutes transported separately or together? This is unclear. Bioturbation depth are very shallow for fully oxic conditions.

This part of the model was significantly improved and is clearly described in the new section 2.2.1. Solid and liquid phases are diffused separately (intraphase mixing) except possibly at the sediment-water interface if the option to allow bioturbation across the SWI is enabled. Solute diffusivity in the sediments is a sum of molecular diffusivity, corrected for tortuosity and relative viscosity following Boudreau (1997), and bioturbation diffusivity, depending on a fixed vertical profile and a time-dependent oxygen status of the bottom layer of the water column (fluff layer). Particulate diffusivity in the sediments is just the bioturbation diffusivity. Solute burial velocity also now differs from particulate burial velocity due to the effects of compaction (Boudreau, 1997). Burial velocities now also depend on depth under an assumption of steady state compaction (Berner, 1970, 1981; Boudreau 1997; Meysman et al., 2005) and additional velocity components can optionally be added to account for modelled particulate fluxes to the SWI and particulate reactions in the sediments (see section 2.2.1 and Appendix B).

The current default "mixed layer" depth for bioturbation is 2 cm (cf. values 5 cm and 1 cm used by Soetaert and Middelburg (2009) for well-mixed and anoxic conditions respectively). The default decay scale for bioturbation diffusivity below the mixed layer is 1 cm, following Soetaert and Middelburg (2009). This information is now included in the run-time input file brom.yaml (see Appendix D). We agree that a 2 cm mixed layer may be too shallow for fully oxic conditions; in such cases the user should increase the mixed layer depth parameter in the brom.yaml file.

Section 2.3. The description of boundary conditions needs more attention. It appears that you use flux boundary conditions for the gases and constant or fixed (at least im- posed) boundary for the others. This may lead to mass balance issues. The boundary conditions at the bottom (depth 12 cm in sediments) are not described: no flux or **no gradient** or fixed concentration? The assumed sulfate concentration is either close to zero or do you mean 25 10-3M? Basically all external sources such as atmosphere and rivers are added to surface layer.

In the new BROM-transport code we allow four different options to define boundary conditions (upper and lower) for each variable. For the upper boundary (air-sea interface) the default option is no flux, unless the flux is specifically parameterized by the (FABM) biogeochemical model. For BROM-biogeochemistry this means that, by default, all variables have no flux surface boundary conditions except oxygen and DIC, which have fluxes parameterized using atmospheric oxygen and $CO_2$ levels prescribed in the brom.yaml file (see Appendix D). Optional fixed (Dirichlet) boundary conditions do indeed imply mass fluxes into or out of the modelled water column, but this need not be unrealistic. We have found in fjord and lake applications that fixed (possibly time-dependent) surface boundary condition can provide a way of modelling missing net influxes of nutrients from rivers. Boundary conditions at depth are no-gradient by default (advective outfluxes can still occur due to burial velocity). External sources (e.g. from rivers) can also be allowed to contribute directly to the model interior by setting the "horizontal mixing" forcings, rather than by setting boundary conditions. This all described in the new section 2.2.5:

"BROM-transport presently allows the user to chose between four different types of boundary condition for each variable and for upper and lower boundaries: 1) no-gradient at the bottom boundary (no diffusive flux) or no-flux at the surface boundary, except where parameterized by the FABM

biogeochemical model (i.e. for O2 and DIC in the case of BROM-biogeochemistry); 2) a fixed constant value; 3) a fixed sinusoidal variation in time defined by amplitude, mean value, and phase parameters; or 4) an arbitrary fixed variation in time read from the input netCDF file. All boundary condition options and parameters are set in the brom.yaml file (see Appendix D). Note that options 2-4 are Dirichlet boundary conditions which define implicit fluxes of matter into and out of the model domain, and that all boundary concentrations should be in units [mmol/m3 total volume (water+solids)]. The default option 1 is generally the preferred choice, but the Dirichlet options can also be useful to allow a simple representation of e.g. fluxes of nutrients into and out of the surface layer due to lateral riverine input. A possible alternative is to use the forcings parameters for horizontal mixing (see equation (1)) to specify horizontal exchanges or restoring terms to observed climatology (see section 2.2.7)."

For the sulfate upper and lower boundary conditions we have used Dirichlet conditions of 25000 uM (or mmol/m3) for both.

Section 3: I stop making detailed feedback because there are too many issues and the referee already spent double the amount of time normally needed for an evaluation.

Again we sincerely apologize for the condition of the original submitted manuscript. We are confident that the new version will not require so much correction.

p. 26, line 5-9: chemoautotrophy indeed involves CO2 consumption and thus has the potential to increase pH. However, the energy required for CO2 fixation is obtained from oxidation of reduced products: usually an acid producing process. With typical growth efficiencies one would produce more acid linked for the energy than consumption of acid by organic matter production. Cable-bacteria spatially disconnect half reactions and can therefore cause a real pH increase. Without detailed model investigations, I suggest removing these sentences. The authors might be right because of the com- plexity of reactions and the many buffering reactions, but it is not convincing as pre- sented here.

This part of discussions was removed from the modified version.

Table 1: it is stated that oxygen is presented in microM O, but sometimes it might be, at other places it is definitely in microM O2.

Corrected. O2 is now always present in microM O2.

Table 2: - Aerobic respiration and denitrification are treated different than Fe, Mn and sulfate reduction regarding DON and PON separation. - For Mn reduction where does the 0.5 come from (half saturation constant hard-coded?) - There are multiple typos which complicate checking. - Where is the (1+ftD(t)) term coming from. ftD is not defined. - Page 41: I guess that NO3 dependence should depend on nitrate and not on ammonia?

Apologies again. Table 2 has now been checked and corrected.

The factor 0.5 in the Mn reduction formulations is not a hard-coded half saturation constant (all half saturation constants are input parameters in fabm.yaml). It is rather there to ensure that the specific Mn reduction rates at high Mn concentration (tanh function tending to unity) and high $H_2S$ concentration (Michaelis-Menten function tending to unity) are indeed set by the limiting rate parameters K_mn_rd1 and K_mn_rd2.

The variable ftD(t) and its corresponding dependence have been removed in the new version.

Apologies for the typo in NO3 dependence. This should have been a combined function of nitrate and ammonium (nitrate uptake suppressed at high ammonium concentrations). It is correct in the new Table 2.4.

Table 3: - I guess that K_Mn_rds should be K_Mnrd_HS? - There are many values assumed, some literature citations would be helpful. I guess that the model is rather insensitive to most of these parameters and their value should therefore be based on literature values.

Sorry, K_Mn_rds was a typo.

Literature citations have been added to the new tables. Table 3 has been checked, corrected, and divided into several tables.

- Why did you choose 2.7 for the Fe/P ratio and not the conventional 10?

Table 4: check carefully: e.g. for phosphate you have hard-coded 2.7 for Fe/P and 0.67 for Mn/P rather than a parameter. Taking stoichiometry as a constant is fine, but do not present

An explanation has been added to the text. We refer to assumptions and numerical experiments described in (Yakushev et al., 2007), where we aimed to analyze the reasons for formation of a typical "phosphate dipole" in the water column, with a minimum just above, and a maximum just beneath the hydrogen sulfide onset. We used extreme values of Fe/P and Mn/P to demonstrate that this phenomenon cannot be explained by Fe (even if Fe/P = 2.7, and not 10), but can be explained by Mn(III).

Table 5: this table is unuseful and I doubt whether the fluxes are all in the right units. Table 6: could be deleted.

Both tables have been deleted.

Figures. All captions need more documentation. For instance it is not even mentioned why some concentrations are presented on two different scales. As written above, reconsider to focus on a few results and elaborate the model results in another paper. The figures as presented now appear more like raw model output.

The figures has been redrawn and carefully selected. As recommended we focus on demonstrating of the model possibilities and not on analyzing the model results.

1 *Response to the review of O.P. Savchuk*

2 Simulation of alternating oxic/anoxic conditions in coastal ecosystems on the fine spatio-
3 temporal scales is useful for studies of specific questions, from an explicit description of the
4 bottom boundary layer to a succession/alteration of multiple electron donor/acceptor agents to
5 details of alkalinity composition and effects on the carbonate system, etc. Therefore the
6 manuscript could be interesting to a wider audience and published also in the main body of
7 Geoscientific Model Development papers. In that case, the manuscript demands a **major**
8 **revision,** because both the form and content are rather sloppily observed and prepared. Many of
9 specific issues and details of such revision have already been indicated by the first reviewer, Prof. J.
10 Middelburg. I concur with almost all of them.

11 We apologize to the Reviewer for the poor condition of our submitted manuscript, and we thank the
12 Reviewer for nevertheless providing a constructive review.  We are confident that this review, in
13 combination with the other two, has led to a major improvement in the model code and description.

15 However, while trying to further expand the list of questions, suggestions, and requests, I got
16 substantial doubts in the suitability of this specific manuscript for this particular journal, based on
17 the following:

18 Categorization of this manuscript as a "model description paper" requires a **comprehensive**
19 **model description**, which internal consistency is verified by **demonstration  of its capacities**,
20 **rather than a detailed validation** of its implementation as would be ex- pected from a "model
21 evaluation paper". The ambiguity of the paper's goals is reflected in repeating expressions like "to
22 develop a model AND analyse seasonal effects". As it  looks now, the manuscript describes a
23 specific model implemented for studies of some  particular biogeochemical questions rather than
24 **presents some finished single product** that can be relatively straightforwardly borrowed and
25 used by interested colleagues.

26 The text was extensively modified to become a comprehensive model description rather than a
27 validation. We use an example of calculations to demonstrate the model capacity (this part was
28 significantly reduced). The code was re-written in many parts and commented to facilitate its use by
29 interested colleagues.

31 Such ambiguity starts already from rather inconsistent definition of objectives. The  title announces
32 "coupled benthic-pelagic model for simulation of seasonal anoxia", the  abstract indicates the goal

as a capturing of "biogeochemical processes occurring at the bottom boundary layer (BBL) AND sediment-water interface (SWI)", the last sentence of "Background" Section indicates the goal as a capturing of "key biogeochemical processes occurring at the bottom boundary layer" only. Even farther, "the main goal of the model was to reproduce the biogeochemical mechanism of transformation of oxic conditions into anoxic in the sediment–water interface". Perhaps, such obscurity reflects also a story of development of BROM from ROLM by substantially expanding list of variables and their interactions. If, as it seems to me, the real focus and achievements lay in the "middle", then almost a sole goal of the water column and sediment parts is to generate consistent boundary conditions for interacting BBL and SWI. From the manuscript it is also unclear, why the focus is on seasonal dynamics and what prevents the reproduction of sporadic short-term alterations or long-term persisting states.

The title and formulations of the goals in the abstract and text have been harmonized. A focus on seasonal dynamics was also deleted from the title following the Reviewer's suggestions. For the example calculations we focus on a seasonal cycle because much of the strongest biogeochemical variability (including deoxygenation) typically occurs on this time scale. However, we are clear in the revised text that BROM can be applied to study variations on a broad range of time scales.

Then, for a further implementation in diverse geographical areas it should be stressed and clearly explained, where from should the user obtain the data about external inputs, internal dynamics and distribution on multiple forms of sulfur, man- ganese, iron, as well as on different functional groups of bacteria. At the least, recom- mendations should be given on some proxies that could be derived from the pelagic ecosystem models with less uncommon sets of variables and processes.

A step-by-step guide to applying the model in given geographical area has been added to the text (Appendix A: Running BROM step-by-step). The issue of missing model inputs/data is now clearly confronted in the General Description (section 2.1.1):

"The model has 33 state variables ($C_i$), described in Table 1. This includes frequently measured components such as hydrogen sulfide ($H_2S$) and phosphate ($PO_4$), as well as rarely measured variables such as elemental sulfur ($S0$), thiosulfate ($S_2O_3$), trivalent manganese species Mn(III), and bacteria. Variables of the latter category were included because their contribution to biogeochemical transformations is believed to be substantial. For instance, bacteria play an important role in many modelled processes and can consume or release nutrients in organic and inorganic forms (Canfield et al., 2005; Kappler et al., 2005). We acknowledge that for many of these additional variables, site-specific estimates of associated model parameters and initial/boundary conditions may be difficult or

impossible to obtain, and may in practice require some crude assumptions and approximations (e.g. universal default parameter values, no-flux boundary conditions, initial conditions from a steady annual cycle). Nevertheless, we believe that for many applications this will be a price worth paying for the additional process resolution/realism provided by BROM for important biogeochemical processes in the BBL and sediments."

Furthermore, there are several ad hoc features and patches pertaining, perhaps, only for this implementation that should be explicitly indicated for a prospective users, for instance, holding sea surface concentrations constant results in non-conservation; prescription constant coefficient of vertical transport in BBL, while arbitrarily modifying it by assumed bioturbation in the sediments; extensive use of squared availabilities (Nutrient/Biomass)ˆ2 instead of concentrations N in nutrient limitation and trophic functions.

In the modified submission we have improved the flexibility of the model code and clarified the use of simplifying assumptions, including further comments and references in the text, Tables, model code, and input .yaml files (Appendices C and D). Regarding boundary conditions, the flexibility of the BROM-transport code has been improved and the options are now described and explained in section 2.2.5:

"BROM-transport presently allows the user to chose between four different types of boundary condition for each variable and for upper and lower boundaries: 1) no-gradient at the bottom boundary (no diffusive flux) or no-flux at the surface boundary, except where parameterized by the FABM biogeochemical model (i.e. for O2 and DIC in the case of BROM-biogeochemistry); 2) a fixed constant value; 3) a fixed sinusoidal variation in time defined by amplitude, mean value, and phase parameters; or 4) an arbitrary fixed variation in time read from the input netCDF file. All boundary condition options and parameters are set in the brom.yaml file (see Appendix D). Note that options 2-4 are Dirichlet boundary conditions which define implicit fluxes of matter into and out of the model domain, and that all boundary concentrations should be in units [mmol/m3 total volume (water+solids)]. The default option 1 is generally the preferred choice, but the Dirichlet options can also be useful to allow a simple representation of e.g. fluxes of nutrients into and out of the surface layer due to lateral riverine input. A possible alternative is to use the forcings parameters for horizontal mixing (see equation (1)) to specify horizontal exchanges or restoring terms to observed climatology (see section 2.2.7)."

Regarding vertical diffusivity, the variation in the BBL can now be parameterized in three ways, as described in section 2.2.1:

1  "Within the water column, the total vertical diffusivity $D = D_m + D_e$ for solutes and $D = D_e$ for

2  particulates, where $D_m$ is a constant molecular diffusivity at infinite dilution, and $D_e$ is the eddy

3  diffusivity read from the input file for the pelagic water column. For the BBL, $D_e$ can be defined as

4  "dynamic", in which case it is linearly interpolated for each day between the deepest input forcing

5  value above the SWI and zero at a depth $h_{DBL}$ above the SWI, where $h_{DBL}$ is the diffusive boundary

6  layer (DBL) thickness (default value 0.5 mm).  This option is likely appropriate for shallow water

7  applications where $D_e$ may be strongly time-dependent within the user-defined BBL (default

8  thickness 0.5 m). Alternatively, a static, fixed profile $D_{eBBL}(z)$ may be more appropriate for deep

9  water BBLs, where time dependence may be weak and deepest values from hydrodynamic

10  models may be relatively far above the SWI. In this case, BROM-transport offers two options for $D$-

11  $_{eBBL}(z)$: 1) a constant value, dropping to zero in the DBL, or 2) a linear variation between a fixed

12  value at the top of the BBL and zero at the top of the DBL.  Option 1) defines a simplest-possible

13  assumption, while option 2) corresponds to the assumption of a log layer for the current speed e.g.

14  (Boudreau and Jorgensen, 2001).  Eddy diffusivity is strictly zero in the DBL, on the SWI, and

15  within the sediments. Diffusivity in the sediments is due to molecular diffusion and bioturbation and

16  is parameterized as described in section 2.2.1."

17   Regarding the use of squared availabilities an explanation has been added to section 2.1.2:

18  "The nutrient limitation and heterotrophic transfer functions are based on squared Monod laws for

19  Nutrient/Biomass ratio, which also stabilizes the system compared with Michaelis-Menten and Ivlev

20  formulations."

21  Fortunately, selected results, ideas and formulations can still be gratefully borrowed by  interested

22  colleagues with appropriate reference to the ever available discussion paper.

23  This is true, but we are confident that the revised paper meets all the requirements of a full, published

24  model description paper in GMD.

*Response to the review of G. Munhoven*

**General comments**

Appreciation of the manuscript
In this paper, E. V. Yakushev and co-authors present a highly complex model suitable to study the coupled biogeochemical processes at the bottom boundary layer, the sediment-water interface and the surface sediment. The model appears to provide an extremely complete description, considering all the processes and primary and secondary chemical reactions that have been taken into account. It is integrated into the *Framework for Aquatic Biogeochemical Models*, FABM (Bruggeman and Bolding, 2014).

Although the model appears to have been skilfully designed and set up, the paper has, unfortunately, a number of weaknesses. It is not suitable for publication in *Geosci- entific Model Development* in its current form – it should nevertheless be possible to **reconsider it after a major revision**.

This paper would definitely have benefited from another round or two of rereading and proofreading. Not even the name of the model is unambiguously given: in the title, the name is *Bottom RedOx Model*, in the model presentation (p. 2, ll. 2–3) it is *Bottom RedOx Layer Model*. The English of the paper needs some thorough revision. There are parts that are acceptable and others that are almost unsuitable for review. I am not going to point out all the **English** errors that I found – they are simply to numerous to key them all in here. There is one British co-author and two co-authors with affiliations to institutions in English-speaking countries or regions: could they please have a look at the manuscript and help to correct it and rewrite where necessary! There are errors (spelling, grammar, syntax, style) on nearly every single page, but sections 3.2.4 (Manganese) and 3.3 (Carbonate system) require particularly close attention.

The paper has been submitted as a "model description paper". Requirements for that type of paper are detailed in http://www.geoscientific-model-development.net/about/ manuscript_types.html#item1. Quite some requirements are not met in this paper.

We ask the Reviewer to accept our sincere apologies for the poor condition of the submitted manuscript. We also wish to convey our gratitude to the Reviewer for nevertheless providing a very detailed and constructive review. We feel confident that this review, in combination with the other two, has contributed to a major improvement in the model code and description.

The model description is not well contextualized. The application presented is for a shallow-water environment, but one may ask where else it could be applicable, and which extensions or adaptations would be required or which simplifications would be possible. The authors mention, e. g., a possible coupling to NEMO,which encompasses almost the complete range of marine environments that one can imagine.

The Background section has been extended to improve the model contextualization. The broad applicability of BROM is now clarified in our final paragraph of the Background, where the distinguishing features of BROM vs. other models are listed:

1  "The goal of this work was to develop a model that captures key biogeochemical processes in the
2  water and sediment and to analyze the changes occurring in the BBL and SWI. As a result, BROM
3  differs from existing biogeochemical models in several key respects. BROM features explicit, detailed
4  descriptions of many chemical transformations under different redox conditions, and tracks the fate of
5  several chemical elements (Mn, Fe, S) and compounds ($MnCO_3$, $FeS$, $S0$, $S_2O_3$) that rarely appear in
6  other models.  BROM also allows for spatially explicit representations of the vertical structure in the
7  sediments and BBL. This distinguishes it from e.g. ERSEM (Butenschon et al., 2016) which has a
8  more detailed representation of benthic biology (meiofauna and different types of macrofauna), but
9  limits its chemistry to the dissolved phase to $CO_2$, $O_2$ and macronutrients, and its vertical structure of
10  sediments to an implicit three-layer representation that relies on equilibrium profiles of solutes and
11  idealized profiles of particulates. Third, BROM offers a near-comprehensive representation of all
12  processes affecting oxygen levels in the BBL and sediments, and should therefore provide a useful
13  tool for studies focused on deoxygenation in deep water and sediments.  Finally, BROM is conceived
14  and programmed as a flexible model that can be applied in a broad range of marine and lake
15  environments and modelling problems. As a component of the Framework for Aquatic Biogeochemical
16  Modelling (FABM, Bruggeman and Bolding, 2014), BROM can be very easily coupled online to any
17  hydrodynamic model within the FABM, and can also be driven offline by hydrodynamic model output
18  saved in netCDF or ascii format (using the purpose-built offline transport solver BROM-transport)."

20  The technical details of the implementation are incomplete, and therefore, the criterion of model
21  reproducibility that the paper should aim for is not met. All to many details are not covered in the
22  description.

24  The level of technical detail in the new manuscript has been substantially increased. BROM-transport
25  is now described in much greater detail in the text.  The BROM-biogeochemistry description has been
26  reworked and the Tables now provide an accurate and exhaustive description of all parameterizations
27  and parameter values.  We have added an Appendix guide "Running BROM step-by-step" and have
28  made the input files (netCDF, fabm.yaml, and brom.yaml) for the demonstration run available on the
29  BROM-transport git repository so that these results can be exactly reproduced.  The .yaml input files
30  that contain further technical details of implementation have also been provided as Appendices.

32  1. The instructions about where to get the code are incomplete.  Much guesswork is currently required
33  to locate the relevant files inside the FABM distribution. This could easily be avoided by, say, three to
34  five extra sentences.

35  This section has been improved and extended. We also added an Appendix "Running BROM step-by-
36  step" to help the reader to run the model locally.

38  There does not seem to exist a way to permanently access the precise model version described in
39  this paper.

40  Now we provide a permanent tags for both BROM and FAMB: BROM-transport tag v1.1.
41  https://github.com/e-yakushev/brom-git.git and the BROM-biogeochemistry code in FABM tag v0.95.3
42  http://fabm.net.

43  Also now there is a Win32 executable file available at https://github.com/e-yakushev/brom-
44  git/releases/tag/v1.1

The limitations of the model and the fundamental software requirements are not given: if the model described here is really BROM-transport (this is not a name found in the paper, but it is the name of the only sensible source code collection that I could find), then the paper needs to state right away that:

- the BROM source code can only be compiled with the Intel Fortran compiler for Windows

- the current version can only use hydrodynamic conditions derived from GOTM (according to the Wiki at https://sourceforge.net/p/fabm/wiki/BROM_ FABM).

Although it is reported on p. 16 (l. 5), that the model was run with the Intel compiler for Windows,[1] it is said nowhere that this is the only way to run it. This is obviously a extremely strongly limitation and I am wondering whether such a restriction is

fundamentally necessary. As far as I can see, FABM itself is written in standard- conforming Fortran 2003 in a portable manner (no hardcoded kind types, etc.) and does not seem to rely on a single compiler for a single platform.
* * *
[1]It would be useful to provide the version number of the compiler used. FABM and BROM require some specific Fortran 2003 features and the Intel compiler only offers full support for Fortran 2003 since version 15. However, a subset offered by earlier versions might be sufficient here. **I strongly encourage the authors to prepare a version of the source code that can be used on other platforms and with alternative compilers**. It should be possible to do this quite rapidly by introducing a few pre-processor directives, which would switch off some extra functionality provided by the Intel Fortran compiler for Windows, but that is not fundamentally required for the model itself. This would increase the usefulness of BROM by orders of magnitude! Else, what is the point in emphasizing that the model code "[. . . ] uses modern software standards: it is coded in object-oriented Fortran 2003, [… ]" (p. 27, ll. 17–18) if in the end, it only compiles with one single compiler on one single platform.

The new version of BROM is platform independent and is currently used by the co-authors under both Windows and Linux.

The model itself seems to be carefully designed and set up. There are a few assumptions regarding the physical environment that may be debatable and that would benefit from a few extra words of explanation (see specific comments below). The set of processes and coupled chemical reactions and equilibria that have been taken into account is extremely complex. It is not obvious if such a high degree of complexity is truly necessary. The model indeed seems to allow a rather accurate simulation of the environment chosen. However, to what extent does it contribute to improve our understanding of the way the environment evolves? It would be interesting to know which are the dominant actors of the system. Unfortunately, the paper does not address this kind of question at all.

The physical environment assumed by the offline 1D solver BROM-transport is now described and explained in much greater detail. Regarding model complexity, we state the philosophy behind BROM in the new section 2.1.1:

"The model has 33 state variables (Ci), described in Table 1. This includes frequently measured components such as hydrogen sulfide (H2S) and phosphate (PO4), as well as rarely measured variables such as elemental sulfur (S0), thiosulfate (S2O3), trivalent manganese species Mn(III), and bacteria. Variables of the latter category were included because their contribution to biogeochemical transformations is believed to be substantial. For instance, bacteria play an important role in many modelled processes and can consume or release nutrients in organic and inorganic forms (Canfield et al., 2005; Kappler et al., 2005). We acknowledge that for many of these additional variables, site-specific estimates of associated model parameters and initial/boundary conditions may be difficult or impossible to obtain, and may in practice require some crude assumptions and approximations (e.g. universal default parameter values, no-flux boundary conditions, initial conditions from a steady annual cycle). Nevertheless, we believe that for many applications this will be a price worth paying for the additional process resolution/realism provided by BROM for important biogeochemical processes in the BBL and sediments."

Regarding the contribution to understanding through model analysis: This is a very important message but we believe that it requires a special study that is beyond the scope of the present description paper. However, we do plan to perform such analysis with a model carefully validated to a natural system, as part of a separate publication.

**2  Specific comments**

2.1  Introduction

The scope of the model, i. e., the bottom boundary layer (BBL), (also known as the benthic boundary layer, or are there differences between those two BBLs?) deserves to be presented in more detail. What is its typical thickness? What influences that thickness? How does it change throughout the global ocean? What are the typical gradients across the BBL? Please do not forget that *Geoscientific Model Development* has a broad lectureship.

The BBL is indeed a focus of BROM, but it is not the only one: BROM also focuses on the upper layers of the sediment.  Also, we are anxious not to lengthen the paper too much through extended discussion or literature review. BROM offers a novel and applicable tool to study water column plus sediment biogeochemistry in an integrated way and with a focus on redox chemistry and deoxygenation.  We want readers to be able to quickly assess whether or not BROM will be useful to them, and to have a detailed documentation of the model if they decide to use it.  Finally, BROM is not a specialized BBL model; it is rather a "benthic-pelagic" coupled model for the water column and sediments. As far as BROM is concerned, the BBL is simply a thin layer of calm water separating the "pelagic" water column from the sediments.  The treatment of the BBL in the current version of BROM is quite simple: the vertical diffusivity is either set to a (low) constant value (the simplest assumption) or it increases linearly from the SWI (roughly corresponding to the assumption of a log layer for current speed, Boudreau and Jorgensen, 2001; Holtappels and Lorke, 2011).  With these considerations and the comments of all reviewers in mind we have included the following paragraph in the new Background:

"The BROM model described herein is a fully coupled benthic-pelagic model with a special focus on deoxygenation and redox biogeochemistry in the sediments and Benthic Boundary Layer (BBL).  The BBL is "the part of the marine environment that is directly influenced by the presence of the interface

between the bed and its overlying water" (Dade et al., 2001). Physical scientists tend to prefer the term "bottom boundary layer", but this is largely synonymous with the BBL (Thorpe, 2005). Within BROM, the term BBL is used to refer to the lower parts of the fluid bottom boundary layer where bottom friction strongly inhibits current speed and vertical mixing, hence including the viscous and logarithmic sublayers up to at most a few metres above the sediment. This calm-water layer plays a critical role in mediating the interaction of the water column and sediment biogeochemistry and in determining e.g. near-bottom oxygen levels, yet it remains poorly resolved in most physical circulation models. For BROM we have developed an accompanying offline transport module "BROM-transport" that uses output from hydrodynamic water column models but solves the advection-diffusion-reaction equations for a "full" grid including both water column and sediments. BROM-transport uses greatly increased spatial resolution near to the SWI, and thereby provides explicit spatial resolution of the BBL and sediments."

I am surprised to read that "the Bottom Boundary Layer (BBL) […] is still understudied" (p. 5, l. 22). On my shelf I have the fine book *The Benthic Boundary Layer: Trans- port Processes and Biogeochemistry* (Boudreau and Jørgensen, 2001). It is nearly 15 years old and BBL research has certainly not come to a rest since that book got published. Please reconsider that statement and provide a fair representation of the existing literature.

We have removed this statement. The summary of existing literature in the Background section has also been expanded to provide a fairer representation.

General model presentation

Scope of the model

In the end, it is not entirely clear what the exact scope of BROM is. In the abstract, BROM is introduced as a model for the biogeochemical process in the bottom boundary layer; in the model description though, we read that "[t]he water column considered in our model spans the sea surface (upper boundary) down to user's defined sediment depth [. . . ]". This is to some extent contradicting as this domain largely exceeds the bottom boundary layer. Please clarify.

BROM was never intended to cover only the bottom boundary layer or to exclusively focus on this. We apologize for the lack of clarity in the original submission. In the new manuscript we have harmonized and clarified the stated goals and scope. In the new Abstract we have:

"The goal of this work was to develop a model that captures key biogeochemical processes in the water and sediments and that simulates the changes occurring in the bottom boundary layer and sediment-water interface."

then in the new Background section we have:

"The goal of this work was to develop a model that captures key biogeochemical processes in the water and sediment and to analyze the changes occurring in the BBL and SWI."

1    Computational aspects

3    It is stated that numerical integration was carried out with the Eulerian scheme (the explicit or the
4    implicit variant? – the extremely short time-steps chosen make me guess it is the former, but it would
5    be good to state this). Is the same Eulerian scheme used for both space and time dimensions?
6    Please specify all the schemes used.

8    The numerical integration options of BROM transport are now described in a dedicated section 2.2.2.
9    The new text reads:

11    "Equations (1-3) are integrated numerically over a single combined grid (water column plus sediments)
12    and using the same model time step in both water column and sediments.  All concentrations are
13    stored internally and input/output in units [mmol/m$^3$ total volume]. Time stepping follows an operator
14    splitting approach (Butenschon et al., 2012): concentrations are successively updated by contributions
15    over one time step of diffusion, bioirrigation, reaction, and advection, in that order.  If any state
16    variable has any 'not-a-number' values at the end of the time step then the program is terminated.

17    Diffusive updates are calculated either by a simple forward-time central-space (FTCS) algorithm or by
18    a semi-implicit, central-space algorithm adapted from a routine in the General Ocean Turbulence
19    Model (GOTM, Umlauf et al., 2005).  Bioirrigation and reaction updates are calculated as forward
20    Euler time steps, using the FABM to compute $R_i$, and advection updates are calculated using a simple
21    first-order upwind differencing scheme. After each update, Dirichlet boundary conditions (see below)
22    are reimposed and all concentrations are low-bounded by a minimum value (default = $10^{-11}$ µM) to
23    avoid negative values.

24    BROM-transport also provides the ability to divide the diffusion and advection updates into smaller
25    time steps related to the sources-minus-sinks time step by fixed factors, since the physical transport
26    processes are often numerically limiting (Butenschon et al., 2012).  The default time step is 0.0025
27    days or 216 seconds, which is much longer than the characteristic equilibration timescale of the $CO_2$
28    kinetics (Zeebe and Wolf-Gladrow, 2001)."

30    Details about the pH solving algorithm can only be looked up in the code. The text only says that
31    "[…]  total pH was calculated using the Newton-Raphson method" (p. 11, ll. 20–21) and that
32    "Carbonate system equilibration was parameterized using the standard approach (i.e. Lewis and
33    Wallace, 1998)" (Table 2). This latter affirmation is meaningless: Lewis and Wallace (1998) neither
34    provide information about the methods used in their program, nor do they define any standard
35    approach. A few more details about how calculations are actually done in BROM would be of order
36    here.

37    The carbonate system code was updated, in particular we added dependence of the carbonic acid
38    constants on pressure, and we implemented the pH calculation method proposed by Munhoven
39    (2013).

40    In the new text, the methods for calculating the carbonate system are described in section 2.1.4:

42    "Equilibration of the carbonate system was considered as a fast process occurring within a few seconds

43    (Zeebe and Wolf-Gladrow, 2001). Accordingly, the equilibrium solution was calculated at every time

44    step using an iterative procedure. The carbonate system was described using standard approaches

(Munhoven, 2013; Roy et al., 1993; Wanninkhof, 2014; Wolf-Gladrow et al., 2007; Zeebe and Wolf-Gladrow, 2001). The set of constants of (Roy et al., 1993) was used for carbonic acid. Constants for boric, hydrofluoric, and hydrogen sulfate alkalinity were calculated according to (Dickson, 1992), for silicic alkalinity according to (Millero, 1995), for ammonia alkalinity according to (Luff et al., 2001), and for hydrogen sulfide alkalinity according to (Luff et al., 2001) and (Volkov, 1984). The ion product of water was calculated according to (Millero, 1995). Total scale pH was calculated using the Newton-Raphson method with the modifications proposed in (Munhoven, 2013). Precipitation and dissolution of calcium carbonate were modelled following the approach of (Luff et al., 2001) (Table 2)."

In general, the text really ought to be more complete and informative about **numerical aspects of the model**. This is what *Geoscientific Model Development* readers expect.

We have addressed numerical aspects more thoroughly in the new text, including dedicated sections on numerical integration and numerical details on the carbonate system calculation (see above).

Rate law expressions

The tables that list all the processes considered in BROM and their rate laws, and that collect the different parameter values are among the most informative parts of the paper. They clearly represent one of its major strengths. Unfortunately no references are given for the parameter values presented in Table 3. There is a large variety of kinetic rate laws that are used in the model (Monod laws, squared Monod laws, laws in tanh, ... ). I think it would be good to have a few words of explanation about the choices operated. Please also complete the references where missing (Table 2, on pp. 41–43 and Table 3, throughout).

We have checked and completed the references in Table 2 and added references for the coefficient values in Table 3. Regarding the use of squared availabilities, an explanation has been added to the text:

"The redox-dependent switches are preferably based on hyperbolic functions that improve system stability compared with discrete switches. The nutrient limitation and trophic functions are preferably based on squared Monod laws for Nutrient/Biomass ratio, which also stabilizes the system compared with Michaelis-Menten and Ivlev formulations."

As mentioned in the general appreciation already, I really wonder if all that complexity is really necessary, or, put the other way around: which minimalist set of process would be sufficient to obtain realistic results?

Motivation for the complexity of BROM is provided in the Background section of the new text:

"The goal of this work was to develop a model that captures key biogeochemical processes in the water and sediment and to analyze the changes occurring in the BBL and SWI. As a result, BROM differs from existing biogeochemical models in several key respects. BROM features explicit, detailed descriptions of many chemical transformations under different redox conditions, and tracks the fate of

several chemical elements (Mn, Fe, and S) and compounds (MnCO3, FeS, S0, S2O3) that rarely appear in other models. BROM also allows for spatially explicit representations of the vertical structure in the sediments and BBL. This distinguishes it from e.g. ERSEM (Butenschön et al., 2015), which has a more detailed representation of benthic biology (meiofauna and different types of macrofauna), but limits its chemistry to the dissolved phase to CO2, O2 and macronutrients, and its vertical structure of sediments to an implicit three-layer representation that relies on equilibrium profiles of solutes and idealized profiles of particulates. Third, BROM offers a near-comprehensive representation of all processes affecting oxygen levels in the BBL and sediments, and should therefore provide a useful tool for studies focused on deoxygenation in deep water and sediments. "

Further explanation of the BROM philosophy regarding model complexity has also been added to section 2.1.1:

"The model has 33 state variables (Ci), described in Table 1. This includes frequently measured components such as hydrogen sulfide (H2S) and phosphate (PO4), as well as rarely measured variables such as elemental sulfur (S0), thiosulfate (S2O3), trivalent manganese species Mn(III), and bacteria. Variables of the latter category were included because their contribution to biogeochemical transformations is believed to be substantial. For instance, bacteria play an important role in many modelled processes and can consume or release nutrients in organic and inorganic forms (Canfield et al., 2005; Kappler et al., 2005). We acknowledge that for many of these additional variables, site-specific estimates of associated model parameters and initial/boundary conditions may be difficult or impossible to obtain, and may in practice require some crude assumptions and approximations (e.g. universal default parameter values, no-flux boundary conditions, initial conditions from a steady annual cycle). Nevertheless, we believe that for many applications this will be a price worth paying for the additional process resolution/realism provided by BROM for important biogeochemical processes in the BBL and sediments."

Miscellanea

Denitrification is considered, and nitrification, but I could not find anything about how nitrogen fixation is dealt with. I would expect that this process is required to avoid an unrealistic drift in the nitrogen inventory.

Corrected.

Total alkalinity

This part of the paper (p. 9) is one of the most disappointing ones. It is very approxi- mate, completely overloaded with information that is ignored in the end. It furthermore contains several errors.

For clarity, it would be best to provide immediately the approximation actually used in the model, and not a hypothetical one, that could have been used. Alkalinity contribu- tions that are not included or that are set to zero should be omitted. The text will be considerably simplified.

Whatever the expression chosen for total alkalinity, it will anyway always remain only an approximation. But even approximations need to be factually correct. Unlike written in the paper, .

- ... H3PO4 is also part of alkalinity and $A_{TPO4} = [HPO_4] + 2[PO_4] - [H_3PO_4]$ — interestingly this is correct in Table 6 (except for a typo) and also in the code;

- ... $NH^+$ is not part of alkalinity (it is the zero-level species) and thus $A_{TNH3}$ = [NH3];

- ... it is the total borate concentration that is estimated from salinity and not [B(OH)$_4^-$] — [B(OH)$_4^-$] is calculated from the state variables just like to others (this is correctly done in the code, fortunately);

- ... F$^-$ is not part of alkalinity, only HF, so that $A_{THF}$ = [HF] — this is also wrong in Table 6 (at 68 μM, it would be barely negligible), but I suggest to discard the $A_{THF}$ term from the alkalinity expression anyway, as it is not included in the model.

-

We apologize for these errors and lack of clarity in the submitted text. The total alkalinity formulation has now been corrected. We have chosen to retain the more general expression for total alkalinity as a starting point and then explicitly neglect the hydrogen sulfate, hydroflouric and nitrous acid terms. We feel that this helps to link our approach with the "classical" formulation.

Although it is specified later on that the stoichiometric constants of Roy et al. (1993) are used for the carbonate system, references for the other constants (e. g., dissociation constants for boric, phosphoric and silicic acids) required to solve the total alkalinity-pH equation are missing. Please provide references for those as well.

The references are now provided in the code (fabm.yaml) and the text:

"The set of constants of (Roy et al., 1993) was used for carbonic acid. Constants for boric, hydrofluoric, and hydrogen sulfate alkalinity were calculated according to (Dickson, 1992), for silicic alkalinity according to (Millero, 1995), for ammonia alkalinity according to (Luff et al., 2001), and for hydrogen sulfide alkalinity according to (Luff et al., 2001) and (Volkov, 1984). The ion product of water was calculated according to (Millero, 1995)."

Finally, the pH scale used in the paper turns out to be the total scale. This should be stated more clearly than it is currently done (at my third reading, I discovered on p. 11 (l. 20) that "total pH was calculated". Please state this more obviously.

We now specify total scale pH twice when the variable is first mentioned in section 2.1:

"Instead, the total scale pH is calculated as a diagnostic variable at every time step as a function of DIC and Alk (which are state variables). In turn, the total scale pH is used in calculations of the chemical equilibrium constants required to describe related processes (i.e. carbonate precipitation/dissolution, carbonate system parameters etc.)."

Physical environment

Porosity

Variable porosity is not included in the current version of BROM. The affirmation that "[. . . ] its effect on [the] vertical transport is incorporated in[to] the values of $K_z$ and $K_{z_{bio}}$, [... ]" (p. 13, l. 17) is rather obscure. $K_{z_{mol}}$ is actually constant so it is not clear how it could take porosity variations into account. I am furthermore not certain that this simplification is really necessary, given the complexity and detailed representation of the rest of the model. Variable porosity should not significantly increase the model's compleity.Furthermore, it appears that a tortuosity corresponding to the porosity value of 90%

was used, with reference to a "value from Boudreau, 1997" (p. 13, l. 22). This is not very meaningful. Boudreau (1997) lists eight theoretically based tortuosity-porosity relationships and three empirical ones. Please specify which one was used here and then cite the original reference.

We agree: this simplification was excessive. BROM-transport has received a major overhaul and now includes variable porosity as a fixed profile following the parameterization of Soetaert et al. (1996). Porosity now distinguishes the solute from particulate dynamics within the sediments assuming intraphase mixing (Boudreau 1997; see section 2.2.1); its effects are now treated explicitly and not folded into the vertical diffusivity (which in fact cannot fully account for porosity variations). The apparent or effective molecular diffusivity now varies with depth due to variable tortuosity. This is described in the new section 2.2.1:

"The total solute diffusivity $D_C = D_m + D_B$, where $D_m$ is the apparent molecular/ionic diffusivity and $D_B$ is the bioturbation diffusivity due to animal movement and ingestion/excretion. The apparent molecular diffusivity $D_m(z) = \theta^{-2} D_0 \frac{\mu_0}{\mu_{sw}}$ is derived from the infinite-dilution molecular diffusivity $D_0$ (an input parameter) assuming a constant relative dynamic viscosity $\frac{\mu_0}{\mu_{sw}}$ (default value 0.94, cf. Boudreau 1997, Table 4.10) and a tortuosity parameterized as: $\theta^2 = 1 - 2\ln\varphi$ from Boudreau (1997) Eqn. 4.120."

Boudreau (1997) is actually the original reference for this tortuosity parameterization. Boudreau himself refers to it as a "modified Weissberg" relation (Boudreau, 1997; Eqn. 4.119) but the empirical fit of the constant "b" is due to Boudreau (Boudreau, 1997; Table 4.12, Fig. 4.10).

Molecular diffusion

BROM uses a species-independent molecular diffusion coefficient. This consider- ably simplifies the advection-diffusion-reaction equations, as the total concentrations a, such as DIC and alkalinity can be transported directly. The reported value $K_{zmol} = 1 \times 10^{-11}$ m$^2$s$^{-1}$ is, however, almost two orders of magnitude lower than those for typ- ical ions: e. g., from Boudreau (1997, Table 4.8), we may calculate diffusion coeffi- cient values of $0.781 \times 10^{-9}$ m$^2$s$^{-1}$ for HCO$^-$, $0.632 \times 10^{-9}$ m$^2$s$^{-1}$ for CO$^{2-}$ and even
3                                                             3
$1.313 \times 10^{-9}$ m$^2$s$^{-1}$ for HS$^-$ (each one for $t = 10\,°$C). These are infinite dilution diffusion

coefficients, but correcting them for tortuosity and for the dynamic viscosity of seawa- ter does not reduce these values by more than 15–20%. How would results change if these much higher values would be used?

We agree that the species-independent molecular diffusivity is a simplification, but as the reviewer states it does substantially simply matters in regard to composite variables. We have retained the species independence in the new code, although as it is now written the user would only have to make small modification to the code to allow species dependent diffusivity ($K_{zmol}$ in the new code is actually stored as a matrix over depth and state variable, with zeros for particulate variables). We agree that the previous default value was too small, even if assumed to account for tortuosity and dynamic viscosity. In the new version, the default (single) value for infinite-dilution molecular diffusivity is $1 \times 10^{-9}$ m$^2$s$^{-1}$ based on the coefficients in Boudreau (1997, Table 4.8) (see brom.yaml in Appendix D). The user is free to change this parameter in the run-time brom.yaml file, where we also state the default value as well as a plausible range $(0.5-2.7) \times 10^{-9}$ m$^2$s$^{-1}$ again derived from Boudreau (1997, Table 4.8). The default value of $1 \times 10^{-9}$ m$^2$s$^{-1}$ was used for the demonstration simulation in the new section 3 (see Appendix D).

2.2.3 Bioturbation
Biotubation is parametrized as a diffusive process, as is common usage. For the biod- iffusion coefficient, it is only stated that it takes a constant value over the top 2 cm and that it decreases

exponentially afterwards. However, I have not been able to find the length scale of this decrease anywhere in the text. Now, one may ask whether it is re- alistic to consider any bioturbation at all in anoxic parts of the sediment, the more since the text already indicates that the maximal bioturbation depth was only 0.5–2.2 cm (p. 13, ll. 15–16). How would this change your conclusions?

We apologize for making the reviewer search to no avail. The exponential decay scale is a user-defined parameter defined in the brom.yaml file (see Appendix D). Here we specify a default value of 1 cm, citing Soetaert and Middelburg (2009). In anoxic conditions, the entire profile of bioturbation diffusivity is scaled down by a Michaelis Menten function of oxygen concentration at the sediment surface. This is described in the new section 2.2.1:

"The bioturbation diffusivity $D_B(z,t)$ is modelled as a Michaelis-Menten function of the dissolved oxygen concentration in the bottom layer of the water column:

$$D_B(z,t) = D_{Bmax}(z) \frac{O_{2s}}{O_{2s}+K_{O2s}}$$
(5)

where $D_{Bmax}(z)$ is a constant over a fixed mixed layer depth in the surface sediments then decays to zero with increasing depth, and $K_{O2s}$ is a half-saturation constant. The rationale for (5) is that the animals (worms etc.) that cause bioturbation require a source of oxygen at the sediment surface for respiration."

Bioirrigation

BROM takes the important process of bioirrigation into account. It is, however, represented as a purely diffusive process. Boudreau (1997) and Aller (2001) make a strong case that it would be more appropriate to represent bioirrigation as a non-local exchange process instead.

The simple possible parameterization, probably it is not enough… Of cource there are other approaches, but …

We agree. A more thorough examination of the literature shows little theoretical or observational support for a local diffusive parameterization of bioirrigation. In the new BROM-transport model, bioirrigation is modelled as a non-local exchange process as proposed in Boudreau (1997). This is described in the new section 2.2.1:

"Finally, the process of bioirrigation, whereby worms flush out their burrows with water from the sediment surface, is modelled as a non-local solute exchange following (Meile et al., 2001; Rutgers Van Der Loeff and Boudreau, 1997; Schlüter et al., 2000):

$$T_{birrC(i)} = \alpha \varphi \frac{O_{2s}}{O_{2s}+K_{O2s}} \left( \hat{C}_{f(i)} - C_i \right)$$
(for solutes)
(10)

where $\alpha(z)$ is the bioirrigation rate in oxic conditions, $\hat{C}_{f(i)}$ is the flushing concentration of solute in the fluff layer, and the Michaelis-Menten function again accounts for the suppression of worm activity in anoxic conditions. The oxic bioirrigation rate $\alpha(z)$ is parameterized as an exponential decay from the

sediment surface as in Schluter et al. (2000).  The total mass transfer to/from the sediment column must be balanced by a flux into/out of the fluff layer (see equation (1)):

$$T_{birr(i)} = \frac{1}{h_f} \frac{O_{2s}}{O_{2s}+K_{O2s}} \int_{z_{SWI}}^{z_{max}} \alpha\varphi\left(C_i - \hat{C}_{f(i)}\right) dz' \qquad\qquad \text{(for} \qquad \text{solutes)}$$

(11)

where $h_f$ is the thickness of the fluff layer and $z_{max}$ is the depth of the bottom of the modelled sediment column. $T_{birrC(i)}, T_{birr(i)} = 0$ for all particulate variables."

Code
On p. 7 (ll. 24–25), it is said that BROM consists of three modules. I did not want to download and install the complete FABM, but nevertheless wanted to inspect the BROM code, to find out more about the technical details that were missing from the paper. This was, however, not entirely straightforward.

Accessibility
After having opened http://fabm.org (which redirects to the FABM project page on SourceForge), I started to search for references to BROM. After some searching around, I detected the first trace of BROM under the "Wiki" tab: section 7 of chap- ter 2 of the User's Guide has the title "BROM-transport + FABM". BROM-transport is most probably the transport model mentioned in the paper (p. 7, l. 2), but that is not clear, since the paper always mentions BROM only. That section provides at least the first useful hint about where to find the BROM biogeochemical modules: under src/models/niva/brom in FABM. Proceeding to the "Code" tab then allowed me to browse to the relevant files (under the indicated directory tree). BROM-transport, however, is not with FABM and must be retrieved from a different repository, located at https://github.com/e-yakushev/BROM-transport, not mentioned in the paper. I suggest that the authors give accurate and comprehensive instructions in the paper

about where the actual BROM source code files are located, both the biogeochemical ones and the main driver. And, please include also information about the license under which the code is distributed.

We apologize for the confusion and wild goose chase.  In the new text we clarified and extended the section "Code Availability".  It now reads:

"The model as presented consists of two components. The first is a set of biogeochemical modules (brom/redox, brom/bio, brom/carb, brom/eqconst), available as part of the official FABM distribution (http://fabm.net) (for a currently-functional direct link please see https://sourceforge.net/p/fabm/code/ci/master/tree/src/models/niva/brom/). The second is a hydrophysical driver (BROM-transport) that provides the 1D vertical context and resolves transport; this is available separately from https://github.com/e-yakushev/brom-git.git. When combined, the 1D BROM model as presented is obtained. Additionally, as BROM's biogeochemical modules are built on FABM, they can be used from a wide range of 1D and 3D hydrodynamic models, including GOTM, GETM, MOM, NEMO and FVCOM (NEMO-FABM and FVCOM-FABM couplers have been developed by the Plymouth Marine Laboratory; contact J.B. for information).

BROM biogeochemical modules follow FABM conventions: they are coded in object-oriented Fortran 2003, have a build system based on CMake, and use YAML files for run-time configuration. The code is platform independent and only requires a Fortran-2003-capable compiler, e.g., gfortran 4.7 or

higher, or the Intel Fortran compiler version 12.1 or higher. The BROM-specific source code is located in the FABM code tree in directory src/models/niva/brom.  The specific version used to produce the results described in this paper is associated with git commit 1581186939a0ff81a230468694bf909a42afc21e. However, we envisage the model to be further developed in a backward compatible manner, and encourage users to use the latest code version.

BROM-transport is coded in Fortran 2003. It includes facilities for producing results as NetCDF files, which can be read by a variety of software on different platforms. The reader should be able to reproduce the results shown in this paper using the BROM-transport and BROM-biogeochemistry code from the above repositories and the netCDF/.yaml input files found in the data/ folder of the BROM-transport repository.  Step-by-step instructions for running BROM are found in Appendix A. BROM-transport as well as BROM biogeochemical modules are distributed under the GNU General Public License (http://www.gnu.org/licenses/)."

Code quality

The code is obviously "work in progress" and appears to undergo continuous changes. There are many lines of code that are commented out, some of them might be important. It is of not clear if they were also commented out when the results described in the paper were calculated.

The code quality and presentation have undergone a major overhaul.  We have uploaded a finished, stable version with all commented-out code deleted.  The reader should be able to reproduce the results shown in this paper using the BROM-transport and BROM-biogeochemistry code from the above repositories and the netCDF/.yaml input files found in the data/ folder of the BROM-transport repository (these .yaml files are also shown in Appendices C and D).  Step-by-step instructions for running BROM are found in Appendix A.

I detected a few coding choices that put portability at risk. While REALs in the three biogeochemistry related modules are declared in a portable way  with  REAL(rk), where rk is an INTEGER parameter whose value gets derived from an appropriate SELECTED_REAL_KIND(...) call, there are some INTEGER(4) declarations that may lead to problems. In BROM-transport, there are numerous REAL(8) declarations, in different source code files. Kind type values – such as the '4' of the INTEGER(4) or the '8' of the REAL(8) declarations – are not standardized and may differ from one compiler to another. Programmers may not assume that they are equal to the ex- pected byte length and for portability reasons kind type values must therefore not be hard-coded.[2] Portable and reliable code would consistently follow the FABM approach,  with the rk parameter derived from SELECTED_REAL_KIND(...)

All REAL(8)declarations in BROM-transport have now been changed to REAL(rk)where rk is inherited from the  SELECTED_REAL_KIND(...)statement in the FABM code, using a command: use fabm_types, only: rk.  All INTEGER(4)declarations have been replaced with INTEGER.

I have come across a few peculiarities or short-cuts in the code that may lead to seri- ous confusion: e. g., in the subroutine phIter in brom_carb.F90, the INTENT(IN) argument Sit_ (the total silicate concentration) is overridden by a local variable Sit,

[2]I know of one compiler where DOUBLE PRECISION is not REAL(8) but REAL(3).

which is set to zero, thus making the code ignore silicate alkalinity. The paper does, however, not state that silicate alkalinity is ignored.

The code has been significantly modified and the mentioned peculiarities have been removed.

The pH calculation routine is neither safeguarded nor does it include diagnostics for possible convergence failures or for early convergence: it simply executes 100 Newton- Raphson iterations, starting from a preset fixed starting value, that furthermore seems to require manual modification from time to time. No diagnostic is included, neither for possible convergence failures nor for early convergence. (Why carry out 100 iterations if convergence is reached after five of them already?)

There are now reliable methods to solve the alkalinity-pH equation, which are guaranteed to converge under any physically meaningful conditions, howsoever exotic, and usually in less than six iterations (**Munhoven**, 2013). These would be particularly recommended in the environments that BROM has been developed for, with complex alkalinity compositions and unusual total concentrations.

We are very grateful to the Reviewer for this suggestion, and have implemented the recommended methods to solve the alkalinity-pH equation. This has been very helpful in regard to computational efficiency.

Carbonate solubility constants do not take any pressure correction into account (the relevant lines are present, but commented out).

Corrected.

Finally, the comments in the code are not always correct, which also creates unnecessary confusion (e. g., the phosphoric alkalinity is not [H2PO4-] + 2.*[HPO4--] + 3.*[PO4-

--] as stated in a comment, but [HPO4--] + 2.*[PO4---] - [H3PO4]. Fortunately it is the latter that is implemented in the code.

Corrected.

Permanent access to the code for model version 1.0

As mentioned in the general appreciation, for model description papers there should exist a way to permanently access the precise model version described in the paper. The GitHub repository for BROM-transport includes a Ver. 1.0 directory, so for the transport model, this seems to be conceivable. The biogeochemical modules that are hosted in the FABM repository are however not clearly tied to version 1.0 of BROM.

It would thus be necessary to provide somehow tagged versions of the source code files for the model version 1.0 described here, or to provide copies of those files as a supplement to the paper.

2 The tag 1.1 for BROM transport is provided, https://github.com/e-yakushev/brom-git/releases/tag/v1.1

4 2.6 Tables

6 The tables contain a wealth of information and represent one of the most useful parts of the paper
7 (with the exception of Table 6, which could be deleted without loss). Un- fortunately, Tables 1 and 4
8 are nearly unreadable because of the small font size. They would clearly benefit from a
9 reorganization of their contents. Table 2 currently spans eight pages, Table 3 six pages. It would be
10 useful to split them into smaller parts, with dedicated captions. While Table 2 still contains extensive
11 references, Table 3 does not contain a single one. Readers ought to know where the adopted
12 parameter values come from or how they have been derived.

14 The tables have been modified following the Reviewer's suggestions.

16 The second column of the row "Alkalinity changes" in Table 2 is completely overloaded. Please
17 reorganize this information.

19 Corrected.

20 Table 6 is not essential for the paper and I suggest to delete it altogether. It also contains errors
21 and except for Canfield et al. (2005), none of the references cited is in the reference list. $A_{THF}$ is
22 certainly not 68 $\mu$M, else it would not be negligible.

24 Table 6 has been deleted.

25 **3 Technical comments**

27 Throughout the paper: change "protolithic" to "protolitic" or "equilibrium" (depending on the context)

28 Corrected.

30 Throughout the paper: change "connected with" to "related to"

31 Corrected.

33 Throughout the paper: please check the usage of the word "parameterized" and "pa-
34 rameterization". For example, in Table 2, it is said that the carbonate system equilibra- tion was
35 parameterized. It were rather the stoichiometric constants that were parame- terized, as a function
36 of temperature, salinity and pressure, but the carbonate system equilibration (it would be more
37 correct to say speciation) was calculated.

38 Corrected.

p. 4, l. 26: "death or flight"? "death or migration" would perhaps be more appropriate

Corrected.

p. 7, l. 15: "changeable" is not appropriate in this context. Perhaps "varying"?

Corrected.

p. 9, ll. 20–25: it is common usage to speak about bor*ate*, phosph*ate* and silic*ate* alkalinity (as with *carbonate alkalinity*) and to reserve the terms bor*ic*, phosphor*ic* and silic*ic* for the corresponding acids (as in *carbonic acid*).

Corrected.

p. 11, l. 20: change "Roy's constants" to "the set of constants of Roy et al." – the co-authors will appreciate

Corrected.

p. 16, ll. 4–5: change "FORTRAN" to "Fortran 2003" (spelling and standard) and change "Intel FORTRAN for Windows Compiler" to "Intel Fortran Compiler for Win- dows", which is the name of the product.

Corrected.

p. 16, l. 6: what is meant by "balanced distribution"?

We meant balanced fluxes in a quasi-stationary sense. This term has been deleted.

pp. 21–26 (section 3.2.4 – section 3.4): please check for the English and rewrite where necessary.

This has been done.

p. 39, rows 10 and 11: "sulfatereduction" should read "sulfate reduction"

Corrected.

p. 40, second-last row, right-hand column: should the "CaCO$_3$" on the last linee not read "caco3_diss−caco3_prec"?

Corrected. Note that in the new Table 2.3, "caco3_prec" has been replaced with "caco3_form".

p. 41, row 7: there is probably some "NO3"-"NH$_3$" mismatch here

Corrected. The correct equation reads:

$$\text{LimNO}_3 = \frac{((\,\text{NO}_3 + \text{NO}_2)/\text{Phy})^2}{\text{K\_nox\_lim}^2 + ((\,\text{NO}_3 + \text{NO}_2)/\text{Phy})^2} \exp\left(-\text{K\_psi}\frac{(\text{NH}_4/\text{Phy})^2}{\text{K\_nh4\_lim}^2 + (\text{NH}_4/\text{Phy})^2}\right)$$

p. 41, rows 7 and 8: the two trailing '2's in exponent seem to be misplaced (they probably belong to the second term in the denominator each time)

Corrected. Please forgive our sloppy editing.

p. 46, in the first row relative to a half-saturation for OM denitrification, "NO$_2$" should probably read "NO$_3$"

Corrected.

Table 6: "[PO$^{2-}_4$]" should read "[PO$^{3-}_4$]"

This table has been deleted.

$2MnO_2 + 7H^+ + HS^- \square + H^{3+} + 4H_2O + S^0$ | $mn\_rd = 0.5 * \left(1 + \tanh(Mn^{4+} - s_{mnrd_{mn4}})\right) * K_{mn_{rd}} * Mn^{4+} * \dfrac{H_2S}{(H_2S + k_{mnrdHS})} mn\_rd1$
$= 0.5 * \left(1 + \tanh(Mn^{4+} - s\_mnrd\_mn4)\right) * K\_mn\_rd1 * Mn^{4+}$
$* \dfrac{H_2S}{(H_2S + K\_mnrd\_hs)}$ |
| Manganese (III) reduction
$2Mn^{3+} + HS^- \square + HS^{2+} + S^0 + H^+$ | $mn\_rd2 = 0.5 * \left(1 + \tanh(Mn^{3+} - s_{mnrd_{mn3}})\right) * K_{mn_{rd2}} * Mn^{3+} * \dfrac{H_2S}{(H_2S + k_{mnrdHS})} mn\_rd2$
$= 0.5 * \left(1 + \tanh(Mn^{3+} - s\_mnrd\_mn3)\right) * K\_mn\_rd2 * Mn^{3+}$
$* \dfrac{H_2S}{(H_2S + K\_mnrd\_hs)}$ |

| MnS formation/dissollution (Davison, 1993) | $mns\_prec = K_{mns_{form}} * max\{0, (om\_mns - 1)\}$ |
| MnS formation/dissollution (Davison, 1993) : $Mn^{2+} + HS^- \leftrightarrow MnS + H^+$ | $mns\_form = K\_mns\_form * max(0, \left(\frac{H_2S * Mn^{2+}}{K\_mns * H^+} - 1\right))$ |
| | $mns\_diss = K_{mns_{diss}} * MnS \, K\_mns\_diss * MnS * max\{(0, (1 - om\_mns)\}$ |
| | $where \, om\_mns = \frac{H_2S*Mn^{2+}}{K_{mns}*H^+}\left(1 - \frac{H_2S*Mn^{2+}}{K\_mns*H^+}\right)$ |
| $MnCO_3$ precipitation/dissolution (Van Capellen, Wang, 1996) $Mn^{2+} + CO_3$ (Van Cappellen and Wang, 1996): $Mn^{2+} + CO_3^{2-} \leftrightarrow \square nCO_3$ | $mnco3\_prec = K_{mnco3_{form}} * K\_mnco3\_pres * max\{(0, (om\_mnco3 - 1)\} \left(\frac{Mn^{2+} * CO_3}{K\_mnco3} - 1\right))$ |
| | $mnco3\_diss = K_{mnco3_{diss}} * MnCO_3 * K\_mnco3\_diss * MnCO_3 * max\{(0, (1 - om_{mnco3})\}$ |
| | $where \, om_{mnco3} = \frac{Mn^{2+}*CO_3}{K_{mnco3}}, \left(1 - \frac{Mn^{2+}*CO_3}{K\_mnco3}\right)$ |
| $MnCO_3$ oxidation by $O_2$ (Morgan, 2005) $MnCO_3$ oxidation by $O_2$ (Morgan, 2000): $2 \, MnCO_3 + O_2 + 2H_2O = \square \, 2 \, MnO_2 + 2HCO_3^- + 2H^+$ | $mn\_co3\_ox = K_{mnco3_{ox}} * MnCO_3 * O_2 \, mnco3\_ox = K\_mnco3\_ox * MnCO_3 * O_2$ |
| Manganese reduction for PON (Boudreau, 1996): $(CH_2O)_{106}(NH_3)_{16}H_3PO_4 + 212MnO_2 + 318CO_2 + 106H_2O \rightarrow 424HCO_3^- + 212 \, Mn^{2+} + 16NH_3 + H_3PO_4$ | $DcPM\_Mn = max\{0, K_{PON_{Mn}} * PON * \frac{Mn^{4+}}{Mn^{4+}+0.5}\} * (K\_PON\_mn * PON * \frac{Mn^{4+}}{Mn^{4+}+0.5} * (1 - 0.5$ $* (1 + \tanh(O_2 - O_2 s_{dn}))(O_2 - O2s\_dn))$ |
| Manganese reduction for DON (Boudreau, 1996): | $DcDM\_Mn = max\{0, K_{PON_{Mn}} * DON * \frac{Mn^{4+}}{Mn^{4+}+0.5}\} * (K\_DON\_mn * DON * \frac{Mn^{4+}}{Mn^{4+}+0.5} * (1 - 0.5 * (1 + \tanh(o2 - O_2 s_{dn})))(O_2 - O2s\_dn))$ |

| Iron | |
|---|---|
| Fe (II) oxidation with $O_2$ (Van Cappellen and Wang, 1996): $4Fe^{2+} + O_2 + 10H_2O \rightarrow 4Fe(OH)_3 + 8H^+$ | $fe\_ox1 = 0.5 * (1 + \tanh(Fe^{2+} - s\_feox\_fe2)) * K\_fe\_ox1 * O_2 * Fe^{2+}$ |
| Fe (II) oxidation with Mn oxide (Van Cappellen and Wang, 1996): $2Fe^{2+} + MnO_2 + 4H_2O \rightarrow \square\square\square\square\square\square\square\square\square_3 + Mn^{2+} + 2\,H^+$ | $fe\_ox2 = 0.5 * \left(1 + \tanh(Fe^{2+} - s\_feox\_fe2)\right) * K\_fe\_ox2 * Mn^{4+} * Fe^{2+}$ |
| Fe (III) reduction (Volkov, 1984): $2Fe(OH)_3 + HS^- + 5H^+ \square 5H(O^{2+} + S^0 + 6H_2O$ | $fe\_rd = 0.5 * (1. + \tanh(Fe^{3+} - s\_feox\_fe3)) * K\_fe\_rd * Fe^{3+} * \dfrac{H_2S}{H_2S + K\_ferd\_hs}$ |
| FeS formation/dissolition (Bektursunova and L'Heureux, 2011) : $Fe^{2+} + HS^- \leftrightarrow FeS + H^+$ | $fes\_prec = K\_fes\_form * \max\left(0, \left(\dfrac{H_2S * Fe^{2+}}{K\_fes * H^+} - 1\right)\right)$

 $fes\_diss = K\_fes\_diss * FeS * \max(0, (1 - om\_fe))$

 $\text{where } om_{fes} = \dfrac{H_2S * Fe^{2+}}{K\_fes * H^+}\left(1 - \dfrac{H_2S * Fe^{2+}}{K\_fes * H^+}\right)$ |
| FeS oxidation (Soetaert et al., 2007): | $fes\_ox = K\_fes\_ox * O_2 * FeS$ |

| | |
|---|---|
| $FeS + 2.25O_2 + 2.5H_2O \rightarrow \square Fe(OH)_3 + 2H^+ + SO_4^{2-}$ | |
| Pyrite formation (Rickard and Luther, 1997; Soetaert et al., 2007): $FeS+H_2S \square\square FeS_2 +H_2$

  | $fes2\_form = $  $K\_fes2\_form * \mathbf{H_2S * FeS}$ |
|

 Pyrite oxidation by $O_2$ (Wijsman et al., 2002): $FeS_2+3.5O_2+H_2O\square H^{2+} +2SO_4^{2-} + 2H^+$ | $fes2\_ox = $  $fes2\_ox * \mathbf{FeS_2 * O_2}$ |
| $FeCO_3$ precipitation/dissolution (Van Cappellen and Wang, 1996):
 $Fe^{2+}+CO_3^- \leftrightarrow \square FeCO_3$ | $feco3\_form = K\_feco3\_form * \max\left(0, \left(\frac{\mathbf{Fe^{2+} * CO_3}}{K\_feco3} - 1\right)\right)$

 $feco3\_diss = K\_feco3\_diss * \mathbf{FeCO_3} * \max\left(0, \left(1 - \frac{\mathbf{Fe^{2+} * CO_3}}{K\_feco3}\right)\right)$ |
| $FeCO_3$ oxidation by $O_2$ (Morgan, 2000):
 $2 FeCO_3 + O_2 + 2H_2O \square 2 FeO_2 + 2HCO_3^- + 2H^+$ | $feco3\_ox = K\_feco3\_ox * \mathbf{FeCO_3 * O_2}$ |
| Iron reduction for DON (Boudreau, 1996):
 $(CH_2O)_{106}(NH_3)_{16}H_3PO_4 + 424\,Fe(OH)_3 + 742CO_2 \rightarrow$
 $848HCO_3^- + 424\,Fe^{2+} + 318\,H_2O + 16NH_3 + H_3PO_4$ |  $DcDM\_Fe$

 $= K\_DON\_fe * \mathbf{DON * Fe^{3+}} * \left(1. - 0.5 * \left(1 + \tanh(\mathbf{O_2} - O2s\_dn)\right)\right)$ |
| Iron reduction for PON (Boudreau, 1996): |  $DcPM\_Fe$

 $= K\_PON\_fe * \mathbf{PON * Fe^{3+}} * \left(1. - 0.5 * \left(1 + \tanh(\mathbf{O_2} - O2s\_dn)\right)\right)$ |
|  | |

| | |
|---|---|
|  $$NH_4^+ + 1.5O_2 \rightarrow NO_2^- + 2H^+ + H_2O$$ |  |
|  $$NO_2^- + 0.5\,O_2 \rightarrow NO_3^-$$ |  |
|  $$NO_2^- + NH_4^+ \rightarrow N_2 + 2H_2O$$ |  |
|  $$0.5CH_2O + NO_3^- \rightarrow NO_2^- + 0.5H_2O + 0.5CO_2$$ |     |

| | |
|---|---|
| POM and DOM denitrification (2d stage) (Anderson et al., 1982)

 $0.75CH_2O + H^+ + NO_2^- \rightarrow 0.5N_2 + 1.25H_2O + 0.75CO_2$ | $denitr2\_PM = K_{NZ4} * PON * Fdonx * \dfrac{NO_z}{NO_z + K_{omno\_NOz}}$

 $denitr2\_DM = K_{NZ4} * DON * Fdnox * \dfrac{NO_z}{NO_z + K_{omno\_NOz}}$

 $denitr2 = denitr2\_PM + denitr2\_DM$ |
| Denitrification of POM and DOM (Richards, 1965)

 $(CH_2O)_{106}(NH_3)_{16}H_3PO_4 + 84.8HNO_3 = 106CO_2 + 42.4N_2 + 148.4H_2O + 16NH_3 + H_3PO_4$ | $DcPM\_NOX = \dfrac{16}{212} * Denitr1\_PM + \dfrac{16}{141.3} Denitr2\_PM$

 $DcDM\_NOX = \dfrac{16}{212} * Denitr1\_DM + \dfrac{16}{141.3} Denitr2\_DM$ |
| **Sulfur** | |
| $S^0$ disproportionation (Canfield et al., 2005):
 $4S^0 + 3H_2O \rightarrow 2H_2S + S_2O_3^{2-} + 2H^+$ | $disprop = K_{dispro} * S^\theta$ $s0\_disp = K\_s0\_disp * \mathbf{S^0}$ |
| Sulphide oxidation with $O_2$ (Volkov, 1984):

 Sulphide oxidation with $O_2$ (Volkov, 1984):
 $2H_2S + O_2 \rightarrow 2S^0 + 2H_2O$ | $hs\_ox = K_{HS_{ox}} * H_2S * O_z$ $K\_hs\_ox * \mathbf{H_2S} * \mathbf{O_2}$ |
| $S^\theta$ oxidation with $O_2$ (Volkov, 1984):

 $S^0$ oxidation with $O_2$ (Volkov, 1984):
 $2S^0 + O_2 + H_2O \rightarrow S_2O_3^{2-} + 2H^+$ | $s0\_ox = K_{S^\theta\_ox} * S^\theta * O_z$ $K\_s0\_ox * \mathbf{S^0} * \mathbf{O_2}$ |
| $S^\theta$ oxidation with $NO_3$ (Kamyshny et al., 2013): | $s0\_no3 = K_{S^\theta_{NO_3}} * NO_3 * S^\theta$ $K\_s0\_no3 * \mathbf{NO_3} * \mathbf{S^0}$ |

| | |
|---|---|
| $\underline{S^0}$ oxidation with $NO_3$ (Kamyshny et al., 2013):
$4S^0 + 3NO_3^- + 7H_2O \rightarrow 4SO_4^{2-} + 3NH_4^+ + 2H^+$ | |
|

$\underline{S_2O_3}$ oxidation with $O_2$: (Volkov, 1984):
$S_2O_3^{2-} + 2O_2 + 2OH^- \rightarrow 2SO_4^{2-} + H_2O$ | $s2o3\_ox = $  $K\_s2o3\_ox * \mathbf{S_2O_3} * \mathbf{O_2}$ |
|

$\underline{S_2O_3}$ oxidation with $NO_3$: (Kamyshny et al., 2013)
$S_2O_3^{2-} + NO_3^- + 2H_2O \rightarrow 2SO_4^{2-} + NH_4^+$ | $s2o3\_no3 = $  $K\_s2o3\_no3 * \mathbf{NO_3} * \mathbf{S_2O_3}$ |
|

 Thiodenitrification:
(Schippers and Jorgensen, 2002; Volkov, 1984) $5H_2S + 8NO_3^- + 2OH^- \rightarrow 5SO_4^{2-} + 4N_2 + 6H_2O$ |  $hs\_no3 = K\_hs\_no3 * \mathbf{H_2S} * \mathbf{NO_3}$ |
| POM  sulfate reduction 1st and 2d stages (Boudreau, 1996):
$(CH_2O)_{106}(NH_3)_{16}H_3PO_4 + 53SO_4^{2-} \rightarrow 106HCO_3^- + 16NH_3 + H_3PO_4 + 53H_2S$ |

 $so4\_rd\_PM = K\_so4\_rd * F\_sox * F\_snx * \mathbf{SO_4} * \mathbf{PON}$
$s2o3\_rd\_PM = K\_s2o3\_rd * F\_sox * F\_snx * \mathbf{S_2O_3} * \mathbf{PON}$
$F\_sox = 1.  \mathbf{O_2} - s\_omso\_o2))$ |

| | |
|---|---|
| |  $- 0.5 * (1. + \tanh(\sim\!\!NO_3 - s_{omso\,no}\sim))$**NO$_3$ $-$ s_omso_no3))** |
| | DcPM__SO4 =  $* (s4_{rd\,PM} + s23_{rd\,PM})^{\frac{16}{53}} * $ (so4_rd_PM + s2o3_rd_PM) |
| DOM sulfate reduction 1st and 2d stages (Boudreau, 1996): |  |
| |  |
| | so4_rd_DM = K_so4_rd * F_sox * F_snx * **SO$_4$** * **DON** |
| | s2o3_rd_DM = K_s2o3_rd * F_sox * F_snx * **S$_2$O$_3$** * **DON** |
| | DcDM__SO4 =  $* (s4_{rd\,DM} + s23_{rd\,DM})^{\frac{16}{53}} * $ (so4_rd_PM + s2o3_rd_PM) |

| | |
|---|---|
|  |  |
|   |    |
|  |   |

| | |
|---|---|
| | 0.5\*s0_no3 – s23_ox – 0.4\*sulfido – 2\*CaCO₃_prec + 2\*CaCO₃ |
| | **Silicate** |
| Dissolution of particulate Si (Popova, Srokosz, 2009) | $sipart\_diss = Si_{part} * K_{si_{part\_diss}}$ |

**2.3. Carbon and Alkalinity**

| Name of Process, reference, reaction | Parameterization in the model |
|---|---|
| $CaCO_3$ formation/dissolution (Luff et al., 2001):
 $Ca^{2+} + CO_3^{2} \leftrightarrow CaCO_3$ | $caco3\_form = K\_caco3\_form * \max\left(0, \left(\frac{\mathbf{Ca^{2+}} * \mathbf{CO_3}}{K\_caco3} - 1\right)\right)$

 $caco3\_diss = K\_caco3\_diss * \mathbf{CaCO_3} * \max\left(0, \left(1 - \frac{\mathbf{Ca^{2+}} * \mathbf{CO_3}}{K\_caco3}\right)\right)^{4.5}$ |
| $CH_4$ formation from PON, methanogenesis (Boudreau, 1996) :
 $(CH_2O)_{106}(NH_3)_{16}H_3PO_4 \, \square$
 $53CO_2 + 53CH_4 + 16NH_3 + H_3PO_4$ | $DcPM\_CH4 = K\_PON\_ch4 * F\_sox * F\_snx * F\_ssx * \mathbf{CH_4} * \mathbf{PON}$
 $F\_sox = 1 - 0.5 * (1. + \tanh(\mathbf{O_2} - s\_omso\_o2))$
 $F\_snx = 1 - 0.5 * (1. + \tanh(\mathbf{NO_3} - s\_omso\_no3))$
 $F\_ssx = 1 - 0.5 * (1. + \tanh(\mathbf{SO_4} - s\_omch\_so4))$ |
| $CH_4$ formation from DON, methanogenesis (Boudreau, 1996) | $DcDM\_CH4 = K\_DON\_ch4 * F\_sox * F\_snx * F\_ssx * \mathbf{CH_4} * \mathbf{DON}$ |
| $CH_4$ oxidation by $O_2$ (Boudreau, 1996) :
 $CH_4 + 2O_2 + \rightarrow CO_2 + 2H_2O$ | $ch4\_o2 = K\_ch4\_o2 * \mathbf{CH_4} * \mathbf{O_2}$ |

| | |
|---|---|
| Alkalinity changes
 (Dickson, 1992; Wolf-Gladrow et al., 2007) | $dAlk = -\,Nitrif1 \,+\, Denitr2\_PM + Denitr2\_DM + 2*(so4_{rd} + s2o3_{rd}) + mn\_ox1 - 3 * mn\_ox2 + 3 * mn\_rd1 - mn\_rd2 - 2 * mns\_form + 2 * mns\_diss - 2 * mnco3\_form + 2 * mnco3\_diss + 26.5 * (DcDM_{Mn} + DcPM_{Mn}) - 2 * fe\_ox1 - fe\_ox2 + 2 * fe\_rd - fes\_form + fes\_diss - 2 * fes\_ox - 2 * fes2\_ox + 53 * (DcDM_{Fe} + DcPM_{Fe}) - 0.5 * Disprop + s0\_ox - 0.5 * s0\_no3 - s2o3\_ox - 0.4 * hs\_no3 - 2 * caco3\_form + 2 * caco3\_diss + GrowthPhy * \left(\dfrac{LimNO3}{LimN}\right) - GrowthPhy * \left(\dfrac{LimNH4}{LimN}\right)$ |

**2.4. Ecosystem processes**

| Name of Process, reference, reaction | Parameterization in the model |
|---|---|
| **Phytoplankton** | |
|  | $Iz = Io * e^{(-k_{Emay}*z)} * e^{(-kc*turbid*0.0001)}$ |
| Influence of the irradiance on photosynthesis | $LimLight = \left(\dfrac{Iz}{Iopt}\right) * e^{(1-Iz/Iopt)}$ |
| Influence of temperature on photosynthesis | $LimT = e^{(bm*t-cm)}$ |
| Dependence of photosynthesis on P | $\cancel{LimP = \dfrac{(PO_4/Phy)^2}{(KPO_4*NkP)^2+(PO_4/Phy)^2}}\ LimP = \dfrac{(PO_4/Phy)^2}{(K\_po4\_lim*r\_n\_p)^2+(PO_4/Phy)^2}$ |
| Dependence of photosynthesis on $NO_3$ | $\cancel{LimNO_3 = \dfrac{((NO_3+NO_2)/Phy)^2}{KNO_3{}^2+((NO_3+NO_2)/Phy)^2}}\ LimNO_3 = \dfrac{((NO_3+NO_2)/Phy)^2}{K\_nox\_lim^2+((NO_3+NO_2)/Phy)^2}\exp\left(-K\_psi\dfrac{(NH_4/Phy)^2}{K\_nh4\_lim^2+(NH_4/Phy)^2}\right)$ |
| Dependence of photosynthesis on $NH_4$ | $\cancel{LimNH_4 = \dfrac{(NH_4/Phy)^2}{KNH_4{}^2+(NH_4/Phy)^2}}\ LimNH_4 = \dfrac{\left(\frac{NH_4}{Phy}\right)^2}{K\_nh4\_lim^2+\left(\frac{NH_4}{Phy}\right)^2}\left(1 - \exp\left(-K\_psi\dfrac{(NH_4/Phy)^2}{K\_nh4\_lim^2+(NH_4/Phy)^2}\right)\right)$ |
| Influence of N on photosynthesis | $LimN = LimNO_3 + LimNH_4$ |
| Growth of phytoplankton | $GrowthPhy = \cancel{KNF}K\_phy\_gro * LimLight * LimT * \min(LimP, LimN) * \cancel{Phy}\textbf{Phy}$ |
| Excretion rate  of phytoplankton | $ExcrPhy = \cancel{K_{FD} * Phy}K\_phy\_exc * \textbf{Phy}$ |
| Phytoplankton mortality rate | $MortPhy = (\cancel{KFP}K\_phy\_mrt + 0.45 * (0.5 - 0.5 * \tanh(\cancel{O_2}\textbf{O}_2 - 60)) + 0.45 * (0.5 - 0.5 * \tanh(\cancel{O_2}\textbf{O}_2 - 20))) * \cancel{Phy}\textbf{Phy}$ |
| **Heterotrophs** | |

| Grazing of Heterotrophs | Grazing = GrazPhy + GrazPOP + GrazBact |
|---|---|
| Grazing of Het. on phytoplankton | $$\text{GrazPhy} = \text{KFZ} * \text{Het} * \frac{(\text{Phy}/(\text{Het}+0.0001))^2}{K_{FY}^2 + (\text{Phy}/(\text{Het}+0.0001))^2} \text{GrazPhy}$$ $$= \text{K\_het\_phy\_gro} * \mathbf{Het} * \frac{(\mathbf{Phy}/(\mathbf{Het}+10^{-4}))^2}{\text{K\_het\_phy\_lim}^2 + (\mathbf{Phy}/(\mathbf{Het}+10^{-4}))^2}$$ |
| Grazing of Het. on detritus | $$\text{GrazPOP} = \text{KPZ} * \text{Het} * \frac{(\frac{\text{PON}}{\text{Het}+0.0001})^2}{(\text{KPP})^2 + (\frac{\text{PON}}{\text{Het}+0.0001})^2} \text{GrazPOP}$$ $$= \text{K\_het\_pom\_gro} * \mathbf{Het} * \frac{(\frac{\mathbf{PON}}{\mathbf{Het}+10^{-4}})^2}{\text{K\_het\_pom\_lim}^2 + (\frac{\mathbf{PON}}{\mathbf{Het}+10^{-4}})^2}$$ |
| Grazing of Het. on bacteria | GrazBact = GrazBaae + GrazBaan + GrazBhae + GrazBhan |
| Grazing of Het. on bacteria autotrophic aerobic | $$\text{GrazBaae} = \text{KPZ} * \text{Het} * \frac{(\text{Baae}/(\text{Het}+0.0001))^2}{\text{limGrazBac}^2 + (\text{Baae}/(\text{Het}+0.0001))^2} \text{GrazBaae}$$ $$= \text{K\_het\_pom\_gro} * \mathbf{Het} * \frac{(\mathbf{Baae}/(\mathbf{Het}+10^{-4}))^2}{\text{limGrazBac}^2 + (\mathbf{Baae}/(\mathbf{Het}+10^{-4}))^2}$$ |
| Grazing of Het. on bacteria autotrophic anaerobic | $$\text{GrazBaan} = 0.5 * \text{KPZ} * \text{Het} * \frac{(\text{Baan}/(\text{Het}+0.0001))^2}{\text{limGrazBac}^2 + (\text{Baan}/(\text{Het}+0.0001))^2} \text{GrazBaan}$$ $$= 0.5 * \text{K\_het\_pom\_gro} * \mathbf{Het} * \frac{(\mathbf{Baan}/(\mathbf{Het}+10^{-4}))^2}{\text{limGrazBac}^2 + (\mathbf{Baan}/(\mathbf{Het}+10^{-4}))^2}$$ |

| | |
|---|---|
| Grazing of Het on bacteria heterotrophic aerobic | $\text{GrazBhae} = \text{KPZ} * \text{Het} * \dfrac{(\text{Bhae}/(\text{Het} + 0.0001))^2}{\text{limGrazBac}^2 + (\text{Bhae}/(\text{Het} + 0.0001))^2}\text{GrazBhae}$ $$= \text{K\_het\_pom\_gro} * \textbf{Het} * \frac{(\textbf{Bhae}/(\textbf{Het} + 10^{-4}))^2}{\text{limGrazBac}^2 + (\textbf{Bhae}/(\textbf{Het} + 10^{-4})^2}$$ |
| Grazing of Het on bacteria heterotrophic anaerobic | $\text{GrazBhan} = 1.3 * \text{KPZ} * \text{Het} * \dfrac{(\text{Bhan}/\text{Het} + 0.0001)^2}{\text{limGrazBac}^2 + (\text{Bhan}/\text{Het} + 0.0001)^2}\text{GrazBhan}$ $$= 1.3 * \text{K\_het\_pom\_gro} * \textbf{Het} * \frac{(\textbf{Bhan}/\textbf{Het} + 0.0001)^2}{\text{limGrazBac}^2 + (\textbf{Bhan}/\textbf{Het} + 10^{-4})^2}$$ |
| Respiration rate of Het | $\text{RespHet} = \text{KZN} * \text{Het}\,\text{K\_het\_res} * \textbf{Het} * (0.5 + 0.5 * \tanh(O_2 - 20)\tanh(\textbf{O}_2 - 20))$ |
| Mortality of Het | $\text{MortHet} = \text{Het} * \big(0.25 + 0.3 * (0.5 - 0.5 * \tanh(O_2 - 20)) + 0.45$ $* (0.5 + 0.4 * \tanh(H_2S - 10))\big)\text{MortHet}$ $$= \textbf{Het} * \begin{pmatrix} 0.25 + 0.3 * \big(0.5 - 0.5 * \tanh(\textbf{O}_2 - 20)\big) \\ + 0.45 * \big(0.5 + 0.4 * \tanh(\textbf{H}_2\textbf{S} - 10)\big) \end{pmatrix}$$ |
| **Bacteria** | |
| Growth rate of Bacteria aerobic autotrophic | $\text{ChemBaae} = (\text{Nitrif1} + \text{Nitrif2} + mn_{ox} + fe_{ox} + s23_{ox} + s0_{ox} + \text{anammox}) * \text{k\_Baae\_gro} * \text{Baae}$ $* \min(\dfrac{(\text{NH}_4 / ((\text{Baae} + 0.0001)^2}{\text{limBaae}^2 + (\text{NH}_4 / (\text{Baae} + 0.0001))^2},$ $\dfrac{(\text{PO}_4 / (\text{Baae} + 0.0001))^2}{\text{limBaae}^2 + (\text{PO}_4 / (\text{Baae} + 0.0001))^2})(\text{ChemBaae}$ $= \text{Nitrif1} + \text{Nitrif2} + \text{mn\_ox1} + \text{fe\_ox1} + \text{s2o3\_ox} + \text{s0\_ox} + \text{anammox}) * \text{k}_{\text{Baae}_{gro}} * \textbf{Baae}$ $* \min(\dfrac{(\textbf{NH}_4 / ((\textbf{Baae} + 10^{-4})^2}{\text{limBaae}^2 + (\textbf{NH}_4 / (\textbf{Baae} + 10^{-4}))^2}, \dfrac{(\textbf{PO}_4 / (\textbf{Baae} + 10^{-4}))^2}{\text{limBaae}^2 + (\textbf{PO}_4 / (\textbf{Baae} + 10^{-4}))^2})$ |

| | |
|---|---|
| Rate of mortality of Bacteria aerobic autotrophic | $\text{MortBaae} = \text{K\_Baae\_mrt} + \text{K\_Baae\_mrt\_h2s} * 0.5 * \left(1 - \tanh(1 - \mathbf{H_2S})\right) * \mathbf{Baae}^2$ |
| Growth rate of Bacteria aerobic heterotrophic | $\text{HetBhae} = (\text{DcPM\_O2} + \text{DcDM\_O2}) * \text{K\_Bhae\_gro} * \mathbf{Bhae} * \dfrac{(\mathbf{DON}/(\mathbf{Bhae}+10^{-4}))^2}{\text{limBhae}^2+(\mathbf{DON}/(\mathbf{Bhae}+10^{-4}))^2}$ |
| Rate of mortality of Bacteria aerobic heterotrophic | $\text{MortBhae} = \text{K\_Bhae\_mrt} + \text{K\_Bhae\_mrt\_h2s} * \mathbf{Bhae} * 0.5 * (1 - \tanh(1 - \mathbf{H_2S}))$ |
| Growth rate of Bacteria anaerobic autotrophic | $\text{ChemBaan} = (\text{mn\_rd1} + \text{mn\_rd2} + \text{fe\_rd} + \text{hs\_ox} + \text{hs\_no3}) * \text{K\_Baan\_gro} * \mathbf{Baan} * \min\left(\dfrac{(\mathbf{NH4}/(\mathbf{Baan} + 10^{-4}))^2}{\text{limBaan}^2 + (\mathbf{NH4}/(\mathbf{Baan} + 10^{-4}))^2}\right.$ |
| Rate of mortality of Bacteria anaerobic autotrophic | $\text{MortBaan} = \text{K\_Baan\_mrt} * \mathbf{Baan}$ |
| Growth rate of Bacteria anaerobic heterotrophic | $\text{HetBhan} = (\text{DcPM\_NOX} + \text{DcDM\_NOX} + \text{DcDM\_Mn} + \text{DcPM\_Mn} + \text{DcDM\_Fe} + \text{DcPM\_Fe} + \text{DcDM\_SO4} + \text{DcPMSO4} + \text{DcDM\_CH4} + \text{DcPM\_CH4}) * \text{K\_Bhan\_gro} * \mathbf{Bhan} * \dfrac{(\mathbf{DON}/(\mathbf{Bhan} + 10^{-4}))^2}{\text{limBhan}^2 + (\mathbf{DON}/(\mathbf{Bhan} + 10^{-4}))^2}$ |

| | |
|---|---|
| Rate of mortality of Bacteria anaerobic heterotrophic | MortBhan $= \mathrm{k_{Bhan_{mrt}}} + \mathrm{K_{Bhan_{mrt}O_2}} * \mathbf{Bhan}$ K_Bhan_mrt + K_Bhan_mrt_o2 $* \mathbf{Bhan} * (0.5 + 0.5 * \tanh(1 - O_z O_2))$ |
| Summarized OM mineralization | $Dc_{OM_{total}} = DcDM_{O_2} + DcPM_{O_2} + DcPM_{NOX} + DcDM_{NOX} + DcDM_{Mn} + DcPM_{Mn} + DcDM_{Fe} + DcPM_{Fe} + DcDM_{SO_4} + DcDM_{SO_4}$ Dc_OM_total $=$ DcDM_O2 $+$ DcPM_O2 $+$ DcPM_NOX $+$ DcDM_NOX $+$ DcDM_Mn $+$ DcPM_Mn $+$ DcDM_Fe $+$ DcPM_Fe $+$ DcDM_SO4 $+$ DcPM_SO4 $+ 0.5 *$ (DcDM_CH4 $+$ DcPM_CH4) |

1 **Table 3. Parameters names, notations, values and units of the coefficients used in the model**

2 **Table 3.1. Nutrients and oxygen**

| Parameter | Notation | Units | |
|---|---|---|---|
| **Manganese** | | | |
| Specific rate of Mn(II) to Mn(III) oxidation with $O_2$ | $K_{mn\_ox}$ | $d^{-1}$ | |
| Specific rate of Mn(IV) to Mn(III) reduction with $H_2S$ | $K_{mn\_rd}$ | $d^{-1}$ | 0.5 |
| Specific rate of Mn(III) to Mn(IV) oxidation with $O_2$ | $K_{mn\_ox2}$ | $d^{-1}$ | 0.2 |
| Specific rate of Mn(III) to Mn(II) reduction with $H_2S$ | $K_{mn\_rd2}$ | $d^{-1}$ | 1.0 |
| Specific rate of formation of MnS from Mn(II) and $H_2S$ | $K_{mns\_form}$ | $d^{-1}$ | $1*10^{-5}$ |
| Specific rate of dissolution of MnS to Mn(II) and $H_2S$ | $K_{mns\_diss}$ | $d^{-1}$ | $5*10^{-4}$ |
| Conditional equilibrium constant for MnS | $K_{mns}$ | M | 1500 |
| Conditional equilibrium constant for $MnCO_3$ | $K_{mnco3}$ | M | 15. |
| Specific rate of $MnCO_3$ dissolution | $K_{mnco3\_diss}$ | $d^{-1}$ | $7*10^{-4}$ |
| Specific rate of $MnCO_3$ formation | $K_{mnco3\_form}$ | $d^{-1}$ | $3*10^{-4}$ |
| Specific rate of DON Oxidation with Mn(IV) | $K_{DON\_Mn}$ | $d^{-1}$ | $1*10^{-3}$ |
| Specific rate of PON Oxidation with Mn(IV) | $K_{PON\_Mn}$ | $d^{-1}$ | $1*10^{-3}$ |
| Threshold value of Mn(II) oxidation | $s_{mnox\_mn2}$ | µM Mn | 0.01 |
| Threshold value of Mn(III) oxidation | $s_{mnox\_mn3}$ | µM Mn | 0.01 |
| Threshold value of Mn(IV) reduction | $s_{mnrd\_mn4}$ | µM Mn | 0.01 |
| Threshold value of Mn(III) reduction | $s_{mnrd\_mn3}$ | µM Mn | 0.01 |
| **Iron** | | | |
| Specific rate of Fe(II) to Fe(III) oxidation with $O_2$ | $K_{fe\_ox}$ | $d^{-1}$ | 0.5 |
| Specific rate of Fe(II) to Fe(III) oxidation with $MnO_2$ | $K_{fe\_ox2}$ | $d^{-1}$ | $1*10^{-3}$ |
| Specific rate of Fe(III) to Fe(II) reduction with $H_2S$ | $K_{fe\_rd}$ | $d^{-1}$ | 0.5 |
| Conditional equilibrium constant for FeS | $K_{FeS}$ | µM | 2510 |
| Specific rate of FeS formation from Fe(II) and $H_2S$ | $K_{Fes\_form}$ | $d^{-1}$ | $5*10^{-4}$ |
| Specific rate of DON oxidation with Fe(III) | $K_{DON\_Fe}$ | $d^{-1}$ | $5*10^{-5}$ |
| Specific rate of PON oxidation with Fe(III) | $K_{PON\_Fe}$ | $d^{-1}$ | $1*10^{-5}$ |
| Specific rate of $FeS_2$ formation by reaction of FeS with $H_2S$ | $K_{FeS2\_form}$ | $d^{-1}$ | $1*10^{-6}$ |
| Specific rate of $FeS_2$ oxidation with $O_2$ | $K_{FeS2\_ox}$ | $d^{-1}$ | $4.4*10^{-4}$ |
| Threshold value of Fe(II) reduction | $s_{feox\_fe2}$ | µM Fe | $1*10^{-3}$ |
| Threshold value of Fe(III) reduction | $s_{ferd\_fe3}$ | µM Fe | 0.01 |

| | | |
|---|---|---|
| **** | | |
|  | $K_{\_hs\_ox}$ | $d^{-1}$ |
|  | $K_{\_s0\_ox}$ | $d^{-1}$ |
|  | $K_{\_S0\_NO3}$ | $d^{-1}$ |
|  | $K_{\_s23\_ox}$ | $d^{-1}$ |
|  | $K_{\_s23\_NO3}$ | $d^{-1}$ |
|  | $K_{\_s4\_rd}$ | $d^{-1}$ |
|  | $K_{\_s23\_rd}$ | $d^{-1}$ |
|  | $K_{dispro}$ | $d^{-1}$ |
| **Nitrogen** | | |
| Specific rate of DON oxidation of with O₂ | K_DON_ox | $d^{-1}$ |
| Specific rate of PON oxidation of with O₂ | K_PON_ox | $d^{-1}$ |
| Temperature control threshold coefficient for OM decay | Tda | ℃ |
| Temperature control coefficient for OM decay | beta_da | |
| Half-saturation constant of O₂ for OM mineralization | K_omox_o2 | μM |
| Specific rate of autolysis, PON to DON | K_PON_DON | $d^{-1}$ |
|  | $K_{psi}$ | – |
| Half saturation constant for uptake of NO₃+NO₂ | K_nox_lim | μM |
| Half saturation constant for uptake of NH₄ | K_nh4_lim | μM |
| Strength of NH4 inhibition of NO3 uptake constant | K_psi | - |
| Specific rate of the 1st stage of nitrification | K_nitrif1 | $d^{-1}$ |
| Specific rate of the 2d stage of nitrification | K_nitrif2 | $d^{-1}$ |
| Specific rate of 1st stage of denitrification | K_denitr1 | $d^{-1}$ |
| Specific rate of 2d stage of denitrification | K_denitr2 | $d^{-1}$ |
| Half-saturation of NO₃ for OM denitrification | k_omno_no3 | μM N |
| Half-saturation of NO₂ for OM denitrification | k_omno_no2 | μM N |
| Specific rate of thiodenitrification | K_hs_no3 | $\mu M^{-1} d^{-1}$ |
| Specific rate of anammox | K_annamox | $d^{-1}$ |
| **Oxygen** | | |
| Half-saturation constant for nitrification | O2s_nf | μM |
| Half-saturation constant for denitrification anammox, Mn reduction | O2s_dn | μM |
| Threshold value of O₂ for OM mineralization | s_omox_o2 | μM |
| Threshold value of O₂ for OM denitrification | s_omno_o2 | μM |
| Threshold value of O₂ for OM sulfate reduction | s_omso_o2 | μM |

| Description | Symbol | Unit | Value | Reference |
|---|---|---|---|---|
| Threshold value of NO for OM sulfate reduction | s_omso_no3 | μM | 5 | 5 (Yakushev, 2013) |
| **Stoichiometric coefficients** | | | | |
| N/P | r_n_p | - | 16 | (Richards, 1965) |
| O/N | r_o_n | - | 6.625 | (Richards, 1965) |
| C/N | r_c_n | - | 8 | (Richards, 1965) |
| Si/N | r_si_n | - | 1 | (Richards, 1965) |
| Fe/N | r_fe_n | - | 26.5 | (Boudreau, 1996) |
| Mn/N | r_mn_n | - | 13.25 | (Boudreau, 1996) |
| **Phosphorus** | | | | |
| Half-saturation constant for uptake of $PO_4$ by phytoplankton | $K_{PO4}$ K_po4_lim | μM | | |
| Fe/P ratio in complexes  with Fe oxides | r_fe_p | | | |
| Mn/P ratio in complexes  with Mn(III) | r_mn_p | | | |
|  | | | | |
|  | $O_{2s\_nf}$ | μM | | |
|  | $O_{2s\_dn}$ | μM | | |
|  | $s_{omox\_o2}$ | μM | | |
|  | $s_{omno\_o2}$ | μM | | |
|  | $s_{omso\_o2}$ | μM | 25 | |
|  | $s_{omso\_no}$ | μM | 5 | |
|  | $k_{mnoxO2}$ | μM O | 2 | |
|  | | | | |
|  | $K_{CaCO3\_diss}$ | $d^{-1}$ | 3 | |
|  | $K_{CaCO3\_prec}$ | $d^{-1}$ | $1*10^{-4}$ | |
| **Silicon** | | | | |
| Specific rate of Si dissolution | K_Sipart sipart_diss | $d^{-1}$ | | |
|  | | | | |
|  | $k_{Baae\_gro}$ | $d^{-1}$ | | |
|  | $k_{Baae\_mrt}$ | $d^{-1}$ | | |
|  | $k_{Baae\_mrt\_h2s}$ | $d^{-1}$ | | |
|  | $k_{Bhae\_gro}$ | $d^{-1}$ | | |
|  | $k_{Bhae\_mrt}$ | $d^{-1}$ | | |
|  | $k_{Bhae\_mrt\_h2s}$ | $d^{-1}$ | | |
|  | $k_{Baan\_gro}$ | $d^{-1}$ | | |

| Description | Symbol | Units | Value |
|---|---|---|---|
| Baan specific rate of mortality | $k_{\_Baan\_mrt}$ | $d^{-1}$ | $5*10^{-3}$ |
| Bhan maximum specific growth rate | $k_{\_Bhan\_gro}$ | $d^{-1}$ | 0.1 |
| Bhan specific rate of mortality | $k_{\_Bhan\_mrt}$ | $d^{-1}$ | $5*10^{-3}$ |
| Bhan increased specific rate of mortality due to $O_2$ | $k_{\_Bhan\_mrt\_o2}$ | $d^{-1}$ | 0.899 |
| **Phytoplankton** | | | |
| Maximum specific growth rate | $K_{NF}$ | $d^{-1}$ | 2.6 |
| Extinction coefficient | $K\_Erlov$ | $m^{-1}$ | 0.05 |
| Surface irradiance | $I_0$ | $W \cdot m^{-2}$ | 80 |
| Optimal irradiance | $I_{opt}$ | $W \cdot m^{-2}$ | 25 |
| 1$^{st}$ coefficient for growth dependence on t | bm | $°C^{-1}$ | 0.12 |
| 2d coefficient for growth dependence on t | cm | – | 1.4 |
| Attenuation constant for the self-shading effect | Kc | $m^2$ mmol $N^{-1}$ | 0.03 |
| Specific respiration rate | $K_{FN}$ | $d^{-1}$ | 0.05 |
| Specific rate of mortality | $K_{FP}$ | $d^{-1}$ | 0.10 |
| Specific rate of excretion | $K_{FD}$ | $d^{-1}$ | 0.05 |
| **Heterotrophs** | | | |
| Maximum specific rate of grazing of Het on Phy | $K_{FZ}$ | $d^{-1}$ | 1.0 |
| Half-saturation constant for uptake of Si by phytoplankton | $K_{FY}$K_si_lim | | |
| Maximum specific rate of grazing of Het on POP | $K_{PZ}$ | $d^{-1}$ | |
| Specific respiration rate | $K_{ZN}$ | $d^{-1}$ | |
| Fe/P ratio in complexes with Fe oxides | $K_{PP}$r_fe_si | | |
| Maximum specific rate of mortality of Het | $K_{ZP}$ | $d^{-1}$ | |
| Food absorbency for Heterotrophs | $U_z$ | – | 0.5 |
| Ratio between dissolved and particulate excretes of Heterotrophs | $H_z$ | – | 0.5 |
| Limiting parameter for bacteria grazing by Het | limGrazBac | – | 2 |
| Limiting parameter for bacteria anaerobic heterotrophic | limBhan | – | 2 |
| Limiting parameter for bacteria aerobic heterotrophic | limBhae | – | 5 |
| Limiting parameter for bacteria anaerobic autotrophic | limBaan | – | 2 |
| Limiting parameter for bacteria aerobic autotrophic | limBaae | – | 1 |

| Sinking | | | |
|---|---|---|---|
| Rate of sinking of Phy | $W_{Phy}$ | $m \cdot d^{-1}$ | 0.1 |
| Rate of sinking of Het | $W_{Het}$ | $m \cdot d^{-1}$ | 1.0 |
| Rate of sinking of bacteria (Bhe,Bae,Bha,Baa) | $W_{Bact}$ | $m \cdot d^{-1}$ | 0.4 |
| Rate of sinking of detritus (POP, PON) | $W_{sed}$ | $m \cdot d^{-1}$ | 5 |
| Rate of accelerated sinking of particles with settled Mn hydroxides | $W_M$ | $m \cdot d^{-1}$ | 7 |

Table

1 **Table 4. Rates of biogeochemical production/consumption of the model compartments**

2 **3.2. Redox metals and sulfur**

| Parameter | Notation | Units | Value | Reference ranges |
|---|---|---|---|---|
| **Manganese** | | | | |
| Specific rate of Mn(II) oxidation to Mn(III) with $O_2$ | K_mn_ox1 | $d^{-1}$ | 0.1 | 0.18-1.9 M/yr; (Tebo, 1991) 2 $d^{-1}$; (Yakushev et al., 2007) |
| Specific rate of Mn(IV) reduction to Mn(III) with $H_2S$ |  | $d^{-1}$ | 0.5 | 22 $d^{-1}$; (Yakushev et al., 2007) |

| | | | | | |
|---|---|---|---|---|---|
| | | $)/2.7$ $(mn\_ox$ $+mn\_rd$ $)/0.67$ $+fe\_rd$ $/2.7+(mn\_ox2+$ $mn\_rd2$ $)$ $/0.67$K_mn_rd1 | | | |

| | | | | | |
|---|---|---|---|---|---|
| **Particulate Organic Nitrogen (PON)** | | $R_{PON}=$ autolis -DcPM_O2 -DcPM_NOX -DcPM_SO4 -DcPM_Mn -DcPM_Fe +MortBaut +MortBautA +MortBhet +MortBhetA +MortPhy +MortHet +Grazing *(1- Uz)*(1- Hz) -GrazPOP | | | |
| **Dissolved Organic Phosphorus (DON)** | | $R_{DON}=$ autolis -DcDM_O2 -DcDM_NOX -DcDM_ -DcDM_Mn -DcDM_Fe -Hetero -HeteroA+ExcrPhy +Grazing *(1- Uz)*Hz | | | |
| **Ammonia (NH₄)** | $R_{NH4}=$ -DcDM_O2 +DcPM_O2 +DcPM_NOX +DcDM_NOX + DcDM_Mn +DcPM_Mn +DcDM_Fe +DcPM_Fe +DcDM_SO4 +DcPM_SO4 -Nitrif1 -anammox +RespHet -GrowthPhy *(LimNH4 /LimN ) -Chemos -ChemosA Specific_rate_of_Mn(III)_oxidation to Mn(IV) with O₂ | K_mn_ox2 | d⁻¹ | 0.2 | 18 d⁻¹; (Yakushev et al., 2008) 1 |
| **Nitrite (NO₂)** | | $R_{NO2}=$ Nitrif1 -Nitrif2 +Denitr1 -Denitr2 -anammox -GrowthPhy *(LimNO3 /LimN ) (NO2 /(NO2 + NO3)) | | | |
| **Nitrate (NO₃)** | | $R_{NO3}=$ Nitrif2 -Denitr1 -sulphido *1.25 -GrowthPhy *(LimNO3 /LimN ) (NO3 /(NO2 + NO3)) | | | |
| **Hydrogen sulphide (H₂S)** Specific rate of Mn(III) reduction to Mn(II) with H₂S | $R_{H2S}=$ -0.5 mn_rd -0.5 mn_rd2 -0.5 fe_rd | d⁻¹ | 1 | 0.96-3.6 M/yr; (Tebo, 1991) 2 d⁻¹; (Yakushev et al., 2007) |

Margin annotations: Deleted Cells; Formatted Table; Inserted Cells; Inserted Cells; Formatted: Justified, Don't adjust space between Latin and Asian text, Don't adjust space between Asian text and numbers; Inserted Cells; Inserted Cells; Formatted: Justified, Don't adjust space between Latin and Asian text, Don't adjust space between Asian text and numbers; Formatted: Right: -0,04 cm, Tab stops: 1,3 cm, Left; Inserted Cells; Inserted Cells; Inserted Cells

| | | | | |
|---|---|---|---|---|
| | |  K_mn_rd2 | | |
|  Specific rate of formation of MnS from Mn(II) and $H_2S$ | K_mns_form | $d^{-1}$ |  | |
|  Specific rate of dissolution of MnS to Mn(II) and $H_2S$ | K_mns_diss | $d^{-1}$ |  | |

Inserted Cells

| | | | | |
|---|---|---|---|---|
| | | | *Disprop* +0.5 *s4_rd* -0.5 *s23_rd* – *S23_NO3* $5*10^{-4}$ | |
| Solubility product for MnS | K_mns | M | 1500 | |
| Solubility product for MnCO$_3$ | K_mnco3 | M | 1 | |
| Specific rate of MnCO$_3$ formation | K_mnco3_form | d$^{-1}$ | $3*10^{-4}$ | $10^{-4}$ – $10^{-2}$ mol/g yr; (Wersin, 1990); (Wollast, 1990) |
| Specific rate of MnCO$_3$ dissolution | K_mnco3_diss | d$^{-1}$ | $7*10^{-4}$ | $10^{-2}$ – $10^{3}$ yr$^{-1}$; (Wersin, 1990; Wollast, 1990) |
| Specific rate of MnCO3 oxidation | *R$_{sor}$ = sulphido s4_rd +s23_ox +fes_ox +mns_ox* K_mnco3_ox | d$^{-1}$ | $27*10^{-4}$ | |
| Specific rate of DON Oxidation with Mn(IV) | K_DON_Mn | d$^{-1}$ | $1*10^{-3}$ | $1*10^{-3}$ (Yakushev et al., 2007) |
| Specific rate of PON Oxidation with Mn(IV) | K_PON_Mn | d$^{-1}$ | $1*10^{-3}$ | $1*10^{-3}$ (Yakushev et al., 2007) |
| Threshold value of Mn(II) oxidation | s_mnox_mn2 | µM Mn | *R$_{Mn2}$ = mn_ox* | $1*10^{-2}$ (Yakushev et al., 2007) |

| | | | +mn_rd2 mns_form +mns_ox +fe_ox2 +2.DcDM_Mn +2.DcPM_Mn $1*10^{-2}$ | |
|---|---|---|---|---|
| Threshold value of Mn(III) oxidation | s_mnox_mn3 | µM Mn | $1*10^{-2}$ | $1*10^{-2}$ (Yakushev et al., 2007) |
|  (Threshold value of Mn(IV)) reduction | s_mnrd_mn4 | µM Mn | $R_{Mn4}$ = mn_ox2 mn_rd fe_ox2 2.DcDM_Mn 2.DcPM_Mn | $1*10^{-2}$(Yakushev et al., 2007) |

Inserted Cells

Inserted Cells

Inserted Cells

| | | | $1*10^{-2}$ | |
|---|---|---|---|---|
| Threshold value of Mn(III)) reduction | $R_{Mn3} =$  s_mnrd_mn3 | µM Mn | $1*10^{-2}$ | $1*10^{-2}$ (Yakushev et al., 2007) |
| Half saturation constant of Mn oxidation | $R_{MnS} =$ K_mnox_o2 | µM O$_2$ | 2 | 2 (Yakushev et al., 2007) |
| **Iron** | | | | |
| Specific rate of Fe(II) to Fe(III) oxidation with O$_2$ | $R_{Fe2} =$  K_fe_ox1 | d$^{-1}$ | 0.5 | $2*10^9$ M/yr; (Boudreau, 1996); 4 d$^{-1}$; (Yakushev et al., 2007) |
| Specific rate of Fe(II) to Fe(III) oxidation with MnO$_2$ | $R_{Fe3} =$  K_fe_ox2 | d$^{-1}$ | $1*10^{-3}$ | $10^4$–$10^8$ M/yr; (Boudreau, 1996); 1 d$^{-1}$; (Yakushev et al., 2007) |
| Specific rate of Fe(III) to Fe(II) reduction with H$_2$S | K_fe_rd | d$^{-1}$ | 0.5 | $1*10^4$ M/yr;(Boudreau, 1996); 0.05d$^{-1}$; (Yakushev et al., 2007) |

| Description | Symbol | Units | Value | Reference |
|---|---|---|---|---|
| Solubility product for FeS | K_fes | μM | 2510 | |
| Specific rate of FeS formation from Fe(II) and $H_2S$ | $R_{FeS} =$ K_fes_form  | $d^{-1}$ | $5*10^{-4}$ | $5*10^{-6}$–$10^{-3}$ M/yr; (Boudreau, 1996; Hunter et al., 1998); (Bektursunova and L'Heureux, 2011) |
|  $R_{DIC} = (DcDM\_O2+DcPM\_O2+DcPM\_NOX+DcDM\_NOX +DcDM\_SO4+DcPM\_SO4+ DcDM\_Mn +DcPM\_Mn +DcDM\_Fe +DcPM\_Fe -Chemos-ChemosA-GrowthPhy+RespHet) CkN$ Specific rate of FeS dissolution to Fe(II) and $H_2S$ | K_fes_diss | $d^{-1}$ | $1*10^{-6}$ | $1*10^{-3}$ $yr^{-1}$ (Hunter et al., 1998); (Bektursunova and L'Heureux, 2011) |
| Specific rate of FeS oxidation with $O_2$ | $R_{Alk} = dAlk$ K_fes_ox | $d^{-1}$ | $1*10^{-3}$ | $2*10^7$–$3*10^5$ M/yr; (Boudreau, 1996); (Van Cappellen and Wang, 1996) |
| Specific rate of DON oxidation with Fe(III) | $R_{Phy} = \dfrac{Grov}{K\_DON\_fe}$ | $d^{-1}$ | $5*10^{-5}$ | $5*10^{-5}$ (Yakushev et al., 2007) |
| Specific rate of PON oxidation with Fe(III) | $R_{Het} = Grazing*U_z - MortHet - K_{ZN}*Het$ K_PON_fe | $d^{-1}$ | $1*10^{-5}$ | $1*10^{-5}$ (Yakushev et al., 2007) |
| Specific rate of $FeS_2$ formation by reaction of FeS with $H_2S$ | $R_{Bhe} = \dfrac{C_{Bhe}}{K\_fes2\_form}$ | $d^{-1}$ | $1*10^{-6}$ | $8.9*10^{-6}$ M/day; (Rickard and Luther, 1997) |
| Specific rate of $FeS_2$ oxidation with $O_2$ | $R_{Bae} = \dfrac{C_{Bae}}{K\_fes2\_ox}$ | $d^{-1}$ | $4.4*10^{-4}$ | |
| Threshold value of Fe(II) reduction | $R_{Bha} = \dfrac{C_{Bha}}{s\_feox\_fe2}$ | μM Fe | $1*10^{-3}$ | $1*10^{-3}$(Yakushev et al., 2007) |
| Threshold value of Fe(III) reduction | | μM | $1*10^{-2}$ | $1*10^{-2}$(Yakushev et al., 2007) |

| | | | | |
|---|---|---|---|---|
| | $R_{Baa} = C_{Baa}$ s_ferd_fe3 | Fe | | |
| Solubility product for FeCO$_3$ | K_feco3 | d$^{-1}$ | 15 | |
| Specific rate of FeCO$_3$ dissolution | K_feco3_dis s | d$^{-1}$ | $7*10^{-4}$ | $2.5*10^{-1}$–$10^{-2}$ yr$^{-1}$; (Wersin, 1990; Wollast, 1990) |
| Specific rate of FeCO$_3$ formation | K_feco3_for m | d$^{-1}$ | $3.4*10^{-4}$ | $10^{-6}$–$10^{-2}$ mol/g yr; (Boudreau, 1996; Wersin, 1990; Wollast, 1990) |
| Specific rate of FeCO$_3$ oxidation with O$_2$ | K_feco3_ox | d$^{-1}$ | $2.7*10^{-3}$ | |
| **Sulfur** | | | | |
| Specific rate of H$_2$S oxidation to S$^0$ of with O$_2$ | K_hs_ox | d$^{-1}$ | 0.5 | 0.5 (Yakushev et al., 2007) |
| Specific rate of S$^0$ oxidation of with O$_2$ | K_s0_ox | d$^{-1}$ | $2*10^{-2}$ | $2*10^{-2}$(Yakushev et al., 2007) |
| Specific rate of S$^0$ oxidation of with NO$_3$ | K_s0_no3 | d$^{-1}$ | 0.9 | 0.9 (Yakushev et al., 2007) |
| Specific rate of S$_2$O$_3$ oxidation with O$_2$ | K_s2o3_ox | d$^{-1}$ | $1*10^{-2}$ | $1*10^{-2}$(Yakushev et al., 2007) |
| Specific rate of S$_2$O$_3$ oxidation with NO$_3$ | K_s2o3_no3 | d$^{-1}$ | $1*10^{-2}$ | $1*10^{-2}$(Yakushev et al., 2007) |
| Specific rate of OM reduction with sulfate | K_so4_rd | d$^{-1}$ | $5*10^{-6}$ | $5*10^{-6}$(Yakushev et al., 2007) |
| Specific rate of OM reduction with thiosulfate | K_s2o3_rd | d$^{-1}$ | $1*10^{-3}$ | $1*10^{-3}$(Yakushev et al., 2007) |
| Specific rate of S$^0$ disproportionation | K_s0_disp | d$^{-1}$ | $1*10^{-3}$ | $1*10^{-3}$(Yakushev et al., 2007) |
| Half-saturation of Mn reduction | K_mnrd_hs | µMS | 1 | 1 (Yakushev et al., 2007) |
| Half-saturation of Fe reduction | K_ferd_hs | µMS | 1 | 1 (Yakushev et al., 2007) |

Split Cells
Deleted Cells
Formatted Table
Inserted Cells
Inserted Cells
Formatted
Formatted
Formatted
Formatted
Formatted
Formatted
Formatted
Deleted Cells
Deleted Cells
Formatted
Formatted Table
Formatted
Formatted
Formatted
Formatted
Inserted Cells
Formatted
Formatted Table
Formatted
Deleted Cells
Deleted Cells
Formatted
Deleted Cells
Formatted
Formatted
Formatted
Formatted
Formatted
Formatted Table
Formatted
Formatted
Formatted
Formatted
Formatted
Deleted Cells
Formatted
Deleted Cells
Formatted

**Table 3.3. Carbon**

| Parameter | Notation | | Units | Value | Reference ranges |
|---|---|---|---|---|---|
| Specific rate of CaCO$_3$ dissolution |  K_caco3_diss |  |  | 3 | wide ranges are given in (Luff et al., 2001) |

| | modelled | observed | modelled | observed | modelled | observed |
|---|---|---|---|---|---|---|
|  Specific rate of CaCO$_3$ formation |  K_caco3_prec |  d$^{-1}$ | | |  2*10$^{-4}$ |  wide ranges are given in (Luff et al., 2001) |
| Solubility product constant for CaCO3 | K_caco3 | | | | | Calculated as a function of T, S (Roy et al., 1993b) |
|  Specific rate of CH$_4$ formation from DON |  K_DON_ch4 |  |  |  d$^{-1}$ |  | *10$^{-5}$ | (Lopes et al., 2011) |
| Specific rate of CH$_4$ formation from PON | K_PON_ch4 | d$^{-1}$ | | | 1*10$^{-5}$ | (Lopes et al., 2011) |
|  Specific rate of CH$_4$ oxidation with O$_2$ |  K_ch4_o2 |  uM$^{-1}$d$^{-1}$ | |   | 0.14 (Lopes et al., 2011) | – |
|  Specific rate of CH$_4$ oxidation with SO$_4$ |  K_ch4_so4 |  uM$^{-1}$d$^{-1}$ | | 0.0000274 | (0.0274 m3 /mol-1 day-1 (Lopes et al., 2011) |  + |  |

| | modelled | observed | modelled | observed | modelled | observed |
|---|---|---|---|---|---|---|
| PO$_4$ | 0 4 | 0 6 | 5 50 | 5 100 | 0.01 0.2 | 1 1.5 |
| Si | 0 300 | 1 150 | 200 1400 | 100 600 | 0.5 15 | 1.7 11 |
| pH | 7.0 8.3 | 6.9 8.4 | 6.6 7.3 | 7.1 7.9 | | |
| DIC | – | – | – | – | 1 20 | 5 50 |
| Alk | 2200 2300 | 2000 3300 | 3000 4900 | 2000 20000 | 1 5 | 3 200 |
| MnII | 0 1.5 | 0 12 | 8 20 | 5 200 | 0.01 0.1 | 3 20 |
| FeII | 0 1.5 | 0 1.6 | 8 40 | 0.5 100 | 0.01 0.1 | 0.03 1 |

Table *Pakhomova et al., 2007; Almroth et al., 2011; Queirós et al., 2014

**3.4. Ecosystem parameters**

**Table 6.** **Typical concentrations (ranges of concentrations) of alkalinity in the seawater (in μM).**

| Parameter | Notation | Units | Value, μM, in oxic (/ anoxic) eond it | Source Reference ranges |
|-----------|----------|-------|-------|-------|
| | | | | |

|  | ions |  |
|---|---|---|
| **Bacteria** | | |
| A~rCO2~Baae maximum specific growth rate | K_Baae_gro | d⁻¹ |

| $2$ | $3.06$ |
|---|---|
| $2*10^{-2}$ | Typical seawater |
| $(HCO_3^-)$ | (Yakushev et al., 2000) |
| $+2(CO_3^{2-})$ | (Millero, 1979) |
| $+2*10^{-2}$ | $200$ |

| | | | | | | |
|---|---|---|---|---|---|---|
| | | | | | 7) | ) |

$A_B$ | [B(OH)$_4^-$] | | 100 | | Typical seawater (Millero, 2008; Dickson, 2012) |

| | | | | | 6-75K_Baae_mrt | Seawater (Canfielde | 5*10$^{-3}$ | 5*10$^{-3}$ (Yakushev et al., 2007) |
|---|---|---|---|---|---|---|---|---|
| $A_{DOM}$ | DOC (15%Baae specific rate of total)mortality | | | | | | | |

| | | | |
|---|---|---|---|
| | et al., 2005; Kepkay, 2000) d-1 | | |
| A~TH2S~Baae increased specific rate of mortality due to H$_2$S | [HS-] [K-B | d-1 | < 0.450.899 | Black Sea (Volkov et al., |

**Inserted Cells**

| | | | |
|---|---|---|---|
| | aae_mrt_h2s | | 20 00) 0.8 99 (Yakushev et al., 2007) |
| As;Bhae maximum specific growth rate | {SiO(OH)3-}K_Bhae_gro | < 10 / 40 d-1 | Black Sea (Volkov et al.,) | 0.5 (Yakushev et al., 2007) |

Inserted Cells

| | | | | | |
|---|---|---|---|---|---|
| | | | | | 2000)-0.5 |
| A₋ₚₒ₄Bhae specific rate of mortality | K_Bhae_mrt | d=1 | t[HPO₄²⁻]+2[PO₄²⁻]+1–t[H₃PO₄ | 2/72*10⁻²(Yakushevet al.; 20 | Black Sea (Volkov et al.; 20 |

Inserted Cells

Inserted Cells

After: 0 pt, Don't adjust s
between Latin and Asian to
adjust space between Asia

Formatted Table

Before: 0 pt, After: 0 pt

cm, Space Before: 0 pt, A

Deleted Cells

| | | | 0.07) | 0) |
| | | }12*10=2 | | |

| A$_{TNH3}$Bhae increased specific rate of mortality due to H$_2$S | [NH$_3$]K_Bhae_mrt_h2s | <24/5d=1 | Black Sea (Volkov et al., 2000 | 0.799 (Yakushev et al., 2007) |

After: 0 pt, Don't adjust s
between Latin and Asian t
adjust space between Asia

Before: 0 pt, After: 0 pt

**Inserted Cells**

| | | | |
|---|---|---|---|
| | | 0.7999 | |
| $A_{THF}$ | [HF] (total F) | 68 | Typical seawater (Millero, 2008; Dickson, 2012) |
| [OH⁻] | [OH⁻] | 8 | Typical seawater (Millero, 2008; Dickson, 2012) |
| [H⁺]Baan maximum specific growth rate | | [H⁺] = 0. | Typic |

**Inserted Cells**

| | | | | |
|---|---|---|---|---|
| | $+1)(pH = 8.151\ NBS)$ K_Baan_gro | [1] | 0071 2 | al seawater (Millero, 1979) 0.12 (Yakushev et al., 2007) |
| Baan specific rate of mortality | K_Baan_mrt | $d^{-1}$ | $1.2*10^{-2}$ | $1.2*10^{-2}$ (Yakushev et al., 20 |

| Description | Symbol | Units | Value | Reference |
|---|---|---|---|---|
| | | | | 07) |
| A~TSO4~Bhan maximum specific growth rate | [HSO4 = ]K_Bhan_gro | $d^{-1}$ | 0.0015-0.0019 | Typical seawater (Hoffmann et al., 2008) 0.19 (Yakushev et al., 2007) |
| Bhan specific rate of mortality | K_Bhan_mrt | $d^{-1}$ | $7*10^{-3}$ | $7*10^{-3}$ (Yakushev et al., 2007) |
| Bhan increased specific rate of mortality due to $O_2$ | K | d | 0 | 0.8 |

Inserted Cells

[revised manuscript text omitted]

**TABLES**

---

## Author Response (AR2)

**Author's response**

**Bottom RedOx Model (BROM, v.1.1): a coupled benthic-pelagic model for simulation of water and sediment biogeochemistry. By E.V.Yakushev, E.A.Protsenko, J.Bruggeman, P.Wallhead, S.V.Pakhomova, S.Yakubov, R.G.J.Bellerby, R.-M. Couture**

> *Topical Editor Decision: Reconsider after major revisions (26 Oct 2016) by Dr. Didier Roche*
> *Comments to the Author:*
> *Dear Dr. Yakushev and co-workers,*
>
> *We have received a new assessment of Dr. Guy Munhoven who recommends minor revisions with a list of comments to be implemented. I do not expect more comments on the current version of your manuscript.*
>
> *I thus recommend that you implement the changes suggested and send me a final version. At this stage, I have to request major revisions (only option in the system) but I follow the advice of minor revisions suggested.*
>
> *With best regards,*
> *Didier Roche*

Dear Didier Roche,

Thank you for your letter and decision. We implemented the changes suggested by the reviewer and uploaded the MS in the system.

These are the answers to 2 comments of Dr. Guy Munhoven, that he made for the text:

> *The model does not seem to include a diffusive boundary layer (which would be typically 1 mm thick in the deep sea), and which impedes the exchange of solutes between sediment porewaters and the overlying seawater. Is this not necessary in the setting chosen here?*

That is correct but we believe that the upper layer of the sediments should serve as an adequate barrier in this case (see text). The code does provide the possibility of an explicit DBL (see section 2.2.7). However, in our view, when considering layers on the scale of 1 mm or less, which is less than the roughness of a typical SWI, it seems somewhat arbitrary whether this barrier layer is modelled as strictly within the sediments or not. Also, note that the upper sediment layer (or DBL) will only be the limiting step if the fluff layer turbulent diffusivity is sufficiently large and the upper layer/DBL is sufficiently thick. In our example we have a fixed linear profile of $D_{eBBL}$ with a value of 3e-7 $m^2s^{-1}$ on the upper interface of the fluff layer. This implies of diffusive timescale or $(0.03)^2/3e-7 = 3000$ seconds. The diffusive timescale across the upper sediment layer (or DBL if we had one) is $(0.5e-3)^2/1e-9 = 250$ seconds, so in this example the limiting step for solute diffusion is in fact the fluff layer.

> *Eq. B2 is only correct if $D\_Bi^{intra}$ (please notice \*intra\*) is the same for all constituents of the solid phase, else $D\_Bi^{intra}$ does not cancel out.*

We believe that Eq. B2 is correct for any valid set of $D\_Bi^{intra}$ because intraphase mixing cannot \*by definition\* affect the total solid volume (see Meysman et al., 2005, section 5.1). Intraphase mixing merely exchanges equal volumes of sediment at different depth levels, resulting in net fluxes of individual components if there are gradients in individual

components, but no net flux of total solid volume (Meysman et al., 2005, Figure 1). In fact this definition can be considered to impose an additional constraint on the set of D_Bi^{intra} (Meysman et al., 2005, Eqn 55).

*I recommend to put these sample brom.yaml and fabm.yaml files into supplementary material. Here, there are line-feed problems (lines are folded that should not, I suppose). T would be more helpful to have correctly formatted versions in separate files.*

Done. These files are given as supplementary material.

p. 61 (figures): the top and bottom parts better had to have the same horizontal scales, else, they are really difficult to interpret

Our goal was to simulate the distributions of both in the water column and the sediments, but the typical concentrations change in orders of magnitude (i.e. for Mn or PO4). To demonstrate this we have  to set  different horizontal concentration scales for the water and the bottom parts of the figures.

Can you please inform us about the further steps of our paper progress.

Please not, that the paper's title and the list of the authors differ from the initial ones available at the web site. Can you please correct this!

On behalf the of the authors,

Evgeniy
* * *
Dr. Evgeniy V.Yakushev
Senior Researcher
Section for Marine Biogeochemistry and Oceanography
Norwegian Institute for Water Research (NIVA)
Gaustadalléen 21
N-0349 Oslo
Norway

Phone: +47 98294079
Switchboard phone/fax: +47 22185100 / 5200